# Offline Primal-Dual Reinforcement Learning for Linear MDPs

## Abstract

Offline Reinforcement Learning (RL) aims to learn a near-optimal policy from a fixed dataset of transitions collected by another policy. This problem has attracted a lot of attention recently, but most existing methods with strong theoretical guarantees are restricted to finite-horizon or tabular settings. In constrast, few algorithms for infinite-horizon settings with function approximation and minimal assumptions on the dataset are both sample and computationally efficient. Another gap in the current literature is the lack of theoretical analysis for the average-reward setting, which is more challenging than the discounted setting. In this paper, we address both of these issues by proposing a primal-dual optimization method based on the linear programming formulation of RL. Our key contribution is a new reparametrization that allows us to derive low-variance gradient estimators that can be used in a stochastic optimization scheme using only samples from the behavior policy. Our method finds an $\varepsilon$-optimal policy with $O(\varepsilon^{-4})$ samples, improving on the previous $O(\varepsilon^{-5})$, while being computationally efficient for infinite-horizon discounted and average-reward MDPs with realizable linear function approximation and partial coverage. Moreover, to the best of our knowledge, this is the first theoretical result for average-reward offline RL.

## 1  Introduction

We study the setting of Offline Reinforcement Learning (RL), where the goal is to learn an $\varepsilon$-optimal policy without being able to interact with the environment, but only using a fixed dataset of transitions collected by a *behavior policy*. Learning from offline data proves to be useful especially when interacting with the environment can be costly or dangerous [16].

In this setting, the quality of the best policy learnable by any algorithm is constrained by the quality of the data, implying that finding an optimal policy without further assumptions on the data is not feasible. Therefore, many methods [23, 33] make a *uniform coverage* assumption, requiring that the behavior policy explores sufficiently well the whole state-action space. However, recent work [17, 31] demonstrated that *partial coverage* of the state-action space is sufficient. In particular, this means that the behavior policy needs only to sufficiently explore the state-actions visited by the optimal policy.

Moreover, like its online counterpart, modern offline RL faces the problem of learning efficiently in environments with very large state spaces, where function approximation is necessary to compactly represent policies and value functions. Although function approximation, especially with neural networks, is widely used in practice, its theoretical understanding in the context of decision-making is still rather limited, even when considering *linear* function approximation.

In fact, most existing sample complexity results for offline RL algorithms are limited either to the tabular and finite horizon setting, by the uniform coverage assumption, or by lack of computational efficiency — see the top section of Table 1 for a summary. Notable exceptions are the recent works of

| Algorithm | Partial Coverage | Polynomial Sample Complexity | Polynomial Computational Complexity | Function Approximation | Infinite Horizon | |
|---|---|---|---|---|---|---|
| | | | | | Discounted | Average-Reward |
| FQI [23] | ✗ | ✓ | ✓ | ✓ | ✓ | ✗ |
| Rashidinejad et al. [31] | ✓ | ✓ | ✓ | ✗ | ✓ | ✗ |
| Jin et al. [14] Zanette et al. [38] | ✓ | ✓ | ✓ | ✓ | ✗ | ✗ |
| Uehara & Sun [32] | ✓ | ✓ | ✗ | ✓ | ✓ | ✗ |
| Cheng et al. [9] | ✓ | $O(\varepsilon^{-5})$ | superlinear | ✓ | ✓ | ✗ |
| Xie et al. [36] | ✓ | $O(\varepsilon^{-5})$ | $O(n^{7/5})$ | ✓ | ✓ | ✗ |
| **Ours** | ✓ | $O(\varepsilon^{-4})$ | $O(n)$ | ✓ | ✓ | ✓ |

Table 1: Comparison of existing offline RL algorithms. The table is divided horizontally in two sections. The upper section qualitatively compares algorithms for easier settings, that is, methods for the tabular or finite-horizon settings or methods which require uniform coverage. The lower section focuses on the setting considered in this paper, that is computationally efficient methods for the infinite horizon setting with function approximation and partial coverage.

Xie et al. [36] and Cheng et al. [9] who provide computationally efficient methods for infinite-horizon discounted MDPs under realizable linear function approximation and partial coverage. Despite being some of the first implementable algorithms, their methods work only with discounted rewards, have superlinear computational complexity and find an $\varepsilon$-optimal policy with $O(\varepsilon^{-5})$ samples – see the bottom section of Table 1 for more details. Therefore, this work is motivated by the following research question:

*Can we design a linear-time algorithm with polynomial sample complexity for the discounted and average-reward infinite-horizon settings, in large state spaces under a partial-coverage assumption?*

We answer this question positively by designing a method based on the linear-programming (LP) formulation of sequential decision making [20]. Albeit less known than the dynamic-programming formulation [3] that is ubiquitous in RL, it allows us to tackle this problem with the powerful tools of convex optimization. We turn in particular to a relaxed version of the LP formulation [21, 2] that considers action-value functions that are linear in known state-action features. This allows to reduce the dimensionality of the problem from the cardinality of the state space to the number of features. This relaxation still allows to recover optimal policies in *linear MDPs* [37, 13], a structural assumption that is widely employed in the theoretical study of RL with linear function approximation.

Our algorithm for learning near-optimal policies from offline data is based on primal-dual optimization of the Lagrangian of the relaxed LP. The use of saddle-point optimization in MDPs was first proposed by Wang & Chen [34] for *planning* in small state spaces, and was extended to linear function approximation by Chen et al. [8], Bas-Serrano & Neu [1], and Neu & Okolo [26]. We largely take inspiration from this latter work, which was the first to apply saddle-point optimization to the *relaxed* LP. However, primal-dual planning algorithms assume oracle access to a transition model, whose samples are used to estimate gradients. In our offline setting, we only assume access to i.i.d. samples generated by a possibly unknown behavior policy. To adapt the primal-dual optimization strategy to this setting we employ a change of variable, inspired by Nachum & Dai [24], which allows easy computation of unbiased gradient estimates.

**Notation.** We denote vectors with bold letters, such as $\boldsymbol{x} \doteq [x_1, \ldots, x_d]^\top \in \mathbb{R}^d$, and use $\boldsymbol{e}_i$ to denote the $i$-th standard basis vector. We interchangeably denote functions $f : \mathcal{X} \to \mathbb{R}$ over a finite set $\mathcal{X}$, as vectors $\boldsymbol{f} \in \mathbb{R}^{|\mathcal{X}|}$ with components $f(x)$, and use $\geq$ to denote element-wise comparison. We denote the set of probability distributions over a measurable set $\mathcal{S}$ as $\Delta_\mathcal{S}$, and the probability simplex in $\mathbb{R}^d$ as $\Delta_d$. We use $\sigma : \mathbb{R}^d \to \Delta_d$ to denote the softmax function defined as $\sigma_i(\boldsymbol{x}) \doteq e^{x_i} / \sum_{j=1}^d e^{x_j}$. We use upper-case letters for random variables, such as $S$, and denote the uniform distribution over a finite set of $n$ elements as $\mathcal{U}(n)$. In the context of iterative algorithms, we use $\mathcal{F}_{t-1}$ to denote the sigma-algebra generated by all events up to the end of iteration $t-1$, and use the shorthand notation $\mathbb{E}_t [\cdot] = \mathbb{E} [\cdot | \mathcal{F}_{t-1}]$ to denote expectation conditional on the history. For nested-loop algorithms, we write $\mathcal{F}_{t,i-1}$ for the sigma-algebra generated by all events up to the end of iteration $i-1$ of round $t$, and $\mathbb{E}_{t,i} [\cdot] = \mathbb{E} [\cdot | \mathcal{F}_{t,i-1}]$ for the corresponding conditional expectation.

## 2 Preliminaries

We study discounted Markov decision processes [MDP, 29] denoted as $(\mathcal{X}, \mathcal{A}, p, r, \gamma)$, with discount factor $\gamma \in [0, 1]$ and finite, but potentially very large, state space $\mathcal{X}$ and action space $\mathcal{A}$. For every state-action pair $(x, a)$, we denote as $p(\cdot \mid x, a) \in \Delta_{\mathcal{X}}$ the next-state distribution, and as $r(x, a) \in [0, 1]$ the reward, which is assumed to be deterministic and bounded for simplicity. The transition function $p$ is also denoted as the matrix $\boldsymbol{P} \in \mathbb{R}^{|\mathcal{X} \times \mathcal{A}| \times |\mathcal{X}|}$ and the reward as the vector $\boldsymbol{r} \in \mathbb{R}^{|\mathcal{X} \times \mathcal{A}|}$. The objective is to find an *optimal policy* $\pi^* : \mathcal{X} \to \Delta_{\mathcal{A}}$. That is, a stationary policy that maximizes the normalized expected return $\rho(\pi^*) \doteq (1 - \gamma)\mathbb{E}_{\pi^*}[\sum_{t=0}^{\infty} r(X_t, A_t)]$, where the initial state $X_0$ is sampled from the initial state distribution $\nu_0$, the other states according to $X_{t+1} \sim p(\cdot|X_t, A_t)$ and where the notation $\mathbb{E}_{\pi}[\cdot]$ is used to denote that the actions are sampled from policy $\pi$ as $A_t \sim \pi(\cdot|X_t)$. Moreover, we define the following quantities for each policy $\pi$: its state-action value function $q^{\pi}(x, a) \doteq \mathbb{E}_{\pi}[\sum_{t=0}^{\infty} \gamma^t r(X_t, A_t) \mid X_0 = x, A_0 = a]$, its value function $v^{\pi}(x) \doteq \mathbb{E}_{\pi}[q^{\pi}(x, A_0)]$, its state occupancy measure $\nu^{\pi}(x) \doteq (1 - \gamma)\mathbb{E}_{\pi}[\sum_{t=0}^{\infty} \mathbb{1}\{X_t = x\}]$, and its state-action occupancy measure $\mu^{\pi}(x, a) \doteq \pi(a|x)\nu^{\pi}(x)$. These quantities are known to satisify the following useful relations, more commonly known respectively as Bellman's equation and flow constraint for policy $\pi$ [4]:

$$\boldsymbol{q}^{\pi} = \boldsymbol{r} + \gamma \boldsymbol{P} \boldsymbol{v}^{\pi} \qquad \boldsymbol{\nu}^{\pi} = (1 - \gamma)\boldsymbol{\nu}_0 + \gamma \boldsymbol{P}^{\intercal} \boldsymbol{\mu}^{\pi} \tag{1}$$

Given this notation, we can also rewrite the normalized expected return in vector form as $\rho(\pi) = (1 - \gamma)\langle \boldsymbol{\nu}_0, \boldsymbol{v}^{\pi} \rangle$ or equivalently as $\rho(\pi) = \langle \boldsymbol{r}, \boldsymbol{\mu}^{\pi} \rangle$.

Our work is based on the linear programming formulation due to Manne [19] (see also 29) which transforms the reinforcement learning problem into the search for an optimal state-action occupancy measure, obtained by solving the following Linear Program (LP):

$$\begin{aligned} \text{maximize} \quad & \langle \boldsymbol{r}, \boldsymbol{\mu} \rangle \\ \text{subject to} \quad & \boldsymbol{E}^{\intercal} \boldsymbol{\mu} = (1 - \gamma)\boldsymbol{\nu}_0 + \gamma \boldsymbol{P}^{\intercal} \boldsymbol{\mu} \\ & \boldsymbol{\mu} \geq 0 \end{aligned} \tag{2}$$

where $\boldsymbol{E} \in \mathbb{R}^{|\mathcal{X} \times \mathcal{A}| \times |\mathcal{X}|}$ denotes the matrix with components $\boldsymbol{E}_{(x,a),x'} \doteq \mathbb{1}\{x = x'\}$. The constraints of this LP are known to characterize the set of valid state-action occupancy measures. Therefore, an optimal solution $\boldsymbol{\mu}^*$ of the LP corresponds to the state-action occupancy measure associated to a policy $\pi^*$ maximizing the expected return, and which is therefore optimal in the MDP. This policy can be extracted as $\pi^*(a|x) \doteq \mu^*(x, a)/\sum_{\bar{a} \in \mathcal{A}} \mu^*(x, \bar{a})$. However, this linear program cannot be directly solved in an efficient way in large MDPs due to the number of constraints and dimensions of the variables scaling with the size of the state space $\mathcal{X}$. Therefore, taking inspiration from the previous works of Bas-Serrano et al. [2], Neu & Okolo [26] we assume the knowledge of a *feature map* $\varphi$, which we then use to reduce the dimension of the problem. More specifically we consider the setting of Linear MDPs [13, 37].

**Definition 2.1** (Linear MDP). An MDP is called linear if both the transition and reward functions can be expressed as a linear function of a given feature map $\varphi : \mathcal{X} \times \mathcal{A} \to \mathbb{R}^d$. That is, there exist $\psi : \mathcal{X} \to \mathbb{R}^d$ and $\boldsymbol{\omega} \in \mathbb{R}^d$ such that, for every $x, x' \in \mathcal{X}$ and $a \in \mathcal{A}$:

$$r(x, a) = \langle \boldsymbol{\varphi}(x, a), \boldsymbol{\omega} \rangle, \qquad p(x' \mid x, a) = \langle \boldsymbol{\varphi}(x, a), \boldsymbol{\psi}(x') \rangle.$$

We assume that for all $x, a$, the norms of all relevant vectors are bounded by known constants as $\|\boldsymbol{\varphi}(x, a)\|_2 \leq D_{\boldsymbol{\varphi}}$, $\|\sum_{x'} \boldsymbol{\psi}(x')\|_2 \leq D_{\boldsymbol{\psi}}$, and $\|\boldsymbol{\omega}\|_2 \leq D_{\boldsymbol{\omega}}$. Moreover, we represent the feature map with the matrix $\boldsymbol{\Phi} \in \mathbb{R}^{|\mathcal{X} \times \mathcal{A}| \times d}$ with rows given by $\boldsymbol{\varphi}(x, a)^{\intercal}$, and similarly we define $\boldsymbol{\Psi} \in \mathbb{R}^{d \times |\mathcal{X}|}$ as the matrix with columns given by $\boldsymbol{\psi}(x)$.

With this notation we can rewrite the transition matrix as $\boldsymbol{P} = \boldsymbol{\Phi}\boldsymbol{\Psi}$. Furthermore, it is convenient to assume that the dimension $d$ of the feature map cannot be trivially reduced, and therefore that the matrix $\boldsymbol{\Phi}$ is full-rank. An easily verifiable consequence of the Linear MDP assumption is that state-action value functions can be represented as a linear combinations of $\varphi$. That is, there exist $\boldsymbol{\theta}^{\pi} \in \mathbb{R}^d$ such that:

$$\boldsymbol{q}^{\pi} = \boldsymbol{r} + \gamma \boldsymbol{P} \boldsymbol{v}^{\pi} = \boldsymbol{\Phi}(\boldsymbol{\omega} + \boldsymbol{\Psi} \boldsymbol{v}^{\pi}) = \boldsymbol{\Phi} \boldsymbol{\theta}^{\pi}. \tag{3}$$

It can be shown that for all policies $\pi$, the norm of $\boldsymbol{\theta}^{\pi}$ is at most $D_{\boldsymbol{\theta}} = D_{\boldsymbol{\omega}} + \frac{D_{\boldsymbol{\psi}}}{1-\gamma}$ (cf. Lemma B.1 in 13). We then translate the linear program (2) to our setting, with the addition of the new variable $\boldsymbol{\lambda} \in \mathbb{R}^d$, resulting in the following new LP and its corresponding dual:

$$
\begin{array}{ll}
\text{maximize} & \langle \boldsymbol{\omega}, \boldsymbol{\lambda} \rangle \\
\text{subject to} & \boldsymbol{E}^{\mathsf{T}} \boldsymbol{\mu} = (1 - \gamma)\boldsymbol{\nu}_0 + \gamma \boldsymbol{\Psi}^{\mathsf{T}} \boldsymbol{\lambda} \\
& \boldsymbol{\lambda} = \boldsymbol{\Phi}^{\mathsf{T}} \boldsymbol{\mu} \\
& \boldsymbol{\mu} \geq 0.
\end{array}
\qquad (4)
\qquad
\begin{array}{ll}
\text{minimize} & (1 - \gamma)\langle \boldsymbol{\nu}_0, \boldsymbol{v} \rangle \\
\text{subject to} & \boldsymbol{\theta} = \boldsymbol{\omega} + \gamma \boldsymbol{\Psi} \boldsymbol{v} \\
& \boldsymbol{E} \boldsymbol{v} \geq \boldsymbol{\Phi} \boldsymbol{\theta}
\end{array}
\qquad (5)
$$

It can be immediately noticed how the introduction of $\boldsymbol{\lambda}$ did not change neither the set of admissible $\boldsymbol{\mu}$s nor the objective, and therefore did not alter the optimal solution. The Lagrangian associated to this set of linear programs is the function:

$$
\begin{aligned}
\mathfrak{L}(\boldsymbol{v}, \boldsymbol{\theta}, \boldsymbol{\lambda}, \boldsymbol{\mu}) &= (1 - \gamma)\langle \boldsymbol{\nu}_0, \boldsymbol{v} \rangle + \langle \boldsymbol{\lambda}, \boldsymbol{\omega} + \gamma \boldsymbol{\Psi} \boldsymbol{v} - \boldsymbol{\theta} \rangle + \langle \boldsymbol{\mu}, \boldsymbol{\Phi} \boldsymbol{\theta} - \boldsymbol{E} \boldsymbol{v} \rangle \\
&= \langle \boldsymbol{\lambda}, \boldsymbol{\omega} \rangle + \langle \boldsymbol{v}, (1 - \gamma)\boldsymbol{\nu}_0 + \gamma \boldsymbol{\Psi}^{\mathsf{T}} \boldsymbol{\lambda} - \boldsymbol{E}^{\mathsf{T}} \boldsymbol{\mu} \rangle + \langle \boldsymbol{\theta}, \boldsymbol{\Phi}^{\mathsf{T}} \boldsymbol{\mu} - \boldsymbol{\lambda} \rangle.
\end{aligned}
\qquad (6)
$$

It is known that finding optimal solutions $(\boldsymbol{\lambda}^\star, \boldsymbol{\mu}^\star)$ and $(\boldsymbol{v}^\star, \boldsymbol{\theta}^\star)$ for the primal and dual LPs is equivalent to finding a saddle point $(\boldsymbol{v}^\star, \boldsymbol{\theta}^\star, \boldsymbol{\lambda}^\star, \boldsymbol{\mu}^\star)$ of the Lagrangian function [5]. In the next section, we will develop primal-dual methods that aim to find approximate solutions to the above saddle-point problem, and convert these solutions to policies with near-optimality guarantees.

# 3 Algorithm and Main Results

This section introduces the concrete setting we study in this paper, and presents our main contributions.

We consider the offline-learning scenario where the agent has access to a dataset $\mathcal{D} = (W_t)_{t=1}^n$, collected by a behavior policy $\pi_B$, and composed of $n$ random observations of the form $W_t = (X_t^0, X_t, A_t, R_t, X_t')$. The random variables $X_t^0, (X_t, A_t)$ and $X_t'$ are sampled, respectively, from the initial-state distribution $\nu_0$, the discounted occupancy measure of the behavior policy, denoted as $\mu_B$, and from $p(\cdot \mid X_t, A_t)$. Finally, $R_t$ denotes the reward $r(X_t, A_t)$. We assume that all observations $W_t$ are generated independently of each other, and will often use the notation $\boldsymbol{\varphi}_t = \boldsymbol{\varphi}(X_t, A_t)$.

Our strategy consists in finding approximately good solutions for the LPs (4) and (5) using stochastic optimization methods, which require access to unbiased gradient estimates of the Lagrangian (Equation 6). The main challenge we need to overcome is constructing suitable estimators based only on observations drawn from the behavior policy. We address this challenge by introducing the matrix $\boldsymbol{\Lambda} = \mathbb{E}_{X, A \sim \mu_B} [\boldsymbol{\varphi}(X, A)\boldsymbol{\varphi}(X, A)^{\mathsf{T}}]$ (supposed to be invertible for the sake of argument for now), and rewriting the gradient with respect to $\boldsymbol{\lambda}$ as

$$
\begin{aligned}
\nabla_{\boldsymbol{\lambda}} \mathfrak{L}(\boldsymbol{\lambda}, \boldsymbol{\mu}; \boldsymbol{v}, \boldsymbol{\theta}) &= \boldsymbol{\omega} + \gamma \boldsymbol{\Psi} \boldsymbol{v} - \boldsymbol{\theta} = \boldsymbol{\Lambda}^{-1} \boldsymbol{\Lambda} \left( \boldsymbol{\omega} + \gamma \boldsymbol{\Psi} \boldsymbol{v} - \boldsymbol{\theta} \right) \\
&= \boldsymbol{\Lambda}^{-1} \mathbb{E} \left[ \boldsymbol{\varphi}(X_t, A_t) \boldsymbol{\varphi}(X_t, A_t)^{\mathsf{T}} \left( \boldsymbol{\omega} + \gamma \boldsymbol{\Psi} \boldsymbol{v} - \boldsymbol{\theta} \right) \right] \\
&= \boldsymbol{\Lambda}^{-1} \mathbb{E} \left[ \boldsymbol{\varphi}(X_t, A_t) \left( R_t + \gamma \boldsymbol{v}(X_t') - \langle \boldsymbol{\theta}, \boldsymbol{\varphi}(X_t, A_t) \rangle \right) \right].
\end{aligned}
$$

This suggests that the vector within the expectation can be used to build an unbiased estimator of the desired gradient. A downside of using this estimator is that it requires knowledge of $\boldsymbol{\Lambda}$. However, this can be sidestepped by a reparametrization trick inspired by Nachum & Dai [24]: introducing the parametrization $\boldsymbol{\beta} = \boldsymbol{\Lambda}^{-1} \boldsymbol{\lambda}$, the objective can be rewritten as

$$
\mathfrak{L}(\boldsymbol{\beta}, \boldsymbol{\mu}; \boldsymbol{v}, \boldsymbol{\theta}) = (1 - \gamma)\langle \boldsymbol{\nu}_0, \boldsymbol{v} \rangle + \langle \boldsymbol{\beta}, \boldsymbol{\Lambda} \left( \boldsymbol{\omega} + \gamma \boldsymbol{\Psi} \boldsymbol{v} - \boldsymbol{\theta} \right) \rangle + \langle \boldsymbol{\mu}, \boldsymbol{\Phi} \boldsymbol{\theta} - \boldsymbol{E} \boldsymbol{v} \rangle.
$$

This can be indeed seen to generalize the tabular reparametrization of Nachum & Dai [24] to the case of linear function approximation. Notably, our linear reparametrization does not change the structure of the saddle-point problem, but allows building an unbiased estimator of $\nabla_{\boldsymbol{\beta}} \mathfrak{L}(\boldsymbol{\beta}, \boldsymbol{\mu}; \boldsymbol{v}, \boldsymbol{\theta})$ without knowledge of $\boldsymbol{\Lambda}$ as

$$
\tilde{\boldsymbol{g}}_{\boldsymbol{\beta}} = \boldsymbol{\varphi}(X_t, A_t) \left( R_t + \gamma \boldsymbol{v}(X_t') - \langle \boldsymbol{\theta}, \boldsymbol{\varphi}(X_t, A_t) \rangle \right).
$$

In what follows, we will use the more general parametrization $\boldsymbol{\beta} = \Lambda^{-c} \boldsymbol{\lambda}$, with $c \in \{1/2, 1\}$, and construct a primal-dual stochastic optimization method that can be implemented efficiently in the offline setting based on the observations above. Using $c = 1$ allows to run our algorithm without knowledge of $\boldsymbol{\Lambda}$, that is, without knowing the behavior policy that generated the dataset, while using $c = 1/2$ results in a tighter bound, at the price of having to assume knowledge of $\boldsymbol{\Lambda}$.

Our algorithm (presented as Algorithm 1) is inspired by the method of Neu & Okolo [26], originally designed for planning with a generative model. The algorithm has a double-loop structure, where

---

**Algorithm 1** Offline Primal-Dual RL

---

**Input:** Learning rates $\alpha, \zeta, \eta$, initial points $\boldsymbol{\theta}_0 \in \mathbb{B}(D_\theta), \boldsymbol{\beta}_1 \in \mathbb{B}(D_\beta), \pi_1$, and data $\mathcal{D} = (W_t)_{t=1}^n$

**for** $t = 1$ **to** $T$ **do**

    Initialize $\boldsymbol{\theta}_{t,1} = \boldsymbol{\theta}_{t-1}$

    **for** $k = 1$ **to** $K - 1$ **do**

        Obtain sample $W_{t,k} = (X_{t,k}^0, X_{t,k}, A_{t,k}, X_{t,k}')$

        $\boldsymbol{\mu}_{t,k} = \pi_t \circ \left[ (1 - \gamma) \boldsymbol{e}_{X_{t,k}^0} + \gamma \langle \boldsymbol{\varphi}(X_{t,k}, A_{t,k}), \boldsymbol{\Lambda}^{c-1} \boldsymbol{\beta}_t \rangle \boldsymbol{e}_{X_{t,k}'} \right]$

        $\tilde{\boldsymbol{g}}_{\boldsymbol{\theta},t,i} = \boldsymbol{\Phi}^\top \boldsymbol{\mu}_{t,k} - \boldsymbol{\Lambda}^{c-1} \boldsymbol{\varphi}(X_{t,k}, A_{t,k}) \langle \boldsymbol{\varphi}(X_{t,k}, A_{t,k}), \boldsymbol{\beta}_t \rangle$

        $\boldsymbol{\theta}_{t,k+1} = \Pi_{\mathbb{B}(D_\theta)} (\boldsymbol{\theta}_{t,k} - \eta \tilde{\boldsymbol{g}}_{\boldsymbol{\theta},t,i})$    *// Stochastic gradient descent*

    **end for**

    $\boldsymbol{\theta}_t = \frac{1}{K} \sum_{k=1}^K \boldsymbol{\theta}_{t,k}$

    Obtain sample $W_t = (X_t^0, X_t, A_t, X_t')$

    $\boldsymbol{v}_t = \boldsymbol{E}^\top (\pi_t \circ \boldsymbol{\Phi} \boldsymbol{\theta}_t)$

    $\tilde{\boldsymbol{g}}_{\boldsymbol{\beta},t} = \boldsymbol{\varphi}(X_t, A) (R_t + \gamma \boldsymbol{v}_t(X_t') - \langle \boldsymbol{\varphi}(X_t, A_t), \boldsymbol{\theta}_t \rangle)$

    $\boldsymbol{\beta}_{t+1} = \Pi_{\mathbb{B}(D_\beta)} (\boldsymbol{\beta}_t + \zeta \tilde{\boldsymbol{g}}_{\boldsymbol{\beta},t})$    *// Stochastic gradient ascent*

    $\pi_{t+1} = \sigma(\alpha \sum_{i=1}^t \boldsymbol{\Phi} \boldsymbol{\theta}_i)$    *// Policy update*

**end for**

**return** $\pi_J$ with $J \sim \mathcal{U}(T)$.

---

at each iteration $t$ we run one step of stochastic gradient ascent for $\boldsymbol{\beta}$, and also an inner loop which runs $K$ iterations of stochastic gradient descent on $\boldsymbol{\theta}$ making sure that $\langle \boldsymbol{\varphi}(x,a), \boldsymbol{\theta}_t \rangle$ is a good approximation of the true action-value function of $\pi_t$. Iterations of the inner loop are indexed by $k$. The main idea of the algorithm is to compute the unbiased estimators $\tilde{\boldsymbol{g}}_{\boldsymbol{\theta},t,k}$ and $\tilde{\boldsymbol{g}}_{\boldsymbol{\beta},t}$ of the gradients $\nabla_{\boldsymbol{\theta}} \mathfrak{L}(\boldsymbol{\beta}_t, \boldsymbol{\mu}_t; \cdot, \boldsymbol{\theta}_{t,k})$ and $\nabla_{\boldsymbol{\beta}} \mathfrak{L}(\boldsymbol{\beta}_t, \cdot; \boldsymbol{v}_t, \boldsymbol{\theta}_t)$, and use them to update the respective variables iteratively. We then define a softmax policy $\pi_t$ at each iteration $t$ using the $\boldsymbol{\theta}$ parameters as $\pi_t(a|x) = \sigma \left( \alpha \sum_{i=1}^{t-1} \langle \boldsymbol{\varphi}(x,a), \boldsymbol{\theta}_i \rangle \right)$. The other higher-dimensional variables $(\boldsymbol{\mu}_t, \boldsymbol{v}_t)$ are defined symbolically in terms of $\boldsymbol{\beta}_t$, $\boldsymbol{\theta}_t$ and $\pi_t$, and used only as auxiliary variables for computing the estimates $\tilde{\boldsymbol{g}}_{\boldsymbol{\theta},t,k}$ and $\tilde{\boldsymbol{g}}_{\boldsymbol{\beta},t}$. Specifically, we set these variables as

$$v_t(x) = \sum_a \pi_t(a|x) \langle \boldsymbol{\varphi}(x,a), \boldsymbol{\theta}_t \rangle, \tag{7}$$

$$\mu_{t,k}(x,a) = \pi_t(a|x) \left( (1 - \gamma) \mathbb{1}\{X_{t,k}^0 = x\} + \gamma \langle \boldsymbol{\varphi}_{t,k}, \boldsymbol{\Lambda}^{c-1} \boldsymbol{\beta}_t \rangle \mathbb{1}\{X_{t,k}' = x\} \right). \tag{8}$$

Finally, the gradient estimates can be defined as

$$\tilde{\boldsymbol{g}}_{\boldsymbol{\beta},t} = \boldsymbol{\Lambda}^{c-1} \boldsymbol{\varphi}_t \left( R_t + \gamma v_t(X_t') - \langle \boldsymbol{\varphi}_t, \boldsymbol{\theta}_t \rangle \right), \tag{9}$$

$$\tilde{\boldsymbol{g}}_{\boldsymbol{\theta},t,k} = \boldsymbol{\Phi}^\top \boldsymbol{\mu}_{t,k} - \boldsymbol{\Lambda}^{c-1} \boldsymbol{\varphi}_{t,k} \langle \boldsymbol{\varphi}_{t,k}, \boldsymbol{\beta}_t \rangle. \tag{10}$$

These gradient estimates are then used in a projected gradient ascent/descent scheme, with the $\ell_2$ projection operator denoted by $\Pi$. The feasible sets of the two parameter vectors are chosen as $\ell_2$ balls of radii $D_\theta$ and $D_\beta$, denoted respectively as $\mathbb{B}(D_\theta)$ and $\mathbb{B}(D_\beta)$. Notably, the algorithm does not need to compute $v_t(x)$, $\mu_{t,k}(x,a)$, or $\pi_t(a|x)$ for all states $x$, but only for the states that are accessed during the execution of the method. In particular, $\pi_t$ does not need to be computed explicitly, and it can be efficiently represented by the single $d$-dimensional parameter vector $\sum_{i=1}^t \boldsymbol{\theta}_i$.

Due to the double-loop structure, each iteration $t$ uses $K$ samples from the dataset $\mathcal{D}$, adding up to a total of $n = KT$ samples over the course of $T$ iterations. Each gradient update calculated by the method uses a constant number of elementary vector operations, resulting in a total computational complexity of $O(|\mathcal{A}|dn)$ elementary operations. At the end, our algorithm outputs a policy selected uniformly at random from the $T$ iterations.

## 3.1  Main result

We are now almost ready to state our main result. Before doing so, we first need to discuss the quantities appearing in the guarantee, and provide an intuitive explanation for them.

Similarly to previous work, we capture the partial coverage assumption by expressing the rate of convergence to the optimal policy in terms of a *coverage ratio* that measures the mismatch between the behavior and the optimal policy. Several definitions of coverage ratio are surveyed by Uehara & Sun [32]. In this work, we employ a notion of *feature* coverage ratio for linear MDPs that defines coverage in feature space rather than in state-action space, similarly to Jin et al. [14], but with a smaller ratio.

**Definition 3.1.** Let $c \in \{1/2, 1\}$. We define the generalized coverage ratio as

$$C_{\varphi,c}(\pi^*; \pi_B) = \mathbb{E}_{(X^*, A^*) \sim \mu^{\pi^*}} [\varphi(X^*, A^*)]^\top \Lambda^{-2c} \mathbb{E}[\varphi(X^*, A^*)].$$

We defer a detailed discussion of this ratio to Section 6, where we compare it with similar notions in the literature. We are now ready to state our main result.

**Theorem 3.2.** *Given a linear MDP (Definition 2.1) such that $\theta^\pi \in \mathbb{B}(D_\theta)$ for any policy $\pi$. Assume that the coverage ratio is bounded $C_{\varphi,c}(\pi^*; \pi_B) \leq D_\beta$. Then, for any comparator policy $\pi^*$, the policy output by an appropriately tuned instance of Algorithm 1 satisfies $\mathbb{E}\left[\langle \mu^{\pi^*} - \mu^{\pi_{out}}, r \rangle\right] \leq \varepsilon$ with a number of samples $n_\epsilon$ that is $O\left(\varepsilon^{-4} D_\theta^4 D_\varphi^{8c} D_\beta^4 d^{2-2c} \log |\mathcal{A}|\right)$.*

The concrete parameter choices are detailed in the full version of the theorem in Appendix A. The main theorem can be simplified by making some standard assumptions, formalized by the following corollary.

**Corollary 3.3.** *Assume that the bound of the feature vectors $D_\varphi$ is of order $O(1)$, that $D_\omega = D_\psi = \sqrt{d}$ and that $D_\beta = c \cdot C_{\varphi,c}(\pi^*; \pi_B)$ for some positive universal constant $c$. Then, under the same assumptions of Theorem 3.2, $n_\varepsilon$ is of order $O\left(\frac{d^4 C_{\varphi,c}(\pi^*; \pi_B)^2 \log |\mathcal{A}|}{d^{2c}(1-\gamma)^4 \varepsilon^4}\right)$.*

## 4 Analysis

This section explains the rationale behind some of the technical choices of our algorithm, and sketches the proof of our main result.

First, we explicitly rewrite the expression of the Lagrangian (6), after performing the change of variable $\lambda = \Lambda^c \beta$:

$$\mathfrak{L}(\beta, \mu; v, \theta) = (1-\gamma)\langle \nu_0, v \rangle + \langle \beta, \Lambda^c(\omega + \gamma \Psi v - \theta) \rangle + \langle \mu, \Phi\theta - Ev \rangle \tag{11}$$

$$= \langle \beta, \Lambda^c \omega \rangle + \langle v, (1-\gamma)\nu_0 + \gamma \Psi^\top \Lambda^c \beta - E^\top \mu \rangle + \langle \theta, \Phi^\top \mu - \Lambda^c \beta \rangle. \tag{12}$$

We aim to find an approximate saddle-point of the above convex-concave objective function. One challenge that we need to face is that the variables $v$ and $\mu$ have dimension proportional to the size of the state space $|\mathcal{X}|$, so making explicit updates to these parameters would be prohibitively expensive in MDPs with large state spaces. To address this challenge, we choose to parametrize $\mu$ in terms of a policy $\pi$ and $\beta$ through the symbolic assignment $\mu = \mu_{\beta,\pi}$, where

$$\mu_{\beta,\pi}(x, a) \doteq \pi(a|x)\Big[(1-\gamma)\nu_0(x) + \gamma\langle \psi(x), \Lambda^c \beta \rangle\Big]. \tag{13}$$

This choice can be seen to satisfy the first constraint of the primal LP (4), and thus the gradient of the Lagrangian (12) evaluated at $\mu_{\beta,\pi}$ with respect to $v$ can be verified to be 0. This parametrization makes it possible to express the Lagrangian as a function of only $\theta, \beta$ and $\pi$ as

$$f(\theta, \beta, \pi) \doteq \mathfrak{L}(\beta, \mu_{\beta,\pi}; v, \theta) = \langle \beta, \Lambda^c \omega \rangle + \langle \theta, \Phi^\top \mu_{\beta,\pi} - \Lambda^c \beta \rangle. \tag{14}$$

For convenience, we also define the quantities $\nu_\beta = E^\top \mu_{\beta,\pi}$ and $v_{\theta,\pi}(s) \doteq \sum_a \pi(a|s) \langle \theta, \varphi(x, a) \rangle$, which enables us to rewrite $f$ as

$$f(\theta, \beta, \pi) = \langle \Lambda^c \beta, \omega - \theta \rangle + \langle v_{\theta,\pi}, \nu_\beta \rangle = (1-\gamma)\langle \nu_0, v_{\theta,\pi} \rangle + \langle \Lambda^c \beta, \omega + \gamma\Psi v_{\theta,\pi} - \theta \rangle. \tag{15}$$

The above choices allow us to perform stochastic gradient / ascent over the low-dimensional parameters $\theta$ and $\beta$ and the policy $\pi$. In order to calculate an unbiased estimator of the gradients, we first

observe that the choice of $\mu_{t,k}$ in Algorithm 1 is an unbiased estimator of $\mu_{\boldsymbol{\beta}_t,\pi_t}$:

$$\mathbb{E}_{t,k}\left[\mu_{t,k}(x,a)\right] = \pi_t(a|x)\Big((1-\gamma)\mathbb{P}(X_{t,k}^0 = x) + \mathbb{E}_{t,k}\left[\mathbb{1}\{X_{t,k}' = x\}\langle\boldsymbol{\varphi}_t,\boldsymbol{\Lambda}^{c-1}\boldsymbol{\beta}_t\rangle\right]\Big)$$

$$= \pi_t(a|x)\Big((1-\gamma)\nu_0(x) + \gamma\sum_{\bar{x},\bar{a}}\mu_B(\bar{x},\bar{a})p(x|\bar{x},\bar{a})\boldsymbol{\varphi}(\bar{x},\bar{a})^\top\boldsymbol{\Lambda}^{c-1}\boldsymbol{\beta}_t\Big)$$

$$= \pi_t(a|x)\Big((1-\gamma)\nu_0(x) + \gamma\boldsymbol{\psi}(x)^\top\boldsymbol{\Lambda}\boldsymbol{\Lambda}^{c-1}\boldsymbol{\beta}_t\Big) = \mu_{\boldsymbol{\beta}_t,\pi_t}(x,a),$$

where we used the fact that $p(x|\bar{x},\bar{a}) = \langle\boldsymbol{\psi}(x),\boldsymbol{\varphi}(\bar{x},\bar{a})\rangle$, and the definition of $\boldsymbol{\Lambda}$. This in turn facilitates proving that the gradient estimate $\tilde{\boldsymbol{g}}_{\boldsymbol{\theta},t,k}$, defined in Equation 10, is indeed unbiased:

$$\mathbb{E}_{t,k}\left[\tilde{\boldsymbol{g}}_{\boldsymbol{\theta},t,k}\right] = \boldsymbol{\Phi}^\top\mathbb{E}_{t,k}\left[\boldsymbol{\mu}_{t,k}\right] - \boldsymbol{\Lambda}^{c-1}\mathbb{E}_{t,k}\left[\boldsymbol{\varphi}_{t,k}\boldsymbol{\varphi}_{t,k}^\top\right]\boldsymbol{\beta}_t = \boldsymbol{\Phi}^\top\boldsymbol{\mu}_{\boldsymbol{\beta}_t,\pi_t} - \boldsymbol{\Lambda}^c\boldsymbol{\beta}_t = \nabla_{\boldsymbol{\theta}}\mathfrak{L}(\boldsymbol{\beta}_t,\boldsymbol{\mu}_t;\boldsymbol{v}_t,\cdot).$$

A similar proof is used for $\tilde{\boldsymbol{g}}_{\boldsymbol{\beta},t}$ and is detailed in Appendix B.3.

Our analysis is based on arguments by Neu & Okolo [26], carefully adapted to the reparametrized version of the Lagrangian presented above. The proof studies the following central quantity that we refer to as *dynamic duality gap*:

$$\mathcal{G}_T(\boldsymbol{\beta}^*,\pi^*;\boldsymbol{\theta}_{1:T}^*) \doteq \frac{1}{T}\sum_{t=1}^{T}(f(\boldsymbol{\beta}^*,\pi^*;\boldsymbol{\theta}_t) - f(\boldsymbol{\beta}_t,\pi_t;\boldsymbol{\theta}_t^*)). \tag{16}$$

Here, $(\boldsymbol{\theta}_t,\boldsymbol{\beta}_t,\pi_t)$ are the iterates of the algorithm, $\boldsymbol{\theta}_{1:T}^* = (\boldsymbol{\theta}_t^*)_{t=1}^T$ a sequence of comparators for $\boldsymbol{\theta}$, and finally $\boldsymbol{\beta}^*$ and $\pi^*$ are fixed comparators for $\boldsymbol{\beta}$ and $\pi$, respectively. Our first key lemma relates the suboptimality of the output policy to $\mathcal{G}_T$ for a specific choice of comparators.

**Lemma 4.1.** *Let* $\boldsymbol{\theta}_t^* \doteq \boldsymbol{\theta}^{\pi_t}$, $\pi^*$ *be any policy, and* $\boldsymbol{\beta}^* = \boldsymbol{\Lambda}^{-c}\boldsymbol{\Phi}^\top\boldsymbol{\mu}^{\pi^*}$. *Then,* $\mathbb{E}\left[\langle\boldsymbol{\mu}^{\pi^*} - \boldsymbol{\mu}^{\boldsymbol{\pi}_{out}},\boldsymbol{r}\rangle\right] = \mathcal{G}_T(\boldsymbol{\beta}^*,\pi^*;\boldsymbol{\theta}_{1:T}^*)$.

The proof is relegated to Appendix B.1. Our second key lemma rewrites the gap $\mathcal{G}_T$ for *any* choice of comparators as the sum of three regret terms:

**Lemma 4.2.** *With the choice of comparators of Lemma 4.1*

$$\mathcal{G}_T(\boldsymbol{\beta}^*,\pi^*;\boldsymbol{\theta}_{1:T}^*) = \frac{1}{T}\sum_{t=1}^{T}\langle\boldsymbol{\theta}_t - \boldsymbol{\theta}_t^*, g_{\boldsymbol{\theta},t}\rangle + \frac{1}{T}\sum_{t=1}^{T}\langle\boldsymbol{\beta}^* - \boldsymbol{\beta}_t, g_{\boldsymbol{\beta},t}\rangle$$

$$+ \frac{1}{T}\sum_{t=1}^{T}\sum_{s}\nu^{\pi^*}(s)\sum_{a}(\pi^*(a|s) - \pi_t(a|s))\langle\boldsymbol{\theta}_t,\boldsymbol{\varphi}(x,a)\rangle,$$

*where* $g_{\boldsymbol{\theta},t} = \boldsymbol{\Phi}^\top\boldsymbol{\mu}_{\boldsymbol{\beta}_t,\pi_t} - \boldsymbol{\Lambda}^c\boldsymbol{\beta}_t$ *and* $g_{\boldsymbol{\beta},t} = \boldsymbol{\Lambda}^c(\boldsymbol{\omega} + \gamma\boldsymbol{\Psi}v_{\boldsymbol{\theta}_t,\pi_t} - \boldsymbol{\theta}_t)$.

The proof is presented in Appendix B.2. To conclude the proof we bound the three terms appearing in Lemma 4.2. The first two of those are bounded using standard gradient descent/ascent analysis (Lemmas B.1 and B.2), while for the latter we use mirror descent analysis (Lemma B.3). The details of these steps are reported in Appendix B.3.

## 5 Extension to Average-Reward MDPs

In this section, we briefly explain how to extend our approach to offline learning in *average reward MDPs*, establishing the first sample complexity result for this setting. After introducing the setup, we outline a remarkably simple adaptation of our algorithm along with its performance guarantees for this setting. The reader is referred to Appendix C for the full details, and to Chapter 8 of Puterman [29] for a more thorough discussion of average-reward MDPs.

In the average reward setting we aim to optimize the objective $\rho^\pi(x) = \liminf_{T\to\infty}\frac{1}{T}\mathbb{E}_\pi\left[\sum_{t=1}^{T}r(x_t,a_t) \mid x_1 = x\right]$, representing the long-term average reward of policy $\pi$ when started from state $x \in \mathcal{X}$. Unlike the discounted setting, the average reward criterion prioritizes long-term frequency over proximity of good rewards due to the absence of discounting which expresses a preference for earlier rewards. As is standard in the related literature, we will assume that $\rho^\pi$ is well-defined for any policy and is independent of the start state, and thus will

use the same notation to represent the scalar average reward of policy $\pi$. Due to the boundedness of the rewards, we clearly have $\rho^\pi \in [0, 1]$. Similarly to the discounted setting, it is possible to define quantities analogous to the value and action value functions as the solutions to the Bellman equations $\boldsymbol{q}^\pi = \boldsymbol{r} - \rho^\pi \mathbf{1} + \boldsymbol{P}\boldsymbol{v}^\pi$, where $\boldsymbol{v}^\pi$ is related to the action-value function as $v^\pi(x) = \sum_a \pi(a|x) q^\pi(x, a)$. We will make the following standard assumption about the MDP (see, e.g., Section 17.4 of Meyn & Tweedie [22]):

**Assumption 5.1.** For all stationary policies $\pi$, the Bellman equations have a solution $\boldsymbol{q}^\pi$ satisfying $\sup_{x,a} q^\pi(x, a) - \inf_{x,a} q^\pi(x, a) < D_q$.

Furthermore, we will continue to work with the linear MDP assumption of Definition 2.1, and will additionally make the following minor assumption:

**Assumption 5.2.** The all ones vector $\mathbf{1}$ is contained in the column span of the feature matrix $\boldsymbol{\Phi}$. Furthermore, let $\boldsymbol{\varrho} \in \mathbb{R}^d$ such that for all $(x, a) \in \mathcal{Z}$, $\langle \boldsymbol{\varphi}(x, a), \boldsymbol{\varrho} \rangle = 1$.

Using these insights, it is straightforward to derive a linear program akin to (2) that characterize the optimal occupancy measure and thus an optimal policy in average-reward MDPs. Starting from this formulation and proceeding as in Sections 2 and 4, we equivalently restate this optimization problem as finding the saddle-point of the reparametrized Lagrangian defined as follows:

$$\mathfrak{L}(\boldsymbol{\beta}, \boldsymbol{\mu}; \rho, \boldsymbol{v}, \boldsymbol{\theta}) = \rho + \langle \boldsymbol{\beta}, \boldsymbol{\Lambda}^c[\boldsymbol{\omega} + \boldsymbol{\Psi}\boldsymbol{v} - \boldsymbol{\theta} - \rho\boldsymbol{\varrho}] \rangle + \langle \boldsymbol{\mu}, \boldsymbol{\Phi}\boldsymbol{\theta} - \boldsymbol{E}\boldsymbol{v} \rangle.$$

As previously, the saddle point can be shown to be equivalent to an optimal occupancy measure under the assumption that the MDP is linear in the sense of Definition 2.1. Notice that the above Lagrangian slightly differs from that of the discounted setting in Equation (11) due to the additional optimization parameter $\rho$, but otherwise our main algorithm can be directly generalized to this objective. We present details of the derivations and the resulting algorithm in Appendix C. The following theorem states the performance guarantees for this method.

**Theorem 5.3.** *Given a linear MDP (Definition 2.1) satisfying Assumption 5.2 and such that $\boldsymbol{\theta}^\pi \in \mathbb{B}(D_{\boldsymbol{\theta}})$ for any policy $\pi$. Assume that the coverage ratio is bounded $C_{\varphi,c}(\pi^*; \pi_B) \leq D_{\boldsymbol{\beta}}$. Then, for any comparator policy $\pi^*$, the policy output by an appropriately tuned instance of Algorithm 2 satisfies $\mathbb{E}\left[ \langle \boldsymbol{\mu}^{\pi^*} - \boldsymbol{\mu}^{\pi_{out}}, \boldsymbol{r} \rangle \right] \leq \varepsilon$ with a number of samples $n_\epsilon$ that is $O\left( \varepsilon^{-4} D_{\boldsymbol{\theta}}^4 D_{\boldsymbol{\varphi}}^{12c-2} D_{\boldsymbol{\beta}}^4 d^{2-2c} \log |\mathcal{A}| \right)$.*

As compared to the discounted case, this additional dependence of the sample complexity on $D_{\boldsymbol{\varphi}}$ is due to the extra optimization variable $\rho$. We provide the full proof of this theorem along with further discussion in Appendix C.

# 6 Discussion and Final Remarks

In this section, we compare our results with the most relevant ones from the literature. Our Table 1 can be used as a reference. As a complement to this section, we refer the interested reader to the recent work by Uehara & Sun [32], which provides a survey of offline RL methods with their coverage and structural assumptions. Detailed computations can be found in Appendix E.

An important property of our method is that it only requires partial coverage. This sets it apart from classic batch RL methods like FQI [11, 23], which require a stronger uniform-coverage assumption. Algorithms working under partial coverage are mostly based on the principle of pessimism. However, our algorithm does not implement any form of explicit pessimism. We recall that, as shown by Xiao et al. [35], pessimism is just one of many ways to achieve minimax-optimal sample efficiency.

Let us now compare our notion of coverage ratio to the existing notions previsouly used in the literature. Jin et al. [14] (Theorem 4.4) rely on a *feature* coverage ratio which can be written as

$$C^\diamond(\pi^*; \pi_B) = \mathbb{E}_{X,A \sim \mu^*} \left[ \boldsymbol{\varphi}(X, A)^\intercal \boldsymbol{\Lambda}^{-1} \boldsymbol{\varphi}(X, A) \right]. \tag{17}$$

By Jensen's inequality, our $C_{\varphi,1/2}$ (Definition 3.1) is never larger than $C^\diamond$. Indeed, notice how the random features in Equation (17) are coupled, introducing an extra variance term w.r.t. $C_{\varphi,1/2}$. Specifically, we can show that $C_{\varphi,1/2}(\pi^*; \pi_B) = C^\diamond(\pi^*; \pi_B) - \mathbb{V}_{X,A \sim \mu^*}\left[ \boldsymbol{\Lambda}^{-1/2} \boldsymbol{\varphi}(X, A) \right]$, where $\mathbb{V}[Z] = \mathbb{E}[\|Z - \mathbb{E}[Z]\|^2]$ for a random vector $Z$. So, besides fine comparisons with existing notions of coverage ratios, we can regard $C_{\varphi,1/2}$ as a low-variance version of the standard feature coverage ratio. However, our sample complexity bounds do not fully take advantage of this low-variance

property, since they scale quadratically with the ratio itself, rather than linearly, as is more common in previous work.

To scale with $C_{\varphi,1/2}$, our algorithm requires knowledge of $\boldsymbol{\Lambda}$, hence of the behavior policy. However, so does the algorithm from Jin et al. [14]. Zanette et al. [38] remove this requirement at the price of a computationally heavier algorithm. However, both are limited to the finite-horizon setting.

Uehara & Sun [32] and Zhang et al. [39] use a coverage ratio that is conceptually similar to Equation (17),

$$C^{\dagger}(\pi^*; \pi_B) = \sup_{y \in \mathbb{R}^d} \frac{y^{\mathsf{T}} \mathbb{E}_{X, A \sim \mu^*} \left[\boldsymbol{\varphi}(X, A) \boldsymbol{\varphi}(X, A)^{\mathsf{T}}\right] y}{y^{\mathsf{T}} \mathbb{E}_{X, A \sim \mu_B} \left[\boldsymbol{\varphi}(X, A) \boldsymbol{\varphi}(X, A)^{\mathsf{T}}\right] y}. \tag{18}$$

Some linear algebra shows that $C^{\dagger} \le C^{\diamond} \le dC^{\dagger}$. Therefore, chaining the previous inequalities we know that $C_{\varphi,1/2} \le C^{\diamond} \le dC^{\dagger}$. It should be noted that the algorithm from Uehara & Sun [32] also works in the representation-learning setting, that is, with unknown features. However, it is far from being efficiently implementable. The algorithm from Zhang et al. [39] instead is limited to the finite-horizon setting.

In the special case of tabular MDPs, it is hard to compare our ratio with existing ones, because in this setting, error bounds are commonly stated in terms of $\sup_{x,a} \mu^*(x,a)/\mu_B(x,a)$, often introducing an explicit dependency on the number of states [e.g., 17], which is something we carefully avoided. However, looking at how the coverage ratio specializes to the tabular setting can still provide some insight. With known behavior policy, $C_{\varphi,1/2}(\pi^*; \pi_B) = \sum_{x,a} \mu^*(x,a)^2/\mu_B(x,a)$ is smaller than the more standard $C^{\diamond}(\pi^*; \pi_B) = \sum_{x,a} \mu^*(x,a)/\mu_B(x,a)$. With unknown behavior, $C_{\varphi,1}(\pi^*; \pi_B) = \sum_{x,a} (\mu^*(x,a)/\mu_B(x,a))^2$ is non-comparable with $C^{\diamond}$ in general, but larger than $C_{\varphi,1/2}$. Interestingly, $C_{\varphi,1}(\pi^*; \pi_B)$ is also equal to $1 + \mathcal{X}^2(\mu^* \| \mu_B)$, where $\mathcal{X}^2$ denotes the chi-square divergence, a crucial quantity in off-distribution learning based on importance sampling [10]. Moreover, a similar quantity to $C_{\varphi,1}$ was used by Lykouris et al. [18] in the context of (online) RL with adversarial corruptions.

We now turn to the works of Xie et al. [36] and Cheng et al. [9], which are the only practical methods to consider function approximation in the infinite horizon setting, with minimal assumption on the dataset, and thus the only directly comparable to our work. They both use the coverage ratio $C_{\mathcal{F}}(\pi^*; \pi_B) = \max_{f \in \mathcal{F}} \|f - \mathcal{T}f\|^2_{\mu^*}/\|f - \mathcal{T}f\|^2_{\mu_B}$, where $\mathcal{F}$ is a function class and $\mathcal{T}$ is Bellman's operator. This can be shown to reduce to Equation (18) for linear MDPs. However, the specialized bound of Xie et al. [36] (Theorem 3.2) scales with the potentially larger ratio from Equation (17). Both their algorithms have superlinear computational complexity and a sample complexity of $O(\varepsilon^{-5})$. Hence, in the linear MDP setting, our algorithm is a strict improvement both for its $O(\varepsilon^{-4})$ sample complexity and its $O(n)$ computational complexity. However, It is very important to notice that no practical algorithm for this setting so far, including ours, can match the minimax optimal sample complexity rate of $O(\varepsilon^2)$ [35, 31]. This leaves space for future work in this area. In particular, by inspecting our proofs, it should be clear the the extra $O(\varepsilon^{-2})$ factor is due to the nested-loop structure of the algorithm. Therefore, we find it likely that our result can be improved using optimistic descent methods [6] or a two-timescale approach [15, 30].

As a final remark, we remind that when $\boldsymbol{\Lambda}$ is unknown, our error bounds scales with $C_{\varphi,1}$, instead of the smaller $C_{\varphi,1/2}$. However, we find it plausible that one can replace the $\boldsymbol{\Lambda}$ with an estimate that is built using some fraction of the overall sample budget. In particular, in the tabular case, we could simply use all data to estimate the visitation probabilities of each-state action pairs and use them to build an estimator of $\boldsymbol{\Lambda}$. Details of a similar approach have been worked out by Gabbianelli et al. [12]. Nonetheless, we designed our algorithm to be flexible and work in both cases.

To summarize, our method is one of the few not to assume the state space to be finite, or the dataset to have global coverage, while also being computationally feasible. Moreover, it offers a significant advantage, both in terms of sample and computational complexity, over the two existing polynomial-time algorithms for discounted linear MDPs with partial coverage [36, 9]; it extends to the challenging average-reward setting with minor modifications; and has error bounds that scale with a low-variance version of the typical coverage ratio. These results were made possible by employing algorithmic principles, based on the linear programming formulation of sequential decision making, that are new in offline RL. Finally, the main direction for future work is to develop a single-loop algorithm to achieve the optimal rate of $\varepsilon^{-2}$, which should also improve the dependence on the coverage ratio from $C_{\varphi,c}(\pi^*; \pi_B)^2$ to $C_{\varphi,c}(\pi^*; \pi_B)$.

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
