# Supplementary Material

## A  Complete statement of Theorem 3.2

**Theorem A.1.** *Consider a linear MDP (Definition 2.1) such that $\boldsymbol{\theta}^\pi \in \mathbb{B}(D_{\boldsymbol{\theta}})$ for all $\pi \in \Pi$. Further, suppose that $C_{\varphi,c}(\pi^*; \pi_B) \leq D_{\boldsymbol{\beta}}$. Then, for any comparator policy $\pi^* \in \Pi$, the policy output by Algorithm 1 satisfies:*

$$\mathbb{E}\left[\langle \boldsymbol{\mu}^{\pi^*} - \boldsymbol{\mu}^{\boldsymbol{\pi}_{out}}, \boldsymbol{r} \rangle\right] \leq \frac{2D_{\boldsymbol{\beta}}^2}{\zeta T} + \frac{\log|\mathcal{A}|}{\alpha T} + \frac{2D_{\boldsymbol{\theta}}^2}{\eta K} + \frac{\zeta G_{\boldsymbol{\beta},c}^2}{2} + \frac{\alpha D_{\boldsymbol{\theta}}^2 D_{\boldsymbol{\varphi}}^2}{2} + \frac{\eta G_{\boldsymbol{\theta},c}^2}{2},$$

*where:*

$$G_{\boldsymbol{\theta},c}^2 = 3D_{\boldsymbol{\varphi}}^2 \left((1-\gamma)^2 + (1+\gamma^2)D_{\boldsymbol{\beta}}^2 \|\boldsymbol{\Lambda}\|_2^{2c-1}\right), \tag{19}$$

$$G_{\boldsymbol{\beta},c}^2 = 3(1 + (1+\gamma^2)D_{\boldsymbol{\varphi}}^2 D_{\boldsymbol{\theta}}^2)D_{\boldsymbol{\varphi}}^{2(2c-1)}. \tag{20}$$

*In particular, using learning rates $\eta = \frac{2D_{\boldsymbol{\theta}}}{G_{\boldsymbol{\theta},c}\sqrt{K}}$, $\zeta = \frac{2D_{\boldsymbol{\beta}}}{G_{\boldsymbol{\beta},c}\sqrt{T}}$, and $\alpha = \frac{\sqrt{2\log|\mathcal{A}|}}{D_{\boldsymbol{\varphi}}D_{\boldsymbol{\theta}}\sqrt{T}}$, and setting $K = T \cdot \frac{2D_{\boldsymbol{\beta}^2}G_{\boldsymbol{\beta},c}^2 + D_{\boldsymbol{\theta}}^2 D_{\boldsymbol{\varphi}}^2 \log|\mathcal{A}|}{2D_{\boldsymbol{\theta}}^2 G_{\boldsymbol{\theta},c}^2}$, we achieve $\mathbb{E}\left[\langle \boldsymbol{\mu}^{\pi^*} - \boldsymbol{\mu}^{\boldsymbol{\pi}_{out}}, \boldsymbol{r} \rangle\right] \leq \epsilon$ with a number of samples $n_\epsilon$ that is*

$$O\left(\epsilon^{-4} D_{\boldsymbol{\theta}}^4 D_{\boldsymbol{\varphi}}^4 D_{\boldsymbol{\beta}}^4 \operatorname{Tr}(\boldsymbol{\Lambda}^{2c-1}) \|\boldsymbol{\Lambda}\|_2^{2c-1} \log|\mathcal{A}|\right).$$

By remark A.2 below, we have that $n_\epsilon$ is simply of order $O\left(\varepsilon^{-4} D_{\boldsymbol{\theta}}^4 D_{\boldsymbol{\varphi}}^{8c} D_{\boldsymbol{\beta}}^4 d^{2-2c} \log|\mathcal{A}|\right)$

*Remark* A.2. When $c = 1/2$, the factor $\operatorname{Tr}(\boldsymbol{\Lambda}^{2c-1})$ is just $d$, the feature dimension, and $\|\boldsymbol{\Lambda}\|_2^{2c-1} = 1$. When $c = 1$ and $\boldsymbol{\Lambda}$ is unknown, both $\|\boldsymbol{\Lambda}\|_2$ and $\operatorname{Tr}(\boldsymbol{\Lambda})$ should be replaced by their upper bound $D_{\boldsymbol{\varphi}}^2$. Then, for $c \in \{1/2, 1\}$, we have that $\operatorname{Tr}(\boldsymbol{\Lambda}^{2c-1}) \|\boldsymbol{\Lambda}\|_2^{2c-1} \leq D_{\boldsymbol{\varphi}}^{8c-4} d^{2-2c}$.

# B  Missing Proofs for the Discounted Setting

## B.1  Proof of Lemma 4.1

Using the choice of comparators described in the lemma, we have

$$\nu_{\boldsymbol{\beta}^*}(s) = (1-\gamma)\nu_0(s) + \gamma\langle\boldsymbol{\psi}(s), \boldsymbol{\Lambda}^c\boldsymbol{\Lambda}^{-c}\boldsymbol{\Phi}^\top\boldsymbol{\mu}^{\pi^*}\rangle$$
$$= (1-\gamma)\nu_0(s) + \sum_{s',a'} P(s|s',a')\mu^{\pi^*}(s',a') = \nu^{\pi^*}(s),$$

hence $\mu_{\boldsymbol{\beta}^*,\pi^*} = \boldsymbol{\mu}^{\pi^*}$. From Equation (14) it is easy to see that

$$f(\boldsymbol{\beta}^*,\pi^*;\boldsymbol{\theta}_t) = \langle\boldsymbol{\Lambda}^{-c}\boldsymbol{\Phi}^\top\boldsymbol{\mu}^*, \boldsymbol{\Lambda}^c\boldsymbol{\omega}\rangle + \langle\boldsymbol{\theta}_t, \boldsymbol{\Phi}^\top\boldsymbol{\mu}^* - \boldsymbol{\Lambda}^c\boldsymbol{\Lambda}^{-c}\boldsymbol{\Phi}^\top\boldsymbol{\mu}^*\rangle$$
$$= \langle\mu^{\pi^*}, \boldsymbol{\Phi}\boldsymbol{\omega}\rangle = \langle\boldsymbol{\mu}^*, \boldsymbol{r}\rangle.$$

Moreover, we also have

$$v_{\boldsymbol{\theta}_t^*,\pi_t}(s) = \sum_a \pi_t(a|s)\langle\boldsymbol{\theta}^{\pi_t}, \boldsymbol{\varphi}(x,a)\rangle$$
$$= \sum_a \pi_t(a|s)q^{\pi_t}(s,a) = v^{\pi_t}(s,a).$$

Then, from Equation (15) we obtain

$$f(\boldsymbol{\theta}_t^*,\boldsymbol{\beta}_t,\pi_t)$$
$$= (1-\gamma)\langle\boldsymbol{\nu}_0, v^{\pi_t}\rangle + \langle\boldsymbol{\beta}_t, \boldsymbol{\Lambda}^c(\boldsymbol{\omega} + \gamma\boldsymbol{\Psi}\boldsymbol{v}^{\pi_t} - \boldsymbol{\theta}^{\pi_t})\rangle$$
$$= (1-\gamma)\langle\boldsymbol{\nu}_0, v^{\pi_t}\rangle + \langle\boldsymbol{\beta}_t, \boldsymbol{\Lambda}^{c-1}\mathbb{E}_{X,A\sim\mu_B}\left[\boldsymbol{\varphi}(X,A)\boldsymbol{\varphi}(X,A)^\top(\boldsymbol{\omega} + \gamma\boldsymbol{\Psi}\boldsymbol{v}^{\pi_t} - \boldsymbol{\theta}^{\pi_t})]\rangle$$
$$= (1-\gamma)\langle\boldsymbol{\nu}_0, v^{\pi_t}\rangle + \langle\boldsymbol{\beta}_t, \boldsymbol{\Lambda}^{c-1}\mathbb{E}_{X,A\sim\mu_B}\left[[r(X,A) + \gamma\langle p(\cdot|X,A), \boldsymbol{v}^{\pi_t}\rangle - \boldsymbol{q}^{\pi_t}(X,A)]\boldsymbol{\varphi}(X,A)]\rangle$$
$$= (1-\gamma)\langle\boldsymbol{\nu}_0, v^{\pi_t}\rangle = \langle\mu^{\pi_t}, \boldsymbol{r}\rangle,$$

where the fourth equality uses that the value functions satisfy the Bellman equation $\boldsymbol{q}^\pi = \boldsymbol{r} + \gamma\boldsymbol{P}\boldsymbol{v}^\pi$ for any policy $\pi$. The proof is concluded by noticing that, since $\boldsymbol{\pi}_{\text{out}}$ is sampled uniformly from $\{\pi_t\}_{t=1}^T$, $\mathbb{E}\left[\langle\boldsymbol{\mu}^{\boldsymbol{\pi}_{\text{out}}}, \boldsymbol{r}\rangle\right] = \frac{1}{T}\sum_{t=1}^T\mathbb{E}\left[\langle\boldsymbol{\mu}^{\pi_t}, \boldsymbol{r}\rangle\right]$. $\qquad\square$

## B.2  Proof of Lemma 4.2

We start by rewriting the terms appearing in the definition of $\mathcal{G}_T$:

$$f(\boldsymbol{\beta}^*,\pi^*;\boldsymbol{\theta}_t) - f(\boldsymbol{\beta}_t,\pi_t;\boldsymbol{\theta}_t^*) = f(\boldsymbol{\beta}^*,\pi^*;\boldsymbol{\theta}_t) - f(\boldsymbol{\beta}^*,\pi_t;\boldsymbol{\theta}_t)$$
$$+ f(\boldsymbol{\beta}^*,\pi_t;\boldsymbol{\theta}_t) - f(\boldsymbol{\beta}_t,\pi_t;\boldsymbol{\theta}_t)$$
$$+ f(\boldsymbol{\beta}_t,\pi_t;\boldsymbol{\theta}_t) - f(\boldsymbol{\beta}_t,\pi_t;\boldsymbol{\theta}_t^*). \tag{21}$$

To rewrite this as the sum of the three regret terms, we first note that

$$f(\boldsymbol{\beta},\pi;\boldsymbol{\theta}) = \langle\boldsymbol{\Lambda}^c\boldsymbol{\beta}, \boldsymbol{\omega} - \boldsymbol{\theta}_t\rangle + \langle\nu_{\boldsymbol{\beta}}, v_{\boldsymbol{\theta}_t,\pi}\rangle,$$

which allows us to write the first term of Equation (21) as

$$f(\boldsymbol{\beta}^*,\pi^*;\boldsymbol{\theta}_t) - f(\boldsymbol{\beta}^*,\pi_t;\boldsymbol{\theta}_t) = \langle\boldsymbol{\Lambda}^c(\boldsymbol{\beta}^* - \boldsymbol{\beta}^*), \boldsymbol{\omega} - \boldsymbol{\theta}_t\rangle + \langle\nu_{\boldsymbol{\beta}^*}, v_{\boldsymbol{\theta}_t,\pi^*} - v_{\boldsymbol{\theta}_t,\pi_t}\rangle$$
$$= \langle\nu_{\boldsymbol{\beta}^*}, \sum_a(\pi^*(a|\cdot) - \pi_t(a|\cdot))\langle\boldsymbol{\theta}_t, \boldsymbol{\varphi}(\cdot,a)\rangle\rangle,$$

and we have already established in the proof of Lemma C.3 that $\nu_{\boldsymbol{\beta}^*}$ is equal to $\nu^{\pi^*}$ for our choice of comparator. Similarly, we use Equation (15) to rewrite the second term of Equation (21) as

$$f(\boldsymbol{\beta}^*,\pi_t;\boldsymbol{\theta}_t) - f(\boldsymbol{\beta}_t,\pi_t;\boldsymbol{\theta}_t) = (1-\gamma)\langle\boldsymbol{\nu}_0, v_{\boldsymbol{\theta}_t,\pi_t} - v_{\boldsymbol{\theta}_t,\pi_t}\rangle + \langle\boldsymbol{\beta}^* - \boldsymbol{\beta}_t, \boldsymbol{\Lambda}^c(\boldsymbol{\omega} + \gamma\boldsymbol{\Psi}v_{\boldsymbol{\theta}_t,\pi_t} - \boldsymbol{\theta}_t)\rangle$$
$$= \langle\boldsymbol{\beta}^* - \boldsymbol{\beta}_t, g_{\boldsymbol{\beta},t}\rangle.$$

Finally, we use Equation (14) to rewrite the third term of Equation (21) as

$$f(\boldsymbol{\beta}_t,\pi_t;\boldsymbol{\theta}_t) - f(\boldsymbol{\beta}_t,\pi_t;\boldsymbol{\theta}_t^*) = \langle\boldsymbol{\beta}_t - \boldsymbol{\beta}_t, \boldsymbol{\Lambda}^c\boldsymbol{\omega}\rangle + \langle\boldsymbol{\theta}_t - \boldsymbol{\theta}_t^*, \boldsymbol{\Phi}^\top\mu_{\boldsymbol{\beta}_t,\pi_t} - \boldsymbol{\Lambda}^c\boldsymbol{\beta}_t\rangle$$
$$= \langle\boldsymbol{\theta}_t - \boldsymbol{\theta}_t^*, g_{\boldsymbol{\theta},t}\rangle.$$

## B.3  Regret bounds for stochastic gradient descent / ascent

**Lemma B.1.** *For any dynamic comparator $\boldsymbol{\theta}_{1:T} \in D_{\boldsymbol{\theta}}$, the iterates $\boldsymbol{\theta}_1, \ldots, \boldsymbol{\theta}_T$ of Algorithm 1 satisfy the following regret bound:*

$$
\mathbb{E}\left[\sum_{t=1}^{T}\langle \boldsymbol{\theta}_t - \boldsymbol{\theta}_t^*, g_{\boldsymbol{\theta},t}\rangle\right] \leq \frac{2TD_{\boldsymbol{\theta}}^2}{\eta K} + \frac{3\eta T D_{\boldsymbol{\varphi}}^2\left((1-\gamma)^2 + (1+\gamma^2)D_{\beta}^2\,\|\boldsymbol{\Lambda}\|_2^{2c-1}\right)}{2}.
$$

*Proof.* First, we use the definition of $\boldsymbol{\theta}_t$ as the average of the inner-loop iterates from Algorithm 1, together with linearity of expectation and bilinearity of the inner product.

$$
\mathbb{E}\left[\sum_{t=1}^{T}\langle \boldsymbol{\theta}_t - \boldsymbol{\theta}_t^*, g_{\boldsymbol{\theta},t}\rangle\right] = \sum_{t=1}^{T}\frac{1}{K}\mathbb{E}\underbrace{\left[\sum_{k=1}^{K}\langle \boldsymbol{\theta}_{t,k} - \boldsymbol{\theta}_t^*, g_{\boldsymbol{\theta},t}\rangle\right]}_{\mathfrak{R}_t}. \tag{22}
$$

We then appeal to standard stochastic gradient descent analysis to bound each term $\mathfrak{R}_t$ separately.

We have already proven in Section 4 that the gradient estimator for $\boldsymbol{\theta}$ is unbiased, that is, $\mathbb{E}_{t,k}\left[\tilde{\boldsymbol{g}}_{\boldsymbol{\theta},t,k}\right] = \boldsymbol{g}_{\theta,t}$. It is also useful to recall here that $\tilde{\boldsymbol{g}}_{\boldsymbol{\theta},t,k}$ does *not* depend on $\boldsymbol{\theta}_{t,k}$. Next, we show that its second moment is bounded. From Equation (10), plugging in the definition of $\mu_{t,k}$ from Equation (8) and using the abbreviations $\boldsymbol{\varphi}_{t,k}^0 = \sum_a \pi_t(a|x_{t,k}^0)\boldsymbol{\varphi}(x_{t,k}^0, a)$, $\boldsymbol{\varphi}_t = \boldsymbol{\varphi}(x_{t,k}, a_{t,k})$, and $\boldsymbol{\varphi}'_{t,k} = \sum_a \pi_t(a|x_{t,k}^0)\boldsymbol{\varphi}(x'_{t,k}, a)$, we have:

$$
\mathbb{E}_{t,k}\left[\|\tilde{\boldsymbol{g}}_{\boldsymbol{\theta},t,i}\|^2\right]
$$
$$
= \mathbb{E}_{t,k}\left[\left\|(1-\gamma)\boldsymbol{\varphi}_{t,k}^0 + \gamma\boldsymbol{\varphi}'_{t,k}\langle\boldsymbol{\varphi}_{tk}, \boldsymbol{\Lambda}^{c-1}\boldsymbol{\beta}_t\rangle - \boldsymbol{\varphi}_{t,k}\langle\boldsymbol{\varphi}_{tk}, \boldsymbol{\Lambda}^{c-1}\boldsymbol{\beta}_t\rangle\right\|^2\right]
$$
$$
\leq 3(1-\gamma)^2\mathcal{D}_{\boldsymbol{\varphi}}^2 + 3\gamma^2\mathbb{E}_{t,k}\left[\left\|\boldsymbol{\varphi}'_{t,k}\langle\boldsymbol{\varphi}_{tk}, \boldsymbol{\Lambda}^{c-1}\boldsymbol{\beta}_t\rangle\right\|^2\right] + 3\mathbb{E}_{t,k}\left[\left\|\boldsymbol{\varphi}_{t,k}\langle\boldsymbol{\varphi}_{tk}, \boldsymbol{\Lambda}^{c-1}\boldsymbol{\beta}_t\rangle\right\|^2\right]
$$
$$
\leq 3(1-\gamma)^2\mathcal{D}_{\boldsymbol{\varphi}}^2 + 3(1+\gamma^2)D_{\boldsymbol{\varphi}}^2\mathbb{E}_{t,k}\left[\langle\boldsymbol{\varphi}_{tk}, \boldsymbol{\Lambda}^{c-1}\boldsymbol{\beta}_t\rangle^2\right]
$$
$$
= 3(1-\gamma)^2\mathcal{D}_{\boldsymbol{\varphi}}^2 + 3(1+\gamma^2)D_{\boldsymbol{\varphi}}^2\boldsymbol{\beta}_t^\top\boldsymbol{\Lambda}^{c-1}\mathbb{E}_{t,k}\left[\boldsymbol{\varphi}_{tk}\boldsymbol{\varphi}_{tk}^\top\right]\boldsymbol{\Lambda}^{c-1}\boldsymbol{\beta}_t
$$
$$
= 3(1-\gamma)^2\mathcal{D}_{\boldsymbol{\varphi}}^2 + 3(1+\gamma^2)D_{\boldsymbol{\varphi}}^2\|\boldsymbol{\beta}_t\|_{\boldsymbol{\Lambda}^{2c-1}}^2.
$$

We can then apply Lemma D.1 with the latter expression as $G^2$, $\mathbb{B}(D_{\boldsymbol{\theta}})$ as the domain, and $\eta$ as the learning rate, obtaining:

$$
\mathbb{E}_t\left[\sum_{k=1}^{K}\langle\boldsymbol{\theta}_{t,k} - \boldsymbol{\theta}_t^*, g_{\boldsymbol{\theta},t}\rangle\right] \leq \frac{\|\boldsymbol{\theta}_{t,1} - \boldsymbol{\theta}_t^*\|_2^2}{2\eta} + \frac{3\eta K D_{\boldsymbol{\varphi}}^2\left((1-\gamma)^2 + (1+\gamma^2)\|\boldsymbol{\beta}_t\|_{\boldsymbol{\Lambda}^{2c-1}}^2\right)}{2}
$$
$$
\leq \frac{2D_{\theta}^2}{\eta} + \frac{3\eta K D_{\boldsymbol{\varphi}}^2\left((1-\gamma)^2 + (1+\gamma^2)\|\boldsymbol{\beta}_t\|_{\boldsymbol{\Lambda}^{2c-1}}^2\right)}{2}.
$$

Plugging this into Equation (22) and bounding $\|\boldsymbol{\beta}_t\|_{\boldsymbol{\Lambda}^{2c-1}}^2 \leq D_{\boldsymbol{\beta}}^2\|\boldsymbol{\Lambda}\|_2^{2c-1}$, we obtain the final result. $\qquad\square$

**Lemma B.2.** *For any comparator $\boldsymbol{\beta} \in D_{\boldsymbol{\beta}}$, the iterates $\boldsymbol{\beta}_1, \ldots, \boldsymbol{\beta}_T$ of Algorithm 1 satisfy the following regret bound:*

$$
\mathbb{E}\left[\sum_{t=1}^{T}\langle\boldsymbol{\beta}^* - \boldsymbol{\beta}_t, g_{\boldsymbol{\beta},t}\rangle\right] \leq \frac{2D_{\boldsymbol{\beta}}^2}{\zeta} + \frac{3\zeta T(1 + (1+\gamma^2)D_{\boldsymbol{\varphi}}^2 D_{\boldsymbol{\theta}}^2)\operatorname{Tr}(\boldsymbol{\Lambda}^{2c-1})}{2}.
$$

*Proof.* We again employ stochastic gradient descent analysis. We first prove that the gradient estimator for $\boldsymbol{\beta}$ is unbiased. Recalling the definition of $\tilde{\boldsymbol{g}}_{\beta,t}$ from Equation (9),

$$
\begin{aligned}
\mathbb{E}\left[\tilde{\boldsymbol{g}}_{\beta,t}|\mathcal{F}_{t-1},\boldsymbol{\theta}_t\right] &= \mathbb{E}\left[\boldsymbol{\Lambda}^{c-1}\boldsymbol{\varphi}_t\left(R_t + \gamma v_t(X_t') - \langle\boldsymbol{\varphi}_t,\boldsymbol{\theta}_t\rangle\right)|\mathcal{F}_{t-1},\boldsymbol{\theta}_t\right] \\
&= \boldsymbol{\Lambda}^{c-1}\left(\mathbb{E}_t\left[\boldsymbol{\varphi}_t\boldsymbol{\varphi}_t^\top\right]\boldsymbol{\omega} + \gamma\mathbb{E}_t\left[\boldsymbol{\varphi}_t v_t(X_t')\right] - \mathbb{E}_t\left[\boldsymbol{\varphi}_t\boldsymbol{\varphi}_t^\top\right]\boldsymbol{\theta}_t\right) \\
&= \boldsymbol{\Lambda}^{c-1}\left(\boldsymbol{\Lambda}\boldsymbol{\omega} + \gamma\mathbb{E}_t\left[\boldsymbol{\varphi}_t v_t(X_t')\right] - \boldsymbol{\Lambda}\boldsymbol{\theta}_t\right) \\
&= \boldsymbol{\Lambda}^{c-1}\left(\boldsymbol{\Lambda}\boldsymbol{\omega} + \gamma\mathbb{E}_t\left[\boldsymbol{\varphi}_t\boldsymbol{P}(\cdot|X_t,A_t)\boldsymbol{v}_t\right] - \boldsymbol{\Lambda}\boldsymbol{\theta}_t\right) \\
&= \boldsymbol{\Lambda}^{c-1}\left(\boldsymbol{\Lambda}\boldsymbol{\omega} + \gamma\mathbb{E}_t\left[\boldsymbol{\varphi}_t\boldsymbol{\varphi}_t^\top\right]\boldsymbol{\Psi}\boldsymbol{v}_t - \boldsymbol{\Lambda}\boldsymbol{\theta}_t\right) \\
&= \boldsymbol{\Lambda}^c(\boldsymbol{\omega} + \gamma\boldsymbol{\Psi}v_{\boldsymbol{\theta}_t,\pi_t} - \boldsymbol{\theta}_t) = \boldsymbol{g}_{\beta,t},
\end{aligned}
$$

recalling that $\boldsymbol{v}_t = \boldsymbol{v}_{\boldsymbol{\theta}_t,\pi_t}$. Next, we bound its second moment. We use the fact that $r \in [0,1]$ and $\|\boldsymbol{v}_t\|_\infty \leq \|\boldsymbol{\Phi}\boldsymbol{\theta}_t\|_\infty \leq D_{\boldsymbol{\varphi}}D_{\boldsymbol{\theta}}$ to show that

$$
\begin{aligned}
\mathbb{E}\left[\|\tilde{\boldsymbol{g}}_{\beta,t}\|_2^2\,|\mathcal{F}_{t-1},\boldsymbol{\theta}_t\right] &= \mathbb{E}\left[\left\|\boldsymbol{\Lambda}^{c-1}\boldsymbol{\varphi}_t[R_t + \gamma v_t(X_t') - \langle\boldsymbol{\theta}_t,\boldsymbol{\varphi}_t\rangle]\right\|_2^2\,|\mathcal{F}_{t-1},\boldsymbol{\theta}_t\right] \\
&\leq 3(1 + (1+\gamma^2)D_{\boldsymbol{\varphi}}^2 D_{\boldsymbol{\theta}}^2)\mathbb{E}_t\left[\boldsymbol{\varphi}_t^\top\boldsymbol{\Lambda}^{2(c-1)}\boldsymbol{\varphi}_t\right] \\
&= 3(1 + (1+\gamma^2)D_{\boldsymbol{\varphi}}^2 D_{\boldsymbol{\theta}}^2)\mathbb{E}_t\left[\mathrm{Tr}(\boldsymbol{\Lambda}^{2(c-1)}\boldsymbol{\varphi}_t\boldsymbol{\varphi}_t^\top)\right] \\
&= 3(1 + (1+\gamma^2)D_{\boldsymbol{\varphi}}^2 D_{\boldsymbol{\theta}}^2)\,\mathrm{Tr}(\boldsymbol{\Lambda}^{2c-1}).
\end{aligned}
$$

Thus, we can apply Lemma D.1 with the latter expression as $G^2$, $\mathbb{B}(D_{\boldsymbol{\beta}})$ as the domain, and $\zeta$ as the learning rate. $\qquad\square$

**Lemma B.3.** *The sequence of policies $\pi_1,\ldots,\pi_T$ of Algorithm 1 satisfies the following regret bound:*

$$
\mathbb{E}\left[\sum_{t=1}^T\sum_{x\in\mathcal{X}}\nu^{\pi^*}(x)\sum_a(\pi^*(a|x) - \pi_t(a|x))\langle\boldsymbol{\theta}_t,\boldsymbol{\varphi}(x,a)\rangle\right] \leq \frac{\log|\mathcal{A}|}{\alpha} + \frac{\alpha T D_{\boldsymbol{\varphi}}^2 D_{\boldsymbol{\theta}}^2}{2}.
$$

*Proof.* We just apply mirror descent analysis, invoking Lemma D.2 with $q_t = \Phi\boldsymbol{\theta}_t$, noting that $\|q_t\|_\infty \leq D_{\boldsymbol{\varphi}}D_{\boldsymbol{\theta}}$. The proof is concluded by trivially bounding the relative entropy as $\mathcal{H}(\pi^*\|\pi_1) = \mathbb{E}_{x\sim\nu^{\pi^*}}\left[\mathcal{D}(\pi(\cdot|x)\|\pi_1(\cdot|x))\right] \leq \log|\mathcal{A}|$. $\qquad\square$

## C  Analysis for the Average-Reward MDP Setting

This section describes the adaptation of our contributions in the main body of the paper to average-reward MDPs (AMDPs). In the offline reinforcement learning setting that we consider, we assume access to a sequence of data points $(X_t, A_t, R_t, X'_t)$ in round $t$ generated by a behaviour policy $\pi_B$ whose occupancy measure is denoted as $\boldsymbol{\mu}_B$. Specifically, we will now draw i.i.d. samples from the *undiscounted* occupancy measure as $X_t, A_t \sim \boldsymbol{\mu}_B$, sample $X'_t \sim p(\cdot | X_t, A_t)$, and compute immediate rewards as $R_t = r(X_t, A_t)$. For simplicity, we use the shorthand notation $\boldsymbol{\varphi}_t = \varphi(X_t, A_t)$ to denote the feature vector drawn in round $t$, and define the matrix $\boldsymbol{\Lambda} = \mathbb{E}\left[\varphi(X_t, A_t)\varphi(X_t, A_t)^\top\right]$.

Before describing our contributions, some definitions are in order. An important central concept in the theory of AMDPs is that of the *relative value functions* of policy $\pi$ defined as

$$
v^\pi(x) = \lim_{T \to \infty} \mathbb{E}_\pi \left[ \sum_{t=0}^{T} r(X_t, A_t) - \rho^\pi \middle| X_0 = x \right],
$$

$$
q^\pi(x, a) = \lim_{T \to \infty} \mathbb{E}_\pi \left[ \sum_{t=0}^{T} r(X_t, A_t) - \rho^\pi \middle| X_0 = x, A_0 = a \right],
$$

where we recalled the notation $\rho^\pi$ denoting the average reward of policy $\pi$ from the main text. These functions are sometimes also called the *bias functions*, and their intuitive role is to measure the total amount of reward gathered by policy $\pi$ before it hits its stationary distribution. For simplicity, we will refer to these functions as value functions and action-value functions below.

By their recursive nature, these value functions are also characterized by the corresponding Bellman equations recalled below for completeness

$$
\boldsymbol{q}^\pi = \boldsymbol{r} - \rho^\pi \mathbf{1} + \boldsymbol{P} \boldsymbol{v}^\pi,
$$

where $\boldsymbol{v}^\pi$ is related to the action-value function as $v^\pi(x) = \sum_a \pi(a|x) q^\pi(x, a)$. We note that the Bellman equations only characterize the value functions up to a constant offset. That is, for any policy $\pi$, and constant $c \in \mathbb{R}$, $\boldsymbol{v}^\pi + c\mathbf{1}$ and $\boldsymbol{q}^\pi + c\mathbf{1}$ also satisfy the Bellman equations. A key quantity to measure the size of the value functions is the *span seminorm* defined for $\boldsymbol{q} \in \mathbb{R}^{\mathcal{X} \times \mathcal{A}}$ as $\|\boldsymbol{q}\|_{\mathrm{sp}} = \sup_{(x,a) \in \mathcal{X} \times \mathcal{A}} q(x, a) - \inf_{(x,a) \in \mathcal{X} \times \mathcal{A}} q(x, a)$. Using this notation, the condition of Assumption 5.1 can be simply stated as requiring $\|\boldsymbol{q}^\pi\|_{\mathrm{sp}} \leq D_q$ for all $\pi$.

Now, let $\pi^*$ denote an optimal policy with maximum average reward and introduce the shorthand notations $\rho^* = \rho^{\pi^*}, \boldsymbol{\mu}^* = \boldsymbol{\mu}^{\pi^*}, \boldsymbol{\nu}^* = \boldsymbol{\nu}^{\pi^*}, \boldsymbol{v}^* = \boldsymbol{v}^{\pi^*}$ and $\boldsymbol{q}^* = \boldsymbol{q}^{\pi^*}$. Under mild assumptions on the MDP that we will clarify shortly, the following Bellman optimality equations are known to characterize bias vectors corresponding to the optimal policy

$$
\boldsymbol{q}^* = \boldsymbol{r} - \rho^* \mathbf{1} + \boldsymbol{P} \boldsymbol{v}^*,
$$

where $\boldsymbol{v}^*$ satisfies $v^*(x) = \max_a q^*(x, a)$. Once again, shifting the solutions by a constant preserves the optimality conditions. It is easy to see that such constant offsets do not influence greedy or softmax policies extracted from the action value functions. Importantly, by a calculation analogous to Equation (3), the action-value functions are exactly realizable under the linear MDP condition (see Definition 2.1) and Assumption 5.2.

Besides the Bellman optimality equations stated above, optimal policies can be equivalently characterized via the following linear program:

$$
\begin{aligned}
\text{maximize} \quad & \langle \boldsymbol{\mu}, \boldsymbol{r} \rangle \\
\text{subject to} \quad & \boldsymbol{E}^\top \boldsymbol{\mu} = \boldsymbol{P}^\top \boldsymbol{\mu} \\
& \langle \boldsymbol{\mu}, \mathbf{1} \rangle = 1 \\
& \boldsymbol{\mu} \geq 0.
\end{aligned}
\tag{23}
$$

This can be seen as the generalization of the LP stated for discounted MDPs in the main text, with the added complication that we need to make sure that the occupancy measures are normalized[1] to 1. By following the same steps as in the main text to relax the constraints and reparametrize the LP, one

---

[1]This is necessary because of the absence of $\nu_0$ in the LP, which would otherwise fix the scale of the solutions.

can show that solutions of the LP under the linear MDP assumption can be constructed by finding the saddle point of the following Lagrangian:

$$\mathfrak{L}(\boldsymbol{\lambda}, \boldsymbol{\mu}; \rho, \boldsymbol{v}, \boldsymbol{\theta}) = \rho + \langle \boldsymbol{\lambda}, \boldsymbol{\omega} + \boldsymbol{\Psi v} - \boldsymbol{\theta} - \rho \boldsymbol{\varrho} \rangle + \langle \boldsymbol{u}, \boldsymbol{\Phi \theta} - \boldsymbol{Ev} \rangle$$
$$= \rho[1 - \langle \boldsymbol{\lambda}, \boldsymbol{\varrho} \rangle] + \langle \boldsymbol{\theta}, \boldsymbol{\Phi}^{\mathsf{T}} \boldsymbol{\mu} - \boldsymbol{\lambda} \rangle + \langle \boldsymbol{v}, \boldsymbol{\Psi}^{\mathsf{T}} \boldsymbol{\lambda} - \boldsymbol{E}^{\mathsf{T}} \boldsymbol{\mu} \rangle.$$

As before, the optimal value functions $\boldsymbol{q}^*$ and $\boldsymbol{v}^*$ are optimal primal variables for the saddle-point problem, as are all of their constant shifts. Thus, the existence of a solution with small span seminorm implies the existence of a solution with small supremum norm.

Finally, applying the same reparametrization $\boldsymbol{\beta} = \boldsymbol{\Lambda}^{-c} \boldsymbol{\lambda}$ as in the discounted setting, we arrive to the following Lagrangian that forms the basis of our algorithm:

$$\mathfrak{L}(\boldsymbol{\beta}, \boldsymbol{\mu}; \rho, \boldsymbol{v}, \boldsymbol{\theta}) = \rho + \langle \boldsymbol{\beta}, \boldsymbol{\Lambda}^c[\boldsymbol{\omega} + \boldsymbol{\Psi v} - \boldsymbol{\theta} - \rho \boldsymbol{\varrho}] \rangle + \langle \boldsymbol{\mu}, \boldsymbol{\Phi \theta} - \boldsymbol{Ev} \rangle.$$

We will aim to find the saddle point of this function via primal-dual methods. As we have some prior knowledge of the optimal solutions, we will restrict the search space of each optimization variable to nicely chosen compact sets. For the $\boldsymbol{\beta}$ iterates, we consider the Euclidean ball domain $\mathbb{B}(D_{\boldsymbol{\beta}}) = \{\boldsymbol{\beta} \in \mathbb{R}^d \mid \|\boldsymbol{\beta}\|_2 \leq D_{\boldsymbol{\beta}}\}$ with the bound $D_{\boldsymbol{\beta}} > \|\boldsymbol{\Phi}^{\mathsf{T}} \boldsymbol{\mu}^*\|_{\boldsymbol{\Lambda}^{-2c}}$. Since the average reward of any policy is bounded in $[0, 1]$, we naturally restrict the $\rho$ iterates to this domain. Finally, keeping in mind that Assumption 5.1 guarantees that $\|\boldsymbol{q}^{\pi}\|_{\mathrm{sp}} \leq D_q$, we will also constrain the $\boldsymbol{\theta}$ iterates to an appropriate domain: $\mathbb{B}(D_{\boldsymbol{\theta}}) = \{\boldsymbol{\theta} \in \mathbb{R}^d \mid \|\boldsymbol{\theta}\|_2 \leq D_{\boldsymbol{\theta}}\}$. We will assume that this domain is large enough to represent all action-value functions, which implies that $D_{\boldsymbol{\theta}}$ should scale at least linearly with $D_q$. Indeed, we will suppose that the features are bounded as $\|\boldsymbol{\varphi}(x, a)\|_2 \leq D_{\boldsymbol{\varphi}}$ for all $(x, a) \in \mathcal{X} \times \mathcal{A}$ so that our optimization algorithm only admits parametric $\boldsymbol{q}$ functions satisfying $\|\boldsymbol{q}\|_{\infty} \leq D_{\boldsymbol{\varphi}} D_{\boldsymbol{\theta}}$. Obviously, $D_{\boldsymbol{\theta}}$ needs to be set large enough to ensure that it is possible at all to represent $\boldsymbol{q}$-functions with span $D_q$.

Thus, we aim to solve the following constrained optimization problem:

$$\min_{\rho \in [0,1], \boldsymbol{v} \in \mathbb{R}^{\mathcal{X}}, \boldsymbol{\theta} \in \mathbb{B}(D_{\boldsymbol{\theta}})} \max_{\boldsymbol{\beta} \in \mathbb{B}(D_{\boldsymbol{\beta}}), \boldsymbol{\mu} \in \mathbb{R}_+^{\mathcal{X} \times \mathcal{A}}} \mathfrak{L}(\boldsymbol{\beta}, \boldsymbol{\mu}; \rho, \boldsymbol{v}, \boldsymbol{\theta}).$$

As done in the main text, we eliminate the high-dimensional variables $\boldsymbol{v}$ and $\boldsymbol{\mu}$ by committing to the choices $\boldsymbol{v} = \boldsymbol{v}_{\boldsymbol{\theta}, \pi}$ and $\boldsymbol{\mu} = \boldsymbol{\mu}_{\boldsymbol{\beta}, \pi}$ defined as

$$v_{\boldsymbol{\theta}, \pi}(x) = \sum_a \pi(a|x) \langle \boldsymbol{\theta}, \boldsymbol{\varphi}(x, a) \rangle,$$
$$\mu_{\boldsymbol{\beta}, \pi}(x, a) = \pi(a|x) \langle \boldsymbol{\psi}(x), \boldsymbol{\Lambda}^c \boldsymbol{\beta} \rangle.$$

This makes it possible to express the Lagrangian in terms of only $\boldsymbol{\beta}, \pi, \rho$ and $\boldsymbol{\theta}$:

$$f(\boldsymbol{\beta}, \pi; \rho, \boldsymbol{\theta}) = \rho + \langle \boldsymbol{\beta}, \boldsymbol{\Lambda}^c[\boldsymbol{\omega} + \boldsymbol{\Psi v}_{\boldsymbol{\theta}, \pi} - \boldsymbol{\theta} - \rho \boldsymbol{\varrho}] \rangle + \langle \boldsymbol{\mu}_{\boldsymbol{\beta}, \pi}, \boldsymbol{\Phi \theta} - \boldsymbol{Ev}_{\boldsymbol{\theta}, \pi} \rangle$$
$$= \rho + \langle \boldsymbol{\beta}, \boldsymbol{\Lambda}^c[\boldsymbol{\omega} + \boldsymbol{\Psi v}_{\boldsymbol{\theta}, \pi} - \boldsymbol{\theta} - \rho \boldsymbol{\varrho}] \rangle$$

The remaining low-dimensional variables $\boldsymbol{\beta}, \rho, \boldsymbol{\theta}$ are then updated using stochastic gradient descent/ascent. For this purpose it is useful to express the partial derivatives of the Lagrangian with respect to said variables:

$$\boldsymbol{g}_{\boldsymbol{\beta}} = \boldsymbol{\Lambda}^c[\boldsymbol{\omega} + \boldsymbol{\Psi v}_{\boldsymbol{\theta}, \pi} - \boldsymbol{\theta} - \rho \boldsymbol{\varrho}]$$
$$g_{\rho} = 1 - \langle \boldsymbol{\beta}, \boldsymbol{\Lambda}^c \boldsymbol{\varrho} \rangle$$
$$\boldsymbol{g}_{\boldsymbol{\theta}} = \boldsymbol{\Phi}^{\mathsf{T}} \boldsymbol{\mu}_{\boldsymbol{\beta}, \pi} - \boldsymbol{\Lambda}^c \boldsymbol{\beta}$$

## C.1 Algorithm for average-reward MDPs

Our algorithm for the AMDP setting has the same double-loop structure as the one for the discounted setting. In particular, the algorithm performs a sequence of outer updates $t = 1, 2, \ldots, T$ on the policies $\pi_t$ and the iterates $\boldsymbol{\beta}_t$, and then performs a sequence of updates $i = 1, 2, \ldots, K$ in the inner loop to evaluate the policies and produce $\boldsymbol{\theta}_t, \rho_t$ and $\boldsymbol{v}_t$. Thanks to the reparametrization $\boldsymbol{\beta} = \boldsymbol{\Lambda}^{-c} \boldsymbol{\lambda}$, fixing $\pi_t = \mathrm{softmax}(\sum_{k=1}^{t-1} \boldsymbol{\Phi \theta}_k)$, $v_t(x) = \sum_{a \in \mathcal{A}} \pi_t(a|x) \langle \boldsymbol{\varphi}(x, a), \boldsymbol{\theta}_t \rangle$ for $x \in \mathcal{X}$, and $\mu_t(x, a) = \pi_t(a|x) \langle \boldsymbol{\psi}(x), \boldsymbol{\Lambda}^c \boldsymbol{\beta}_t \rangle$ in round $t$ we can obtain unbiased estimates of the gradients of $f$ with respect to $\boldsymbol{\theta}, \boldsymbol{\beta}$, and $\rho$. For each primal update $t$, the algorithm uses a single sample transition $(X_t, A_t, R_t, X_t')$ generated by the behavior policy $\pi_B$ to compute an unbiased estimator

**Algorithm 2** Offline primal-dual method for Average-reward MDPs

---

**Input:** Learning rates $\zeta, \alpha, \xi, \eta$, initial iterates $\boldsymbol{\beta}_1 \in \mathbb{B}(D_{\boldsymbol{\beta}}), \rho_0 \in [0, 1], \boldsymbol{\theta}_0 \in \mathbb{B}(D_{\boldsymbol{\theta}}), \pi_1 \in \Pi,$

**for** $t = 1$ **to** $T$ **do**
   *// Stochastic gradient descent*:
   Initialize: $\boldsymbol{\theta}_t^{(1)} = \boldsymbol{\theta}_{t-1}$;
   **for** $i = 1$ **to** $K$ **do**
      Obtain sample $W_{t,i} = (X_{t,i}, A_{t,i}, R_{t,i}, X'_{t,i})$;
      Sample $A'_{t,i} \sim \pi_t(\cdot|X'_{t,i})$;

      Compute $\tilde{g}_{\rho,t,i} = 1 - \langle \boldsymbol{\varphi}_{t,i}, \boldsymbol{\Lambda}^{c-1}\boldsymbol{\beta}_t \rangle$;
            $\tilde{\boldsymbol{g}}_{\boldsymbol{\theta},t,i} = \boldsymbol{\varphi}'_{t,i} \langle \boldsymbol{\varphi}_{t,i}, \boldsymbol{\Lambda}^{c-1}\boldsymbol{\beta}_t \rangle - \boldsymbol{\varphi}_{t,i} \langle \boldsymbol{\varphi}_{t,i}, \boldsymbol{\Lambda}^{c-1}\boldsymbol{\beta}_t \rangle$;

      Update $\rho_t^{(i+1)} = \Pi_{[0,1]}(\rho_t^{(i)} - \xi\tilde{g}_{\rho,t,i})$;
            $\boldsymbol{\theta}_t^{(i+1)} = \Pi_{\mathbb{B}(D_{\boldsymbol{\theta}})}(\boldsymbol{\theta}_t^{(i)} - \eta\tilde{\boldsymbol{g}}_{\boldsymbol{\theta},t,i})$.
   **end for**
   Compute $\rho_t = \frac{1}{K}\sum_{i=1}^K \rho_t^{(i)}$;
         $\boldsymbol{\theta}_t = \frac{1}{K}\sum_{i=1}^K \boldsymbol{\theta}_t^{(i)}$;

   *// Stochastic gradient ascent*:
   Obtain sample $W_t = (X_t, A_t, R_t, X'_t)$;
   Compute $v_t(X'_t) = \sum_a \pi_t(a|X'_t) \langle \boldsymbol{\varphi}(X'_t, a), \boldsymbol{\theta}_t \rangle$;
   Compute $\tilde{\boldsymbol{g}}_{\boldsymbol{\beta},t} = \boldsymbol{\Lambda}^{c-1}\boldsymbol{\varphi}_t[R_t + v_t(X'_t) - \langle \boldsymbol{\theta}_t, \boldsymbol{\varphi}_t \rangle - \rho_t]$;
   Update $\boldsymbol{\beta}_{t+1} = \Pi_{\mathbb{B}(D_{\boldsymbol{\beta}})}(\boldsymbol{\beta}_t + \zeta\tilde{\boldsymbol{g}}_{\boldsymbol{\beta},t})$;

   *// Policy update*:
   Compute $\pi_{t+1} = \sigma\left(\alpha\sum_{k=1}^t \boldsymbol{\Phi}\boldsymbol{\theta}_k\right)$.
**end for**
**Return:** $\pi_J$ with $J \sim \mathcal{U}(T)$.

---

of the first gradient $g_{\boldsymbol{\beta}}$ for that round as $\tilde{\boldsymbol{g}}_{\boldsymbol{\beta},t} = \boldsymbol{\Lambda}^{c-1}\boldsymbol{\varphi}_t[R_t + v_t(X'_t) - \langle \boldsymbol{\theta}_t, \boldsymbol{\varphi}_t \rangle - \rho_t]$. Then, in iteration $i = 1, \cdots, K$ of the inner loop within round $t$, we sample transitions $(X_{t,i}, A_{t,i}, R_{t,i}, X'_{t,i})$ to compute gradient estimators with respect to $\rho$ and $\boldsymbol{\theta}$ as:

$$\tilde{g}_{\rho,t,i} = 1 - \langle \boldsymbol{\varphi}_{t,i}, \boldsymbol{\Lambda}^{c-1}\boldsymbol{\beta}_t \rangle$$
$$\tilde{\boldsymbol{g}}_{\boldsymbol{\theta},t,i} = \boldsymbol{\varphi}'_{t,i} \langle \boldsymbol{\varphi}_{t,i}, \boldsymbol{\Lambda}^{c-1}\boldsymbol{\beta}_t \rangle - \boldsymbol{\varphi}_{t,i} \langle \boldsymbol{\varphi}_{t,i}, \boldsymbol{\Lambda}^{c-1}\boldsymbol{\beta}_t \rangle.$$

We have used the shorthand notation $\boldsymbol{\varphi}_{t,i} = \varphi(X_{t,i}, A_{t,i})$, $\boldsymbol{\varphi}'_{t,i} = \varphi(X'_{t,i}, A'_{t,i})$. The update steps are detailed in the pseudocode presented as Algorithm 2.

We now state the general form of our main result for this setting in Theorem C.1 below.

**Theorem C.1.** *Consider a linear MDP (Definition 2.1) such that $\boldsymbol{\theta}^\pi \in \mathbb{B}(D_{\boldsymbol{\theta}})$ for all $\pi \in \Pi$. Further, suppose that $C_{\varphi,c}(\pi^*; \pi_B) \leq D_{\boldsymbol{\beta}}$. Then, for any comparator policy $\pi^* \in \Pi$, the policy output by Algorithm 2 satisfies:*

$$\mathbb{E}\left[\langle \boldsymbol{\mu}^{\pi^*} - \boldsymbol{\mu}^{\pi_{out}}, \boldsymbol{r} \rangle\right] \leq \frac{2D_{\boldsymbol{\beta}}^2}{\zeta T} + \frac{\log|\mathcal{A}|}{\alpha T} + \frac{1}{2\xi K} + \frac{2D_{\boldsymbol{\theta}}^2}{\eta K} + \frac{\zeta G_{\boldsymbol{\beta},c}^2}{2} + \frac{\alpha D_{\boldsymbol{\theta}}^2 D_{\boldsymbol{\varphi}}^2}{2} + \frac{\xi G_{\rho,c}^2}{2} + \frac{\eta G_{\boldsymbol{\theta},c}^2}{2},$$

*where*

$$G_{\boldsymbol{\beta},c}^2 = \text{Tr}(\boldsymbol{\Lambda}^{2c-1})(1 + 2D_{\boldsymbol{\theta}}D_{\boldsymbol{\varphi}})^2, \tag{24}$$

$$G_{\rho,c}^2 = 2\left(1 + D_{\boldsymbol{\beta}}^2 \|\boldsymbol{\Lambda}\|_2^{2c-1}\right), \tag{25}$$

$$G_{\boldsymbol{\theta},c}^2 = 4D_{\boldsymbol{\varphi}}^2 D_{\boldsymbol{\beta}}^2 \|\boldsymbol{\Lambda}\|_2^{2c-1}. \tag{26}$$

583 *In particular, using learning rates $\zeta = \frac{2D_{\boldsymbol{\beta}}}{G_{\boldsymbol{\beta},c}\sqrt{T}}$, $\alpha = \frac{\sqrt{2\log|\mathcal{A}|}}{D_{\boldsymbol{\theta}}D_{\boldsymbol{\varphi}}\sqrt{T}}$, $\xi = \frac{1}{G_{\rho,c}\sqrt{K}}$, and $\eta = \frac{2D_{\boldsymbol{\theta}}}{G_{\boldsymbol{\theta},c}\sqrt{K}}$,*

584 *and setting $K = T \cdot \frac{4D_{\boldsymbol{\beta}^2}G_{\boldsymbol{\beta},c}^2 + 2D_{\boldsymbol{\theta}}^2 D_{\boldsymbol{\varphi}}^2 \log|\mathcal{A}|}{G_{\rho,c}^2 + 4D_{\boldsymbol{\theta}}^2 G_{\boldsymbol{\theta},c}^2}$, we achieve $\mathbb{E}\left[\langle \boldsymbol{\mu}^{\pi^*} - \boldsymbol{\mu}^{\boldsymbol{\pi}_{out}}, \boldsymbol{r}\rangle\right] \leq \epsilon$ with a number*

585 *of samples $n_\epsilon$ that is*

$$O\left(\epsilon^{-4} D_{\boldsymbol{\theta}}^4 D_{\boldsymbol{\varphi}}^4 D_{\boldsymbol{\beta}}^4 \operatorname{Tr}(\boldsymbol{\Lambda}^{2c-1}) \|\boldsymbol{\Lambda}\|_2^{2(2c-1)} \log|\mathcal{A}|\right).$$

586 By remark A.2, we have that $n_\epsilon$ is of order $O\left(\varepsilon^{-4} D_{\boldsymbol{\theta}}^4 D_{\boldsymbol{\varphi}}^{12c-2} D_{\boldsymbol{\beta}}^4 d^{2-2c} \log|\mathcal{A}|\right)$.

587 **Corollary C.2.** *Assume that the bound of the feature vectors $D_{\boldsymbol{\varphi}}$ is of order $O(1)$, that $D_{\boldsymbol{\omega}} = D_{\boldsymbol{\psi}} =$*
588 $\sqrt{d}$ *which together imply $D_{\boldsymbol{\theta}} \leq \sqrt{d} + 1 + \sqrt{d}D_q = O(\sqrt{d}D_q)$ and that $D_{\boldsymbol{\beta}} = c \cdot C_{\varphi,c}(\pi^*; \pi_B)$ for*
589 *some positive universal constant c. Then, under the same assumptions of Theorem 3.2, $n_\varepsilon$ is of order*
590 $O\left(\varepsilon^{-4} D_q^4 C_{\varphi,c}(\pi^*; \pi_B)^2 d^{4-2c} \log|\mathcal{A}|\right)$.

591 Recall that $C_{\varphi,1/2}$ is always smaller than $C_{\varphi,1}$, but using $c = 1/2$ in the algorithm requires knowledge
592 of the covariance matrix $\boldsymbol{\Lambda}$, and results in a slightly worse dependence on the dimension.

593 The proof of Theorem C.1 mainly follows the same steps as in the discounted case, with some added
594 difficulty that is inherent in the more challenging average-reward setup. Some key challenges include
595 treating the additional optimization variable $\rho$ and coping with the fact that the optimal parameters
596 $\boldsymbol{\theta}^*$ and $\boldsymbol{\beta}^*$ are not necessarily unique any more.

597 ## C.2 Analysis

598 We now prove our main result regarding the AMDP setting in Theorem C.1. Following the derivations
599 in the main text, we study the dynamic duality gap defined as

$$\mathcal{G}_T(\boldsymbol{\beta}^*, \pi^*; \rho_{1:T}^*, \boldsymbol{\theta}_{1:T}^*) = \frac{1}{T}\sum_{t=1}^{T}\left(f(\boldsymbol{\beta}^*, \pi^*; \rho_t, \boldsymbol{\theta}_t) - f(\boldsymbol{\beta}_t, \pi_t; \rho_t^*, \boldsymbol{\theta}_t^*)\right). \tag{27}$$

600 First we show in Lemma C.3 below that, for appropriately chosen comparator points, the expected
601 suboptimality of the policy returned by Algorithm 2 can be upper bounded in terms of the expected
602 dynamic duality gap.

603 **Lemma C.3.** *Let $\boldsymbol{\theta}_t^*$ such that $\langle \boldsymbol{\varphi}(x,a), \boldsymbol{\theta}_t^* \rangle = \langle \boldsymbol{\varphi}(x,a), \boldsymbol{\theta}^{\pi_t} \rangle - \inf_{(x,a)\in\mathcal{X}\times\mathcal{A}} \langle \boldsymbol{\varphi}(x,a), \boldsymbol{\theta}^{\pi_t} \rangle$ holds*
604 *for all $(x,a) \in \mathcal{X} \times \mathcal{A}$, and let $\boldsymbol{v}_t^*$ be defined as $\boldsymbol{v}_t^*(x) = \sum_{a\in\mathcal{A}} \pi_t(a|x) \langle \boldsymbol{\varphi}(x,a), \boldsymbol{\theta}_t^* \rangle$ for all x. Also,*
605 *let $\rho_t^* = \rho^{\pi_t}$, $\pi^*$ be an optimal policy, and $\boldsymbol{\beta}^* = \boldsymbol{\Lambda}^{-c}\boldsymbol{\Phi}^{\top}\boldsymbol{\mu}^*$ where $\boldsymbol{\mu}^*$ is the occupancy measure of*
606 $\pi^*$. *Then, the suboptimality gap of the policy output by Algorithm 2 satisfies*

$$\mathbb{E}_T\left[\langle \boldsymbol{\mu}^* - \boldsymbol{\mu}^{\boldsymbol{\pi}_{out}}, \boldsymbol{r}\rangle\right] = \mathcal{G}_T(\boldsymbol{\beta}^*, \pi^*; \rho_{1:T}^*, \boldsymbol{\theta}_{1:T}^*).$$

607 *Proof.* Substituting $(\boldsymbol{\beta}^*, \pi^*) = (\boldsymbol{\Lambda}^{-c}\boldsymbol{\Phi}^{\top}\boldsymbol{\mu}^*, \pi^*)$ in the first term of the dynamic duality gap we have

$$
\begin{aligned}
f(\boldsymbol{\beta}^*, \pi^*; \rho_t, \boldsymbol{\theta}_t) &= \rho_t + \langle \boldsymbol{\Lambda}^{-c}\boldsymbol{\Phi}^{\top}\boldsymbol{\mu}^*, \boldsymbol{\Lambda}^c[\boldsymbol{\omega} + \boldsymbol{\Psi}\boldsymbol{v}_{\boldsymbol{\theta}_t,\pi^*} - \boldsymbol{\theta}_t - \rho_t\boldsymbol{\varrho}]\rangle \\
&= \rho_t + \langle \boldsymbol{\mu}^*, \boldsymbol{r} + \boldsymbol{P}\boldsymbol{v}_{\boldsymbol{\theta}_t,\pi^*} - \boldsymbol{\Phi}\boldsymbol{\theta}_t - \rho_t\mathbf{1}\rangle \\
&= \langle \boldsymbol{\mu}^*, \boldsymbol{r}\rangle + \langle \boldsymbol{\mu}^*, \boldsymbol{E}\boldsymbol{v}_{\boldsymbol{\theta}_t,\pi^*} - \boldsymbol{\Phi}\boldsymbol{\theta}_t\rangle + \rho_t[1 - \langle \boldsymbol{\mu}^*, \mathbf{1}\rangle] \\
&= \langle \boldsymbol{\mu}^*, \boldsymbol{r}\rangle.
\end{aligned}
$$

608 Here, we have used the fact that $\boldsymbol{\mu}^*$ is a valid occupancy measure, so it satisfies the flow constraint
609 $\boldsymbol{E}^{\top}\boldsymbol{\mu}^* = \boldsymbol{P}^{\top}\boldsymbol{\mu}^*$ and the normalization constraint $\langle \boldsymbol{\mu}^*, \mathbf{1}\rangle = 1$. Also, in the last step we have used the
610 definition of $\boldsymbol{v}_{\boldsymbol{\theta}_t,\pi^*}$ that guarantees that the following equality holds:

$$\langle \boldsymbol{\mu}^*, \boldsymbol{\Phi}\boldsymbol{\theta}_t\rangle = \sum_{x\in\mathcal{X}} \nu^*(x) \sum_{a\in\mathcal{A}} \pi^*(a|x) \langle \boldsymbol{\theta}_t, \boldsymbol{\varphi}(x,a)\rangle = \sum_{x\in\mathcal{X}} \nu^*(x) v_{\boldsymbol{\theta}_t,\pi^*}(x) = \langle \boldsymbol{\mu}^*, \boldsymbol{E}\boldsymbol{v}_{\boldsymbol{\theta}_t,\pi^*}\rangle.$$

For the second term in the dynamic duality gap, using that $\pi_t$ is $\mathcal{F}_{t-1}$-measurable we write

$$
\begin{aligned}
f(\boldsymbol{\beta}_t, &\pi_t; \rho_t^*, \boldsymbol{\theta}_t^*)\\
&= \rho_t^* + \langle \boldsymbol{\beta}_t, \boldsymbol{\Lambda}^c[\boldsymbol{\omega} + \boldsymbol{\Psi} \boldsymbol{v}_{\boldsymbol{\theta}_t^*, \pi_t} - \boldsymbol{\theta}_t^* - \rho_t^* \boldsymbol{\varrho}]\rangle\\
&= \rho_t^* + \langle \boldsymbol{\beta}_t, \boldsymbol{\Lambda}^{c-1}\mathbb{E}_t\left[\boldsymbol{\varphi}_t \boldsymbol{\varphi}_t^\mathsf{T}[\boldsymbol{\omega} + \boldsymbol{\Psi} \boldsymbol{v}_{\boldsymbol{\theta}_t^*, \pi_t} - \boldsymbol{\theta}_t^* - \rho_t^* \boldsymbol{\varrho}]\right]\rangle\\
&= \rho_t^* + \left\langle \boldsymbol{\beta}_t, \mathbb{E}_t\left[\boldsymbol{\Lambda}^{c-1}\boldsymbol{\varphi}_t\left[R_t + \sum_{x,a} p(x|X_t, A_t)\pi_t(a|x)\langle\boldsymbol{\varphi}(x,a), \boldsymbol{\theta}_t^*\rangle - \langle\boldsymbol{\varphi}(X_t, A_t), \boldsymbol{\theta}_t^*\rangle - \rho_t^*\right]\right]\right\rangle\\
&= \rho^{\pi_t} + \left\langle \boldsymbol{\beta}_t, \mathbb{E}_t\left[\boldsymbol{\Lambda}^{c-1}\boldsymbol{\varphi}_t\left[R_t + \sum_{x,a} p(x|X_t, A_t)\pi_t(a|x)\langle\boldsymbol{\varphi}(x,a), \boldsymbol{\theta}^{\pi_t}\rangle - \langle\boldsymbol{\varphi}(X_t, A_t), \boldsymbol{\theta}^{\pi_t}\rangle - \rho^{\pi_t}\right]\right]\right\rangle\\
&= \rho^{\pi_t} + \langle \boldsymbol{\beta}_t, \mathbb{E}_t\left[\boldsymbol{\Lambda}^{c-1}\boldsymbol{\varphi}_t[r(X_t, A_t) + \langle p(\cdot|X_t, A_t), v^{\pi_t}\rangle - q^{\pi_t}(X_t, A_t) - \rho^{\pi_t}]\right]\rangle\\
&= \rho^{\pi_t} = \langle \boldsymbol{\mu}^{\pi_t}, r\rangle,
\end{aligned}
$$

where in the fourth equality we used that $\langle\boldsymbol{\varphi}(x,a) - \boldsymbol{\varphi}(x',a'), \boldsymbol{\theta}_t^*\rangle = \langle\boldsymbol{\varphi}(x,a) - \boldsymbol{\varphi}(x',a'), \theta^{\pi_t}\rangle$ holds for all $x, a, x', a'$ by definition of $\theta_t^*$. Then, the last equality follows from the fact that the Bellman equations for $\pi_t$ imply $q^{\pi_t}(x,a) + \rho^{\pi_t} = r(x,a) + \langle p(\cdot|x,a), \boldsymbol{v}^{\pi_t}\rangle$.

Combining both expressions for $f(\boldsymbol{\beta}^*, \pi^*; \rho_t, \boldsymbol{\theta}_t)$ and $f(\boldsymbol{\beta}_t, \pi_t; \rho_t^*, \boldsymbol{\theta}_t^*)$ in the dynamic duality gap we have:

$$
\mathcal{G}_T(\boldsymbol{\beta}^*, \pi^*; \rho_{1:T}^*, \boldsymbol{\theta}_{1:T}^*) = \frac{1}{T}\sum_{t=1}^T (\langle\boldsymbol{\mu}^* - \boldsymbol{\mu}^{\pi_t}, r\rangle) = \mathbb{E}_T\left[\langle\boldsymbol{\mu}^* - \boldsymbol{\mu}^{\pi_\text{out}}, r\rangle\right].
$$

The second equality follows from noticing that, since $\boldsymbol{\pi}_\text{out}$ is sampled uniformly from $\{\pi_t\}_{t=1}^T$, $\mathbb{E}\left[\langle\boldsymbol{\mu}^{\boldsymbol{\pi}_\text{out}}, r\rangle\right] = \frac{1}{T}\sum_{t=1}^T \mathbb{E}\left[\langle\boldsymbol{\mu}^{\pi_t}, r\rangle\right]$. This completes the proof. $\qquad\square$

Having shown that for well-chosen comparator points the dynamic duality gap equals the expected suboptimality of the output policy of Algorithm 2, it remains to relate the gap to the optimization error of the primal-dual procedure. This is achieved in the following lemma.

**Lemma C.4.** *For the same choice of comparators $(\boldsymbol{\beta}^*, \pi^*; \rho_{1:T}^*, \boldsymbol{\theta}_{1:T}^*)$ as in Lemma C.3 the dynamic duality gap associated with the iterates produced by Algorithm 2 satisfies*

$$
\begin{aligned}
\mathbb{E}\left[\mathcal{G}_T(\boldsymbol{\beta}^*, \pi^*; \rho_{1:T}^*, \boldsymbol{\theta}_{1:T}^*)\right]&\\
&\leq \frac{2D_{\boldsymbol{\beta}}^2}{\zeta T} + \frac{\mathcal{H}(\pi^*\|\pi_1)}{\alpha T} + \frac{1}{2\xi K} + \frac{2D_{\boldsymbol{\theta}}^2}{\eta K}\\
&\quad + \frac{\zeta \operatorname{Tr}(\boldsymbol{\Lambda}^{2c-1})(1 + 2D_{\boldsymbol{\varphi}}D_{\boldsymbol{\theta}})^2}{2} + \frac{\alpha D_{\boldsymbol{\varphi}}^2 D_{\boldsymbol{\theta}}^2}{2} + \xi\left(1 + D_{\boldsymbol{\beta}}^2\|\boldsymbol{\Lambda}\|_2^{2c-1}\right) + 2\eta D_{\boldsymbol{\varphi}}^2 D_{\boldsymbol{\beta}}^2\|\boldsymbol{\Lambda}\|_2^{2c-1}.
\end{aligned}
$$

*Proof.* The first part of the proof follows from recognising that the dynamic duality gap can be rewritten in terms of the total regret of the primal and dual players in the algorithm. Formally, we write

$$
\begin{aligned}
\mathcal{G}_T(&\boldsymbol{\beta}^*, \pi^*; \rho_{1:T}^*, \boldsymbol{\theta}_{1:T}^*)\\
&= \frac{1}{T}\sum_{t=1}^T (f(\boldsymbol{\beta}^*, \pi^*; \rho_t, \boldsymbol{\theta}_t) - f(\boldsymbol{\beta}_t, \pi_t; \rho_t, \boldsymbol{\theta}_t)) + \frac{1}{T}\sum_{t=1}^T (f(\boldsymbol{\beta}_t, \pi_t; \rho_t, \boldsymbol{\theta}_t) - f(\boldsymbol{\beta}_t, \pi_t; \rho_t^*, \boldsymbol{\theta}_t^*)).
\end{aligned}
$$

Using that $\boldsymbol{\beta}^* = \boldsymbol{\Lambda}^{-c}\boldsymbol{\Phi}^\mathsf{T}\boldsymbol{\mu}^*$, $\boldsymbol{q}_t = \langle\boldsymbol{\varphi}(x,a), \boldsymbol{\theta}_t\rangle$, $\boldsymbol{v}_t = \boldsymbol{v}_{\boldsymbol{\theta}_t, \pi_t}$ and that $\boldsymbol{g}_{\boldsymbol{\beta},t} = \boldsymbol{\Lambda}^c[\boldsymbol{\omega} + \boldsymbol{\Psi}\boldsymbol{v}_t - \boldsymbol{\theta}_t - \rho_t\boldsymbol{\varrho}]$, we see that term in the first sum can be simply rewritten as

$$
\begin{aligned}
f(\boldsymbol{\beta}^*, &\pi^*; \rho_t, \boldsymbol{\theta}_t) - f(\boldsymbol{\beta}_t, \pi_t; \rho_t, \boldsymbol{\theta}_t)\\
&= \langle\boldsymbol{\beta}^*, \boldsymbol{\Lambda}^c[\boldsymbol{\omega} + \boldsymbol{\Psi}\boldsymbol{v}_{\boldsymbol{\theta}_t, \pi^*} - \boldsymbol{\theta}_t - \rho_t\boldsymbol{\varrho}]\rangle - \langle\boldsymbol{\beta}_t, \boldsymbol{\Lambda}^c[\boldsymbol{\omega} + \boldsymbol{\Psi}\boldsymbol{v}_{\boldsymbol{\theta}_t, \pi_t} - \boldsymbol{\theta}_t - \rho_t\boldsymbol{\varrho}]\rangle\\
&= \langle\boldsymbol{\beta}^* - \boldsymbol{\beta}_t, \boldsymbol{\Lambda}^c[\boldsymbol{\omega} + \boldsymbol{\Psi}\boldsymbol{v}_t - \boldsymbol{\theta}_t - \rho_t\boldsymbol{\varrho}]\rangle + \langle\boldsymbol{\Psi}^\mathsf{T}\boldsymbol{\Lambda}^c\boldsymbol{\beta}^*, \boldsymbol{v}_{\boldsymbol{\theta}_t, \pi^*} - \boldsymbol{v}_{\boldsymbol{\theta}_t, \pi_t}\rangle\\
&= \langle\boldsymbol{\beta}^* - \boldsymbol{\beta}_t, \boldsymbol{g}_{\boldsymbol{\beta},t}\rangle + \sum_{x \in \mathcal{X}} \nu^*(x)\langle\pi^*(\cdot|x) - \pi_t(\cdot|x), \boldsymbol{q}_t(x, \cdot)\rangle.
\end{aligned}
$$

In a similar way, using that $\boldsymbol{E}^\mathsf{T}\boldsymbol{\mu}_t = \boldsymbol{\Psi}^\mathsf{T}\boldsymbol{\Lambda}^c\boldsymbol{\beta}_t$ and the definitions of the gradients $g_{\rho,t}$ and $\boldsymbol{g}_{\boldsymbol{\theta},t}$, the term in the second sum can be rewritten as

$$
\begin{aligned}
f(\boldsymbol{\beta}_t, &\pi_t; \rho_t, \boldsymbol{\theta}_t) - f(\boldsymbol{\beta}_t, \pi_t; \rho_t^*, \boldsymbol{\theta}_t^*) \\
&= \rho_t + \langle \boldsymbol{\beta}_t\,, \boldsymbol{\Lambda}^c[\boldsymbol{\omega} + \boldsymbol{\Psi}\boldsymbol{v}_{\boldsymbol{\theta}_t,\pi_t} - \boldsymbol{\theta}_t - \rho_t\boldsymbol{\varrho}]\rangle - \rho_t^* - \langle \boldsymbol{\beta}_t\,, \boldsymbol{\Lambda}^c[\boldsymbol{\omega} + \boldsymbol{\Psi}\boldsymbol{v}_{\boldsymbol{\theta}_t^*,\pi_t} - \boldsymbol{\theta}_t^* - \rho_t^*\boldsymbol{\varrho}]\rangle \\
&= (\rho_t - \rho_t^*)[1 - \langle \boldsymbol{\beta}_t, \boldsymbol{\Lambda}^c\boldsymbol{\varrho}\rangle] - \langle \boldsymbol{\theta}_t - \boldsymbol{\theta}_t^*, \boldsymbol{\Lambda}^c\boldsymbol{\beta}_t\rangle + \langle \boldsymbol{E}^\mathsf{T}\boldsymbol{\mu}_t, \boldsymbol{v}_{\boldsymbol{\theta}_t,\pi_t} - \boldsymbol{v}_{\boldsymbol{\theta}_t^*,\pi_t}\rangle \\
&= (\rho_t - \rho_t^*)[1 - \langle \boldsymbol{\beta}_t, \boldsymbol{\Lambda}^c\boldsymbol{\varrho}\rangle] - \langle \boldsymbol{\theta}_t - \boldsymbol{\theta}_t^*, \boldsymbol{\Lambda}^c\boldsymbol{\beta}_t\rangle + \langle \boldsymbol{\Phi}^\mathsf{T}\boldsymbol{\mu}_t, \boldsymbol{\theta}_t - \boldsymbol{\theta}_t^*\rangle \\
&= (\rho_t - \rho_t^*)[1 - \langle \boldsymbol{\beta}_t, \boldsymbol{\Lambda}^c\boldsymbol{\varrho}\rangle] + \langle \boldsymbol{\theta}_t - \boldsymbol{\theta}_t^*, \boldsymbol{\Phi}^\mathsf{T}\boldsymbol{\mu}_t - \boldsymbol{\Lambda}^c\boldsymbol{\beta}_t\rangle \\
&= (\rho_t - \rho_t^*)g_{\rho,t} + \langle \boldsymbol{\theta}_t - \boldsymbol{\theta}_t^*, \boldsymbol{g}_{\boldsymbol{\theta},t}\rangle = \frac{1}{K}\sum_{i=1}^{K}\left((\rho_t^{(i)} - \rho_t^*)g_{\rho,t} + \left\langle \boldsymbol{\theta}_t^{(i)} - \boldsymbol{\theta}_t^*, \boldsymbol{g}_{\boldsymbol{\theta},t}\right\rangle\right).
\end{aligned}
$$

Combining both terms in the duality gap concludes the first part of the proof. As shown below the dynamic duality gap is written as the error between iterates of the algorithm from respective comparator points in the direction of the exact gradients. Formally, we have

$$
\begin{aligned}
\mathcal{G}_T(\boldsymbol{\beta}^*, \pi^*; \rho_{1:T}^*, \boldsymbol{\theta}_{1:T}^*) = \frac{1}{T}\sum_{t=1}^{T}&\left(\langle \boldsymbol{\beta}^* - \boldsymbol{\beta}_t\,, \boldsymbol{g}_{\boldsymbol{\beta},t}\rangle + \sum_{x\in\mathcal{X}}\nu^*(x)\langle \pi^*(\cdot|x) - \pi_t(\cdot|x), \boldsymbol{q}_t(x,\cdot)\rangle\right) \\
&+ \frac{1}{TK}\sum_{t=1}^{T}\sum_{i=1}^{K}\left((\rho_t^{(i)} - \rho_t^*)g_{\rho,t} + \left\langle \boldsymbol{\theta}_t^{(i)} - \boldsymbol{\theta}_t^*, \boldsymbol{g}_{\boldsymbol{\theta},t}\right\rangle\right).
\end{aligned}
$$

Then, implementing techniques from stochastic gradient descent analysis in the proof of Lemmas C.5 to C.7 and mirror descent analysis in Lemma B.3, the expected dynamic duality gap can be upper bounded as follows:

$$
\begin{aligned}
\mathbb{E}\left[\mathcal{G}_T(\boldsymbol{\beta}^*, \pi^*; \rho_{1:T}^*, \boldsymbol{\theta}_{1:T}^*)\right]& \\
\leq \frac{2D_{\boldsymbol{\beta}}^2}{\zeta T} + \frac{\mathcal{H}(\pi^*\|\pi_1)}{\alpha T} &+ \frac{1}{2\xi K} + \frac{2D_{\boldsymbol{\theta}}^2}{\eta K} \\
&+ \frac{\zeta\,\mathrm{Tr}(\boldsymbol{\Lambda}^{2c-1})(1+2D_{\boldsymbol{\varphi}}D_{\boldsymbol{\theta}})^2}{2} + \frac{\alpha D_{\boldsymbol{\varphi}}^2 D_{\boldsymbol{\theta}}^2}{2} + \xi\left(1 + D_{\boldsymbol{\beta}}^2\|\boldsymbol{\Lambda}\|_2^{2c-1}\right) + 2\eta D_{\boldsymbol{\varphi}}^2 D_{\boldsymbol{\beta}}^2\|\boldsymbol{\Lambda}\|_2^{2c-1}.
\end{aligned}
$$

This completes the proof $\qquad\square$

**Proof of Theorem C.1** First, we bound the expected suboptimality gap by combining Lemma C.3 and C.4. Next, bearing in mind that the algorithm only needs $T(K+1)$ total samples from the behavior policy we optimize the learning rates to obtain a bound on the sample complexity, thus completing the proof. $\qquad\square$

## C.3 Missing proofs for Lemma C.4

In this section we prove Lemmas C.5 to C.7 used in the proof of Lemma C.4. It is important to recall that sample transitions $(X_k, A_k, R_t, X_k')$ in any iteration $k$ are generated in the following way: we draw i.i.d state-action pairs $(X_k, A_k)$ from $\boldsymbol{\mu}_B$, and for each state-action pair, the next $X_k'$ is sampled from $p(\cdot|X_k, A_k)$ and immediate reward computed as $R_t = r(X_k, A_k)$. Precisely in iteration $i$ of round $t$ where $k = (t, i)$, since $(X_{t,i}, A_{t,i})$ are sampled i.i.d from $\boldsymbol{\mu}_B$ at this time step, $\mathbb{E}_{t,i}\left[\boldsymbol{\varphi}_{t,i}\boldsymbol{\varphi}_{t,i}^\mathsf{T}\right] = \mathbb{E}_{(x,a)\sim\boldsymbol{\mu}_B}\left[\boldsymbol{\varphi}(x,a)\boldsymbol{\varphi}(x,a)^\mathsf{T}\right] = \boldsymbol{\Lambda}$.

**Lemma C.5.** *The gradient estimator $\tilde{\boldsymbol{g}}_{\boldsymbol{\beta},t}$ satisfies $\mathbb{E}\left[\tilde{\boldsymbol{g}}_{\boldsymbol{\beta},t}|\mathcal{F}_{t-1}, \boldsymbol{\theta}_t\right] = \boldsymbol{g}_{\boldsymbol{\beta},t}$ and*

$$
\mathbb{E}\left[\|\tilde{\boldsymbol{g}}_{\boldsymbol{\beta},t}\|_2^2\right] \leq \mathrm{Tr}(\boldsymbol{\Lambda}^{2c-1})(1+2D_{\boldsymbol{\varphi}}D_{\boldsymbol{\theta}})^2.
$$

*Furthermore, for any $\boldsymbol{\beta}^*$ with $\boldsymbol{\beta}^* \in \mathbb{B}(D_{\boldsymbol{\beta}})$, the iterates $\boldsymbol{\beta}_t$ satisfy*

$$
\mathbb{E}\left[\sum_{t=1}^{T}\langle \boldsymbol{\beta}^* - \boldsymbol{\beta}_t\,, \boldsymbol{g}_{\boldsymbol{\beta},t}\rangle\right] \leq \frac{2D_{\boldsymbol{\beta}}^2}{\zeta} + \frac{\zeta T\,\mathrm{Tr}(\boldsymbol{\Lambda}^{2c-1})(1+2D_{\boldsymbol{\varphi}}D_{\boldsymbol{\theta}})^2}{2}. \tag{28}
$$

*Proof.* For the first part, we remind that $\pi_t$ is $\mathcal{F}_{t-1}$-measurable and $\boldsymbol{v}_t$ is determined given $\pi_t$ and $\boldsymbol{\theta}_t$. Then, we write

$$
\begin{aligned}
\mathbb{E}\left[\tilde{\boldsymbol{g}}_{\boldsymbol{\beta},t}\,|\mathcal{F}_{t-1},\boldsymbol{\theta}_t\right] &= \mathbb{E}\left[\boldsymbol{\Lambda}^{c-1}\boldsymbol{\varphi}_t[R_t + v_t(X_t') - \langle\boldsymbol{\theta}_t,\boldsymbol{\varphi}_t\rangle - \rho_t]\,|\mathcal{F}_{t-1},\boldsymbol{\theta}_t\right] \\
&= \mathbb{E}\left[\boldsymbol{\Lambda}^{c-1}\boldsymbol{\varphi}_t[R_t + \mathbb{E}_{x'\sim p(\cdot|X_t,A_t)}\left[v_t(x')\right] - \langle\boldsymbol{\theta}_t,\boldsymbol{\varphi}_t\rangle - \rho_t]\,|\mathcal{F}_{t-1},\boldsymbol{\theta}_t\right] \\
&= \mathbb{E}\left[\boldsymbol{\Lambda}^{c-1}\boldsymbol{\varphi}_t[R_t + \langle p(\cdot|X_t,A_t),\boldsymbol{v}_t\rangle - \langle\boldsymbol{\theta}_t,\boldsymbol{\varphi}_t\rangle - \rho_t]\,|\mathcal{F}_{t-1},\boldsymbol{\theta}_t\right] \\
&= \mathbb{E}\left[\boldsymbol{\Lambda}^{c-1}\boldsymbol{\varphi}_t\boldsymbol{\varphi}_t^{\mathsf{T}}[\boldsymbol{\omega} + \boldsymbol{\Psi}\boldsymbol{v}_t - \boldsymbol{\theta}_t - \rho_t\boldsymbol{\varrho}]\,|\mathcal{F}_{t-1},\boldsymbol{\theta}_t\right] \\
&= \boldsymbol{\Lambda}^{c-1}\mathbb{E}\left[\boldsymbol{\varphi}_t\boldsymbol{\varphi}_t^{\mathsf{T}}\,|\mathcal{F}_{t-1},\boldsymbol{\theta}_t\right][\boldsymbol{\omega} + \boldsymbol{\Psi}\boldsymbol{v}_t - \boldsymbol{\theta}_t - \rho_t\boldsymbol{\varrho}] \\
&= \boldsymbol{\Lambda}^{c}[\boldsymbol{\omega} + \boldsymbol{\Psi}\boldsymbol{v}_t - \boldsymbol{\theta}_t - \rho_t\boldsymbol{\varrho}] = \boldsymbol{g}_{\boldsymbol{\beta},t}.
\end{aligned}
$$

Next, we use the facts that $r \in [0,1]$ and $\|\boldsymbol{v}_t\|_\infty \le \|\boldsymbol{\Phi}\boldsymbol{\theta}_t\|_\infty \le D_{\boldsymbol{\varphi}}D_{\boldsymbol{\theta}}$ to show the following bound:

$$
\begin{aligned}
\mathbb{E}\left[\|\tilde{\boldsymbol{g}}_{\boldsymbol{\beta},t}\|_2^2\,|\mathcal{F}_{t-1},\boldsymbol{\theta}_t\right] &= \mathbb{E}\left[\left\|\boldsymbol{\Lambda}^{c-1}\boldsymbol{\varphi}_t[R_t + v_t(X_t') - \langle\boldsymbol{\theta}_t,\boldsymbol{\varphi}_t\rangle]\right\|_2^2\,|\mathcal{F}_{t-1},\boldsymbol{\theta}_t\right] \\
&= \mathbb{E}\left[|R_t + v_t(X_t') - \langle\boldsymbol{\theta}_t,\boldsymbol{\varphi}_t\rangle|\left\|\boldsymbol{\Lambda}^{c-1}\boldsymbol{\varphi}_t\right\|_2^2\,|\mathcal{F}_{t-1},\boldsymbol{\theta}_t\right] \\
&\le \mathbb{E}\left[(1+2D_{\boldsymbol{\varphi}}D_{\boldsymbol{\theta}})^2\left\|\boldsymbol{\Lambda}^{c-1}\boldsymbol{\varphi}_t\right\|_2^2\,|\mathcal{F}_{t-1},\boldsymbol{\theta}_t\right] \\
&= (1+2D_{\boldsymbol{\varphi}}D_{\boldsymbol{\theta}})^2\mathbb{E}\left[\boldsymbol{\varphi}_t^{\mathsf{T}}\boldsymbol{\Lambda}^{2(c-1)}\boldsymbol{\varphi}_t\,|\mathcal{F}_{t-1},\boldsymbol{\theta}_t\right] \\
&= (1+2D_{\boldsymbol{\varphi}}D_{\boldsymbol{\theta}})^2\mathbb{E}\left[\mathrm{Tr}(\boldsymbol{\Lambda}^{2(c-1)}\boldsymbol{\varphi}_t\boldsymbol{\varphi}_t^{\mathsf{T}})\,|\mathcal{F}_{t-1},\boldsymbol{\theta}_t\right] \\
&\le \mathrm{Tr}(\boldsymbol{\Lambda}^{2c-1})(1+2D_{\boldsymbol{\varphi}}D_{\boldsymbol{\theta}})^2.
\end{aligned}
$$

The last step follows from the fact that $\boldsymbol{\Lambda}$, hence also $\boldsymbol{\Lambda}^{2c-1}$, is positive semi-definite, so $\mathrm{Tr}(\boldsymbol{\Lambda}^{2c-1}) \ge 0$. Having shown these properties, we appeal to the standard analysis of online gradient descent stated as Lemma D.1 to obtain the following bound

$$
\mathbb{E}\left[\sum_{t=1}^T\langle\boldsymbol{\beta}^* - \boldsymbol{\beta}_t,\boldsymbol{g}_{\boldsymbol{\beta},t}\rangle\right] \le \frac{\|\boldsymbol{\beta}_1 - \boldsymbol{\beta}^*\|_2^2}{2\zeta} + \frac{\zeta T\,\mathrm{Tr}(\boldsymbol{\Lambda}^{2c-1})(1+2D_{\boldsymbol{\varphi}}D_{\boldsymbol{\theta}})^2}{2}.
$$

Using that $\|\boldsymbol{\beta}^*\|_2 \le D_{\boldsymbol{\beta}}$ concludes the proof. $\qquad\square$

**Lemma C.6.** *The gradient estimator $\tilde{g}_{\rho,t,i}$ satisfies $\mathbb{E}_{t,i}\left[\tilde{g}_{\rho,t,i}\right] = g_{\rho,t}$ and $\mathbb{E}_{t,i}\left[\tilde{g}_{\rho,t,i}^2\right] \le 2 + 2D_{\boldsymbol{\beta}}^2\|\boldsymbol{\Lambda}\|_2^{2c-1}$. Furthermore, for any $\rho_t^* \in [0,1]$, the iterates $\rho_t^{(i)}$ satisfy*

$$
\mathbb{E}\left[\sum_{i=1}^K(\rho_t^{(i)} - \rho_t^*)g_{\rho,t}\right] \le \frac{1}{2\xi} + \xi K\left(1 + \|\boldsymbol{\beta}_t\|_{\boldsymbol{\Lambda}^{2c-1}}^2\right).
$$

*Proof.* For the first part of the proof, we use that $\boldsymbol{\beta}_t$ is $\mathcal{F}_{t,i-1}$-measurable, to obtain

$$
\begin{aligned}
\mathbb{E}_{t,i}\left[\tilde{g}_{\rho,t,i}\right] &= \mathbb{E}_{t,i}\left[1 - \left\langle\boldsymbol{\varphi}_{t,i},\boldsymbol{\Lambda}^{c-1}\boldsymbol{\beta}_t\right\rangle\right] \\
&= \mathbb{E}_{t,i}\left[1 - \left\langle\boldsymbol{\varphi}_{t,i}\boldsymbol{\varphi}_{t,i}^{\mathsf{T}}\boldsymbol{\varrho},\boldsymbol{\Lambda}^{c-1}\boldsymbol{\beta}_t\right\rangle\right] \\
&= 1 - \left\langle\boldsymbol{\Lambda}^c\boldsymbol{\varrho},\boldsymbol{\beta}_t\right\rangle = g_{\rho,t}.
\end{aligned}
$$

In addition, using Young's inequality and $\|\boldsymbol{\beta}_t\|_{\boldsymbol{\Lambda}^{2c-1}}^2 \le D_{\boldsymbol{\beta}}^2\|\boldsymbol{\Lambda}\|_2^{2c-1}$ we show that

$$
\begin{aligned}
\mathbb{E}_{t,i}\left[\tilde{g}_{\rho,t,i}^2\right] &= \mathbb{E}_{t,i}\left[\left(1 - \left\langle\boldsymbol{\varphi}_{t,i},\boldsymbol{\Lambda}^{c-1}\boldsymbol{\beta}_t\right\rangle\right)^2\right] \\
&\le 2 + 2\mathbb{E}_{t,i}\left[\boldsymbol{\beta}_t^{\mathsf{T}}\boldsymbol{\Lambda}^{c-1}\boldsymbol{\varphi}_{t,i}\boldsymbol{\varphi}_{t,i}^{\mathsf{T}}\boldsymbol{\Lambda}^{c-1}\boldsymbol{\beta}_t\right] \\
&= 2 + 2\|\boldsymbol{\beta}_t\|_{\boldsymbol{\Lambda}^{2c-1}}^2 \le 2 + 2D_{\boldsymbol{\beta}}^2\|\boldsymbol{\Lambda}\|_2^{2c-1}.
\end{aligned}
$$

For the second part, we appeal to the standard online gradient descent analysis of Lemma D.1 to bound on the total error of the iterates:

$$
\mathbb{E}\left[\sum_{i=1}^K(\rho_t^{(i)} - \rho_t^*)g_{\rho,t}\right] \le \frac{\left(\rho_t^{(1)} - \rho_t^*\right)^2}{2\xi} + \xi K\left(1 + D_{\boldsymbol{\beta}}^2\|\boldsymbol{\Lambda}\|_2^{2c-1}\right).
$$

Using that $\left(\rho_t^{(1)} - \rho_t^*\right)^2 \le 1$ concludes the proof. $\qquad\square$

**Lemma C.7.** *The gradient estimator* $\tilde{\boldsymbol{g}}_{\boldsymbol{\theta},t,i}$ *satisfies* $\mathbb{E}_{t,i}\left[\tilde{\boldsymbol{g}}_{\boldsymbol{\theta},t,i}\right] = \boldsymbol{g}_{\boldsymbol{\theta},t,i}$ *and* $\mathbb{E}_{t,i}\left[\|\tilde{\boldsymbol{g}}_{\boldsymbol{\theta},t,i}\|_2^2\right] \leq$ $4D_{\boldsymbol{\varphi}}^2 D_{\boldsymbol{\beta}}^2\|\boldsymbol{\Lambda}\|_2^{2c-1}$. *Furthermore, for any* $\boldsymbol{\theta}_t^*$ *with* $\|\boldsymbol{\theta}_t^*\|_2 \leq D_{\boldsymbol{\theta}}$, *the iterates* $\boldsymbol{\theta}_t^{(i)}$ *satisfy*

$$\mathbb{E}\left[\sum_{i=1}^{K}\left\langle \boldsymbol{\theta}_t^{(i)} - \boldsymbol{\theta}_t^*, \boldsymbol{g}_{\boldsymbol{\theta},t,i}\right\rangle\right] \leq \frac{2D_{\boldsymbol{\theta}}^2}{\eta} + 2\eta K D_{\boldsymbol{\varphi}}^2 D_{\boldsymbol{\beta}}^2\|\boldsymbol{\Lambda}\|_2^{2c-1}. \tag{29}$$

*Proof.* Since $\boldsymbol{\beta}_t, \pi_t, \rho_t^i$ and $\boldsymbol{\theta}_t^i$ are $\mathcal{F}_{t,i-1}$-measurable, we obtain

$$\begin{aligned}
\mathbb{E}_{t,i}\left[\tilde{\boldsymbol{g}}_{\boldsymbol{\theta},t,i}\right] &= \mathbb{E}_{t,i}\left[\boldsymbol{\varphi}_{t,i}'\left\langle\boldsymbol{\varphi}_{t,i},\boldsymbol{\Lambda}^{c-1}\boldsymbol{\beta}_t\right\rangle - \boldsymbol{\varphi}_{t,i}\left\langle\boldsymbol{\varphi}_{t,i},\boldsymbol{\Lambda}^{c-1}\boldsymbol{\beta}_t\right\rangle\right] \\
&= \boldsymbol{\Phi}^{\mathsf{T}}\mathbb{E}_{t,i}\left[\boldsymbol{e}_{X_{t,i}',A_{t,i}'}\left\langle\boldsymbol{\varphi}_{t,i},\boldsymbol{\Lambda}^{c-1}\boldsymbol{\beta}_t\right\rangle\right] - \mathbb{E}_{t,i}\left[\boldsymbol{\varphi}_{t,i}\boldsymbol{\varphi}_{t,i}^{\mathsf{T}}\right]\boldsymbol{\Lambda}^{c-1}\boldsymbol{\beta}_t \\
&= \boldsymbol{\Phi}^{\mathsf{T}}\mathbb{E}_{t,i}\left[\left[\pi_t \circ p(\cdot|X_t, A_t)\right]\left\langle\boldsymbol{\varphi}_{t,i},\boldsymbol{\Lambda}^{c-1}\boldsymbol{\beta}_t\right\rangle\right] - \boldsymbol{\Lambda}^c\boldsymbol{\beta}_t \\
&= \boldsymbol{\Phi}[\pi_t \circ \boldsymbol{\Psi}^{\mathsf{T}}\mathbb{E}_{t,i}\left[\boldsymbol{\varphi}_{t,i}\boldsymbol{\varphi}_{t,i}^{\mathsf{T}}\right]\boldsymbol{\Lambda}^{c-1}\boldsymbol{\beta}_t] - \boldsymbol{\Lambda}^c\boldsymbol{\beta}_t \\
&= \boldsymbol{\Phi}[\pi_t \circ \boldsymbol{\Psi}^{\mathsf{T}}\boldsymbol{\Lambda}^c\boldsymbol{\beta}_t] - \boldsymbol{\Lambda}^c\boldsymbol{\beta}_t \\
&= \boldsymbol{\Phi}^{\mathsf{T}}\boldsymbol{\mu}_t - \boldsymbol{\Lambda}^c\boldsymbol{\beta}_t = \boldsymbol{g}_{\boldsymbol{\theta},t}.
\end{aligned}$$

Next, we consider the squared gradient norm and bound it via elementary manipulations as follows:

$$\begin{aligned}
\mathbb{E}_{t,i}\left[\left\|\tilde{\boldsymbol{g}}_{\boldsymbol{\theta},t,i}\right\|_2^2\right] &= \mathbb{E}_{t,i}\left[\left\|\boldsymbol{\varphi}_{t,i}'\left\langle\boldsymbol{\varphi}_{t,i},\boldsymbol{\Lambda}^{c-1}\boldsymbol{\beta}_t\right\rangle - \boldsymbol{\varphi}_{t,i}\left\langle\boldsymbol{\varphi}_{t,i},\boldsymbol{\Lambda}^{c-1}\boldsymbol{\beta}_t\right\rangle\right\|_2^2\right] \\
&\leq 2\mathbb{E}_{t,i}\left[\left\|\boldsymbol{\varphi}_{t,i}'\left\langle\boldsymbol{\varphi}_{t,i},\boldsymbol{\Lambda}^{c-1}\boldsymbol{\beta}_t\right\rangle\right\|_2^2\right] + 2\mathbb{E}_{t,i}\left[\left\|\boldsymbol{\varphi}_{t,i}\left\langle\boldsymbol{\varphi}_{t,i},\boldsymbol{\Lambda}^{c-1}\boldsymbol{\beta}_t\right\rangle\right\|_2^2\right] \\
&= 2\mathbb{E}_{t,i}\left[\boldsymbol{\beta}_t^{\mathsf{T}}\boldsymbol{\Lambda}^{c-1}\boldsymbol{\varphi}_{t,i}\left\|\boldsymbol{\varphi}_{t,i}'\right\|_2^2\boldsymbol{\varphi}_{t,i}^{\mathsf{T}}\boldsymbol{\Lambda}^{c-1}\boldsymbol{\beta}_t\right] + 2\mathbb{E}_{t,i}\left[\boldsymbol{\beta}_t^{\mathsf{T}}\boldsymbol{\Lambda}^{c-1}\boldsymbol{\varphi}_{t,i}\left\|\boldsymbol{\varphi}_{t,i}\right\|_2^2\boldsymbol{\varphi}_{t,i}^{\mathsf{T}}\boldsymbol{\Lambda}^{c-1}\boldsymbol{\beta}_t\right] \\
&\leq 2D_{\boldsymbol{\varphi}}^2\mathbb{E}_{t,i}\left[\boldsymbol{\beta}_t^{\mathsf{T}}\boldsymbol{\Lambda}^{c-1}\boldsymbol{\varphi}_{t,i}\boldsymbol{\varphi}_{t,i}^{\mathsf{T}}\boldsymbol{\Lambda}^{c-1}\boldsymbol{\beta}_t\right] + 2D_{\boldsymbol{\varphi}}^2\mathbb{E}_{t,i}\left[\boldsymbol{\beta}_t^{\mathsf{T}}\boldsymbol{\Lambda}^{c-1}\boldsymbol{\varphi}_{t,i}\boldsymbol{\varphi}_{t,i}^{\mathsf{T}}\boldsymbol{\Lambda}^{c-1}\boldsymbol{\beta}_t\right] \\
&= 2D_{\boldsymbol{\varphi}}^2\mathbb{E}_{t,i}\left[\boldsymbol{\beta}_t^{\mathsf{T}}\boldsymbol{\Lambda}^{c-1}\boldsymbol{\Lambda}\boldsymbol{\Lambda}^{c-1}\boldsymbol{\beta}_t\right] + 2D_{\boldsymbol{\varphi}}^2\mathbb{E}_{t,i}\left[\boldsymbol{\beta}_t^{\mathsf{T}}\boldsymbol{\Lambda}^{c-1}\boldsymbol{\Lambda}\boldsymbol{\Lambda}^{c-1}\boldsymbol{\beta}_t\right] \\
&\leq 4D_{\boldsymbol{\varphi}}^2\|\boldsymbol{\beta}_t\|_{\boldsymbol{\Lambda}^{2c-1}}^2 \leq 4D_{\boldsymbol{\varphi}}^2 D_{\boldsymbol{\beta}}^2\|\boldsymbol{\Lambda}\|_2^{2c-1}.
\end{aligned}$$

Having verified these conditions, we appeal to the online gradient descent analysis of Lemma D.1 to show the bound

$$\mathbb{E}\left[\sum_{i=1}^{K}\left\langle\boldsymbol{\theta}_t^{(i)} - \boldsymbol{\theta}_t^*, \boldsymbol{g}_{\boldsymbol{\theta},t}\right\rangle\right] \leq \frac{\left\|\boldsymbol{\theta}_t^{(1)} - \boldsymbol{\theta}_t^*\right\|_2^2}{2\eta} + 2\eta K D_{\boldsymbol{\varphi}}^2 D_{\boldsymbol{\beta}}^2\|\boldsymbol{\Lambda}\|_2^{2c-1}.$$

We then use that $\left\|\boldsymbol{\theta}_t^* - \boldsymbol{\theta}_t^{(1)}\right\|_2 \leq 2D_{\boldsymbol{\theta}}$ for $\boldsymbol{\theta}_t^*, \boldsymbol{\theta}_t^{(1)} \in \mathbb{B}(D_{\boldsymbol{\theta}})$, thus concluding the proof. $\qquad\square$

## D  Auxiliary Lemmas

The following is a standard result in convex optimization proved here for the sake of completeness—we refer to Nemirovski & Yudin [25], Zinkevich [40], Orabona [28] for more details and comments on the history of this result.

**Lemma D.1** (Online Stochastic Gradient Descent). *Given $y_1 \in \mathbb{B}(D_y)$ and $\eta > 0$, define the sequences $y_2, \cdots, y_{n+1}$ and $h_1, \cdots, h_n$ such that for $k = 1, \cdots, n$,*

$$y_{k+1} = \Pi_{\mathbb{B}(D_y)} \left( y_k + \eta \widehat{h}_k \right),$$

*and $\widehat{h}_k$ satisfies $\mathbb{E}\left[ \widehat{h}_k \, | \mathcal{F}_{k-1} \right] = h_k$ and $\mathbb{E}\left[ \left\| \widehat{h}_k \right\|_2^2 | \mathcal{F}_{k-1} \right] \leq G^2$. Then, for $y^* \in \mathbb{B}(D_y)$:*

$$\mathbb{E}\left[ \sum_{k=1}^{n} \langle y^* - y_k, h_k \rangle \right] \leq \frac{\|y_1 - y^*\|_2^2}{2\eta} + \frac{\eta n G^2}{2}.$$

*Proof.* We start by studying the following term:

$$
\begin{aligned}
\|y_{k+1} - y^*\|_2^2 &= \left\| \Pi_{\mathbb{B}(D_y)}(y_k + \eta \widehat{h}_k) - y^* \right\|_2^2 \\
&\leq \left\| y_k + \eta \widehat{h}_k - y^* \right\|_2^2 \\
&= \|y_k - y^*\|_2^2 - 2\eta \left\langle y^* - y_k, \widehat{h}_k \right\rangle + \eta^2 \left\| \widehat{h}_k \right\|_2^2.
\end{aligned}
$$

The inequality is due to the fact that the projection operator is a non-expansion with respect to the Euclidean norm. Since $\mathbb{E}\left[ \widehat{h}_k \, | \mathcal{F}_{k-1} \right] = h_k$, we can rearrange the above equation and take a conditional expectation to obtain

$$
\begin{aligned}
\langle y^* - y_k, h_k \rangle &\leq \frac{\|y_k - y^*\|_2^2 - \mathbb{E}\left[ \|y_{k+1} - y^*\|_2^2 | \mathcal{F}_{k-1} \right]}{2\eta} + \frac{\eta}{2} \mathbb{E}\left[ \left\| \widehat{h}_k \right\|_2^2 | \mathcal{F}_{k-1} \right] \\
&\leq \frac{\|y_k - y^*\|_2^2 - \mathbb{E}\left[ \|y_{k+1} - y^*\|_2^2 | \mathcal{F}_{k-1} \right]}{2\eta} + \frac{\eta G^2}{2},
\end{aligned}
$$

where the last inequality is from $\mathbb{E}\left[ \left\| \widehat{h}_k \right\|_2^2 | \mathcal{F}_{k-1} \right] \leq G^2$. Finally, taking a sum over $k = 1, \cdots, n$, taking a marginal expectation, evaluating the resulting telescoping sum and upper-bounding negative terms by zero we obtain the desired result as

$$
\begin{aligned}
\mathbb{E}\left[ \sum_{k=1}^{n} \left\langle y^* - y_k, \hat{h}_k \right\rangle \right] &\leq \frac{\|y_1 - y^*\|_2^2 - \mathbb{E}\left[ \|y_{n+1} - y^*\|_2^2 \right]}{2\eta} + \frac{\eta}{2} \sum_{k=1}^{n} G^2 \\
&\leq \frac{\|y_1 - y^*\|_2^2}{2\eta} + \frac{\eta n G^2}{2}.
\end{aligned}
$$

$\square$

The next result is a similar regret analysis for mirror descent with the relative entropy as its distance generating function. Once again, this result is standard, and we refer the interested reader to Nemirovski & Yudin [25], Cesa-Bianchi & Lugosi [7], Orabona [28] for more details. For the analysis, we recall that $\mathcal{D}$ denotes the relative entropy (or Kullback–Leibler divergence), defined for any $p, q \in \Delta_{\mathcal{A}}$ as $\mathcal{D}(p\|q) = \sum_a p(a) \log \frac{p(a)}{q(a)}$, and that, for any two policies $\pi, \pi'$, we define the conditional entropy[2] $\mathcal{H}(\pi\|\pi') \doteq \sum_{x \in \mathcal{X}} \nu^\pi(x) \mathcal{D}(\pi(\cdot|x)\|\pi'(\cdot|x))$.

---

[2]Technically speaking, this quantity is the conditional entropy between the occupancy measures $\mu^\pi$ and $\mu^{\pi'}$. We will continue to use this relatively imprecise terminology to keep our notation light, and we refer to Neu et al. [27] and Bas-Serrano et al. [2] for more details.

**Lemma D.2** (Mirror Descent). *Let $q_t, \ldots, q_T$ be a sequence of functions from $\mathcal{X} \times \mathcal{A}$ to $\mathbb{R}$ so that $\|q_t\|_\infty \leq D_q$ for $t = 1, \ldots, T$. Given an initial policy $\pi_1$ and a learning rate $\alpha > 0$, define the sequence of policies $\pi_2, \ldots, \pi_{T+1}$ such that, for $t = 1, \ldots, T$:*

$$\pi_{t+1}(a|x) \propto \pi_t e^{\alpha q_t(x,a)}.$$

*Then, for any comparator policy $\pi^*$:*

$$\sum_{t=1}^{T} \sum_{x \in \mathcal{X}} \nu^{\pi^*}(x) \langle \pi^*(\cdot|x) - \pi_t(\cdot|x), q_t(x, \cdot) \rangle \leq \frac{\mathcal{H}(\pi^* \| \pi_1)}{\alpha} + \frac{\alpha T D_q^2}{2}.$$

*Proof.* We begin by studying the relative entropy between $\pi^*(\cdot|x)$ and iterates $\pi_t(\cdot|x), \pi_{t+1}(\cdot|x)$ for any $x \in \mathcal{X}$:

$$
\begin{aligned}
\mathcal{D}\left(\pi^*(\cdot|x) \| \pi_{t+1}(\cdot|x)\right) &= \mathcal{D}\left(\pi^*(\cdot|x) \| \pi_t(\cdot|x)\right) - \sum_{a \in \mathcal{A}} \pi^*(a|x) \log \frac{\pi_{t+1}(a|x)}{\pi_t(a|x)} \\
&= \mathcal{D}\left(\pi^*(\cdot|x) \| \pi_t(\cdot|x)\right) - \sum_{a \in \mathcal{A}} \pi^*(a|x) \log \frac{e^{\alpha q_t(x,a)}}{\sum_{a' \in \mathcal{A}} \pi_t(a'|x) e^{\alpha q_t(x,a')}} \\
&= \mathcal{D}\left(\pi^*(\cdot|x) \| \pi_t(\cdot|x)\right) - \alpha \langle \pi^*(\cdot|x), q_t(x, \cdot) \rangle + \log \sum_{a \in \mathcal{A}} \pi_t(a|x) e^{\alpha q_t(x,a)} \\
&= \mathcal{D}\left(\pi^*(\cdot|x) \| \pi_t(\cdot|x)\right) - \alpha \langle \pi^*(\cdot|x) - \pi_t(\cdot|x), q_t(x, \cdot) \rangle \\
&\quad + \log \sum_{a \in \mathcal{A}} \pi_t(a|x) e^{\alpha q_t(x,a)} - \alpha \sum_{a \in \mathcal{A}} \pi_t(a|x) q_t(x, a) \\
&\leq \mathcal{D}\left(\pi^*(\cdot|x) \| \pi_t(\cdot|x)\right) - \alpha \langle \pi^*(\cdot|x) - \pi_t(\cdot|x), q_t(x, \cdot) \rangle + \frac{\alpha^2 \|q_t(x, \cdot)\|_\infty^2}{2}
\end{aligned}
$$

where the last inequality follows from Hoeffding's lemma (cf. Lemma A.1 in [7]). Next, we rearrange the above equation, sum over $t = 1, \cdots, T$, evaluate the resulting telescoping sum and upper-bound negative terms by zero to obtain

$$\sum_{t=1}^{T} \langle \pi^*(\cdot|x) - \pi_t(\cdot|x), q_t(x, \cdot) \rangle \leq \frac{\mathcal{D}\left(\pi^*(\cdot|x) \| \pi_1(\cdot|x)\right)}{\alpha} + \frac{\alpha \|q_t(x, \cdot)\|_\infty^2}{2}.$$

Finally, using that $\|q_t\|_\infty \leq D_q$ and taking an expectation with respect to $x \sim \nu^{\pi^*}$ concludes the proof. $\qquad\square$

 **E   Detailed Computations for Comparing Coverage Ratios**

705  For ease of comparison, we just consider discounted linear MDPs (Definition 2.1).

706  **Definition E.1.** Recall the following definitions of coverage ratio given by different authors in the
707  offline RL literature:

708      1. $C_{\varphi,c}(\pi^*; \pi_B) = \mathbb{E}_{X,A\sim\mu^*}\left[\varphi(X,A)\right]^\top \mathbf{\Lambda}^{-2c}\mathbb{E}_{X,A\sim\mu^*}\left[\varphi(X,A)\right]$          (Ours)

709      2. $C^\diamond(\pi^*; \pi_B) = \mathbb{E}_{X,A\sim\mu^*}\left[\varphi(X,A)^\intercal\mathbf{\Lambda}^{-1}\varphi(X,A)\right]$          (e.g., Jin et al. [14])

710      3. $C^\dagger(\pi^*; \pi_B) = \sup_{y\in\mathbb{R}^d} \frac{y^\intercal\mathbb{E}_{X,A\sim\mu^*}\left[\varphi(X,A)\varphi(X,A)^\intercal\right]y}{y^\intercal\mathbb{E}_{X,A\sim\mu_B}\left[\varphi(X,A)\varphi(X,A)^\intercal\right]y}$          (e.g., Uehara & Sun [32])

711      4. $C_{\mathcal{F},\pi}(\pi^*; \pi_B) = \max_{f\in\mathcal{F}} \frac{\|f-\mathcal{T}^\pi f\|_{\mu^*}^2}{\|f-\mathcal{T}^\pi f\|_{\mu_B}^2}$          (e.g., Xie et al. [36]),

712  where $c \in \{1,2\}$, $\mathbf{\Lambda} = \mathbb{E}_{X,A\sim\mu_B}\left[\varphi(X,A)\varphi(X,A)^\intercal\right]$ (assumed invertible), $\mathcal{F} \subseteq \mathbb{R}^{\mathcal{X}\times\mathcal{A}}$, and
713  $\mathcal{T}^\pi : \mathcal{F} \to \mathbb{R}$ defined as $(\mathcal{T}^\pi f)(x,a) = r(x,a) + \gamma\sum_{x',a'} p(x'|x,a)\pi(a'|x')f(x',a')$ is the
714  Bellman operator associated to policy $\pi$.

715  The following is a generalization of the low-variance property from Section 6.

716  **Proposition E.2.** *Let* $\mathbb{V}\left[Z\right] = \mathbb{E}[\|Z - \mathbb{E}\left[Z\right]\|^2]$ *for a random vector $Z$. Then*

$$C_{\varphi,c}(\pi^*; \pi_B) = \mathbb{E}_{X,A\sim\mu^*}\left[\varphi(X,A)^\intercal\mathbf{\Lambda}^{-2c}\varphi(X,A)\right] - \mathbb{V}_{X,A\sim\mu^*}\left[\mathbf{\Lambda}^{-c}\varphi(X,A)\right].$$

717  *Proof.* We just rewrite $C_{\varphi,c}$ from Definition E.1 as

$$C_{\varphi,c}(\pi^*; \pi_B) = \left\|\mathbb{E}_{X,A\sim\mu^*}\left[\mathbf{\Lambda}^{-c}\varphi(X,A)\right]\right\|^2.$$

718  The result follows from the elementary property of variance $\mathbb{V}\left[Z\right] = \mathbb{E}[\|Z\|^2] - \|\mathbb{E}[Z]\|^2$.          $\square$

719  **Proposition E.3.** $C^\dagger(\pi^*; \pi_B) \leq C^\diamond(\pi^*; \pi_B) \leq dC^\dagger(\pi^*; \pi_B)$.

720  *Proof.* Let $(X^*, A^*) \sim \mu^*$ and $\mathbf{M} = \mathbb{E}\left[\varphi(X^*, A^*)\varphi(X^*, A^*)\right]$. First, we rewrite $C^\diamond$ as

$$\begin{aligned}
C^\diamond(\pi^*; \pi_B) &= \mathbb{E}\left[\varphi(X^*, A^*)^\intercal\mathbf{\Lambda}^{-1}\varphi(X^*, A^*)\right] \\
&= \mathbb{E}\left[\mathrm{Tr}(\varphi(X^*, A^*)^\intercal\mathbf{\Lambda}^{-1}\varphi(X^*, A^*))\right] \\
&= \mathbb{E}\left[\mathrm{Tr}(\varphi(X^*, A^*)\varphi(X^*, A^*)^\intercal\mathbf{\Lambda}^{-1})\right] &(30) \\
&= \mathrm{Tr}(\mathbf{M}\mathbf{\Lambda}^{-1}) &(31) \\
&= \mathrm{Tr}(\mathbf{\Lambda}^{-1/2}\mathbf{M}\mathbf{\Lambda}^{-1/2}), &(32)
\end{aligned}$$

721  where we have used the cyclic property of the trace (twice) and linearity of trace and expectation.
722  Note that, since $\mathbf{\Lambda}$ is positive definite, it admits a unique positive definite matrix $\mathbf{\Lambda}^{1/2}$ such that
723  $\mathbf{\Lambda} = \mathbf{\Lambda}^{1/2}\mathbf{\Lambda}^{1/2}$. We rewrite $C^\dagger$ in a similar fashion

$$\begin{aligned}
C^\dagger(\pi^*; \pi_B) &= \sup_{y\in\mathbb{R}^d} \frac{y^\intercal\mathbf{M}y}{y^\intercal\mathbf{\Lambda}y} \\
&= \sup_{z\in\mathbb{R}^d} \frac{z^\intercal\mathbf{\Lambda}^{-1/2}\mathbf{M}\mathbf{\Lambda}^{-1/2}z}{z^\intercal z} &(33) \\
&= \lambda_{\max}(\mathbf{\Lambda}^{-1/2}\mathbf{M}\mathbf{\Lambda}^{-1/2}), &(34)
\end{aligned}$$

724  where $\lambda_{\max}$ denotes the maximum eigenvalue of a matrix. We have used the fact that both $\mathbf{M}$ and
725  $\mathbf{\Lambda}$ are positive definite and the min-max theorem. Since the quadratic form $\mathbf{\Lambda}^{-1/2}\mathbf{M}\mathbf{\Lambda}^{-1/2}$ is also
726  positive definite, and the trace is the sum of the (positive) eigenvalues, we get the desired result.          $\square$

727  **Proposition E.4** (cf. the proof of Theorem 3.2 from [36]). *Let* $\mathcal{F} = \{f_{\boldsymbol{\theta}} : (x,a) \mapsto \langle\varphi(x,a), \boldsymbol{\theta}\rangle | \boldsymbol{\theta} \in$
728  $\Theta \subseteq \mathbb{R}^d\}$ *where $\varphi$ is the feature map of the linear MDP. Then*

$$C_{\mathcal{F},\pi}(\pi^*; \pi_B) \leq C^\dagger(\pi^*; \pi_B),$$

729  *with equality if* $\Theta = \mathbb{R}^d$.

*Proof.* Fix any policy $\pi$ and let $\mathcal{T} = \mathcal{T}^{\pi}$. By linear Bellman completeness of linear MDPs [13], $\mathcal{T}f \in \mathcal{F}$ for any $f \in \mathcal{F}$. For $f_{\boldsymbol{\theta}} : (x,a) \mapsto \langle \boldsymbol{\varphi}(x,a), \boldsymbol{\theta} \rangle$, let $\mathcal{T}\boldsymbol{\theta} \in \Theta$ be defined so that $\mathcal{T}f_{\boldsymbol{\theta}} : (x,a) \mapsto \langle \boldsymbol{\varphi}(x,a), \mathcal{T}\boldsymbol{\theta} \rangle$. Then

$$C_{\mathcal{F},\pi}(\pi^*; \pi_B) = \max_{f \in \mathcal{F}} \frac{\mathbb{E}_{X,A\sim\mu^*}\left[(f(X,A) - \mathcal{T}f(X,A))^2\right]}{\mathbb{E}_{X,A\sim\mu_B}\left[(f(X,A) - \mathcal{T}f(X,A))^2\right]} \tag{35}$$

$$\leq \max_{\boldsymbol{\theta}\in\mathbb{R}^d} \frac{\mathbb{E}_{X,A\sim\mu^*}\left[\langle\boldsymbol{\varphi}(X,A), \boldsymbol{\theta} - \mathcal{T}\boldsymbol{\theta}\rangle^2\right]}{\mathbb{E}_{X,A\sim\mu_B}\left[\langle\boldsymbol{\varphi}(X,A), \boldsymbol{\theta} - \mathcal{T}\boldsymbol{\theta}\rangle^2\right]} \tag{36}$$

$$= \max_{y\in\mathbb{R}^d} \frac{\mathbb{E}_{X,A\sim\mu^*}\left[\langle\boldsymbol{\varphi}(X,A), y\rangle^2\right]}{\mathbb{E}_{X,A\sim\mu_B}\left[\langle\boldsymbol{\varphi}(X,A), y\rangle^2\right]} \tag{37}$$

$$= \max_{y\in\mathbb{R}^d} \frac{y^{\mathsf{T}}\mathbb{E}_{X,A\sim\mu^*}\left[\boldsymbol{\varphi}(X,A)\boldsymbol{\varphi}(X,A)^{\mathsf{T}}\right]y}{y^{\mathsf{T}}\mathbb{E}_{X,A\sim\mu_B}\left[\boldsymbol{\varphi}(X,A)\boldsymbol{\varphi}(X,A)^{\mathsf{T}}\right]y}, \tag{38}$$

where the inequality in Equation (36) holds with equality if $\Theta = \mathbb{R}^d$. $\qquad\square$