# OpenReview forum: "Offline Primal-Dual Reinforcement Learning for Linear MDPs"
_NeurIPS.cc/2023/Conference — Submitted to NeurIPS 2023_

### Official Review · Reviewer_vy6G · 2023-06-11

**Soundness:** 3 good
**Presentation:** 3 good
**Contribution:** 3 good
**Rating:** 6
**Confidence:** 3

**Summary:**

This paper studied offline RL in linear MDP setting, where the transition and reward have low-rank structures and the feature $\phi$ is known. The authors formulated the problem in a primal-dual way and proposed a gradient-based algorithm. They provided convergence guarantees, which only requires coverage over optimal policy.

**Strengths:**

The paper writing is clear and easy to follow.

The discussion and comparison with previous works is very detailed.

The algorithm is computationally efficient. The algorithm design has some interesting points, especially the reparameterization design to avoid knowlegde of $\Lambda^{-1}$ and updates for variables $v$ and $u$.

The coverage assumption seems weaker than previous literatures.

**Weaknesses:**

I didn't see too much technical novelty in the method and proof.

The setting is linear MDP, which is kind of restrictive.

Convergance rate is kind of far away from optimal.

**Questions:**

Given the knowledge of $\Lambda$, why not consider the reparameterization $\beta = \lambda$, i.e., set $c=0$, but consider $c=1/2$? It seems to me the RHS of Eq. in Line 141 can be estimated unbiasedly?

Line 154 claimed that $c = 1/2$ results in a tighter bound, however, according to Theorem 3.2, and Corollary 3.3, the larger $c$ is, the lower dependence on $d$ the sample complexity will have. So how the "tighter bound" is reflected? It seems contradictory.

**Limitations:**

N.A.

---

> ### Author Rebuttal · Authors · 2023-08-09
>
> We appreciate your feedback on our work. In response, we kindly highlight that the major technical novelty of our work is a reparametrization trick with which we adapt the conventional LP formulation to a novel framework for offline learning via primal-dual optimization. This allows us to derive low-variance gradient estimators that can be used in a stochastic optimization scheme using only samples from the behavior policy. Furthermore, our proof technique uses arguments from Neu et. al (2022), but tailored to this novel offline learning framework. Along the way, we also utilize a notion of coverage - coverage in a single direction in feature space, which at least for $c=1/2$ is provably smaller than other standard coverage conditions. As we mention in our general response, these findings are novel in LP based methods for offline RL, and are quite significant in our opinion.
>
> While we strongly believe that our work presents novel ideas which are worth sharing, we agree that there is room for improvement.
>
> Regarding our choice of $c$, yes with $c=0$ we can still estimate the gradients unbiasedly provided we know $\Lambda$. However, the variance of the resulting gradient estimators is prohibitively large, which leads to a rather weak regret bound that scales with the inverse of the smallest eigenvalue of Lambda. In other words, one has to assume a form of uniform coverage condition in the feature space, which is much much stronger than the conditions that our estimators require. Though other choices may lead to interesting results, our guarantees are to be understood for $c \in {1/2, 1}$. That said, we leave the question of looking into other values for future work, and will mention this in the final version.
>
> Finally, in line 154 of our work, by "tighter bound", we actually refer to the dependence on the coverage ratio. For example, in the tabular setting with $\Phi$ as the identity matrix, the coverage ratio when $c=1/2$ is $C_{\phi,1/2}(\pi^*;\pi_{B})=\sum_{x,a}\frac{\mu(x,a)^2}{\mu_B(x,a)}$
> while for $c=1$ it becomes $C_{\phi,1}(\pi^*;\pi_{B})=\sum_{x,a}\frac{\mu^{*}(x,a)^{2}}{\mu_{B}(x,a)^{2}}$ acquiring a factor of $1/\mu_{B}(x,a)$ in the sum which can be large. Thanks for noticing this, we are going to make our statement more clear in the final draft.

---

> > ### Comment · Reviewer_vy6G · 2023-08-14
> > **Acknowledged**
> >
> > Thanks for the response. I think my questions are addressed and I will take the response into consideration in the final decision period.

---

> > > ### Author Response · Authors · 2023-08-17
> > >
> > > Thank you, we are glad we could be of help and remain available in case you have more questions.

---

### Official Review · Reviewer_7fb2 · 2023-07-12

**Soundness:** 2 fair
**Presentation:** 3 good
**Contribution:** 3 good
**Rating:** 6
**Confidence:** 3

**Summary:**

This paper proposed an primal-dual framework for offline reinforcement learning in linear MDP Contrary to the more common case of finite horizon, they considered the case of infinite horizon with discounted reward. They reduced the problem of offline reinforcement learning to a problem about solving the saddle-point of a Lagrange form. They designed an algorithm which uses stochastic gradient-based optimization to solve the saddle point. They provide a sample complexity of O(\eps^-4) for both cases of discounted MDP and averaged-reward MDP, and their algorithm is also computational efficient.

To summarize, the formulation of offline RL into a linear programming problem is very interesting. The proof seems very solid, and I like the comparison for the concentrability constant in the last discussion section. The comparison for the constant C is thorough and very good.

However, I still have some questions about some details in the main text.

**Strengths:**

1. The formulation of offline reinforcement learning to a linear programming problem is very good.
2. The algorithm is clearly motivated by solving the saddle points of a Lagrange form. The algorithm itself is simple and computationally efficient, with a guaranteed sample complexity for both discounted MDP and averaged-reward MDP.
3. They proposed a new concentrability constant C and compare it to other constants appearing in other literatures about offline RL. I think the understanding of the relationship of these concentrability constant is basically correct and very clearly expressed.
4. The proof seems very solid and the result in averaged-reward case is new.

**Weaknesses:**

1. I have some question about your comparison to previous results. Your main references are Cheng et al and Xie et al.

1.1 For Xie et al, the Theorem 3.2 in https://proceedings.neurips.cc/paper_files/paper/2021/file/34f98c7c5d7063181da890ea8d25265a-Paper.pdf implies that their sample complexity is O(1/\eps^2) when applied in linear function approximation. This result is based on assumption3 in their paper. This assumption naturally holds in your paper since you consider linear MDP and they consider the case of 'linear function approximation' (for their difference, see point 2). So it is natural for you to compare your sample complexity to this result, not the O(1/\eps^5) one. [notice that, their algorithm in section 3 is computationally inefficient]

1.2 In Theorem 4.1 in Xie's paper, their sample complexity is O(1/\eps^5) when applied on general function approximation, and O(1/\eps^3) when reduced to linear function approximation case (see paragraph 'Dependence oon T'). Again, their assumption for linear function approximation holds in your case. **This algorithm, however, is computationally efficient.** So you should also compare with this alg with  O(1/\eps^3) sample complexity.

1.3 In Cheng's paper, in theorem 5, their sample complexity seems to be O(1/\eps^3), not O(1/\eps^5). I wonder how you derive their sample complexity in Table one.

1.4 I am not sure how you get the O(n^{7/5}) computational complexity for Xie's paper. Could you derive it in more detail?

2. I think in many places in your paper, you confuse the two terms: linear MDP and linear function approximation. Your case is called linear MDP instead of linear function approximation, so I suggest you changing the wrong terms. For reference, both Xie's paper and Cheng's paper consider the 'linear function approximation' case.

**Questions:**

1. In Line 82 and 87, there are typos in the definition of rho and state occupancy measure? There should be a power of gamma in the sum?

2. In line 174-176, you calculate the computational complexity of your algorithm. Can you show this in more detail? I am not sure why each gradient update needs a constant number of elementary vector operation. What is your definition for elementary vector operation?

3. In section 3 and 4, you use c \in {1/2, 1} to state all your results. I wonder whether c can only take these two values? If so, then why cannot c take other values (for example, 2 or 3 or 1/4)? If not, then how will the algorithm behave and how will the sample complexity be when we take c to be other values? If c can take multiple values besides 1/2 and 1, how can you tune c to get the best performance?

4. In theorem 3.2, the sample complexity depend on constant c by D_\phi^{8c} * d^{-2c}, right? Does the constant hidden by the big O depend on c or nor? If not, then suppose D_\phi^{8} / d^2 < 1, if we take c to be large or even converge to the infinify, then the term inside the big O will approach zero. How can that happen? (Let's suppose D)\beta is a constant and does not depend on c). For example, in the most common setting, c = 1 and d > 1, then when we take c \to \infty, the term in the bracket wiil be zero. If the term hidden by the big O depends on constant c, then you should point it out in the paper.

5. In Line 198, you say D_\beta = c * C for some positive universal constant c. Is this c multiplied before the concentrabilty coefficient the same as that c in definition 3.1? If not, then you should use another notation.

6. In line 315, you claim C_{\phi,1} = 1 + chi-square divergence. However, C_{\phi,1} is \sum (\mu^*)^2 / (\mu_B)^2 and the chi-square divergence should be \sum (\mu^*)^2 / (\mu_B), right? So they should not be equal. For the definition of chi-square divergence, see https://en.wikipedia.org/wiki/F-divergence.

7. In line 77, you claim the state space is finite but potentially very large; but in Line 338, you claim that 'not to assume the state space to be finite'. From the proof, I think the latter one is correct, and the first one is just for deriving some equations like equation 1 and 2. So I think you should write somewhere in section 3 that you relax this setting of finite state space and the derivation there also holds for infinite state space but finite action space.

**Limitations:**

/

---

> ### Author Rebuttal · Authors · 2023-08-09
>
> We thank you for the very careful reading of our paper.
>
> Indeed, the works you reference provide many different algorithm variations and sample complexity bounds. Our comparison could definitely be improved a bit. We plan to fix this in the final draft of our paper, by adding to the appendix a more detailed explanation.
>
> Most of the algorithms provided by Xie et al make use of computational oracles with an unspecified complexity, which makes it hard to categorize these methods as computationally efficient. In particular, in their Section 4 they present a "practical algorithm" and prove a bound of $O(\varepsilon^{-5})$ for it. However, this algorithm still makes use of an optimization oracle to solve an inherently nonconvex problem (which may be hard in general). They also claim that when reduced to the linear MDP setting the algorithm achieves $O(\varepsilon^{-3})$. We find this claim believable, but have to point out that no proof is offered in the paper. Working out the details may be possible, but we can see several nontrivial challenges. For instance, the concrete implementation of their algorithm in Appendix D does not match the specifications needed in the analysis (e.g., it uses an unbounded function class which prevents using a covering-number argument in the general analysis). In any case, we will point out their improved rate in the final version of our paper, but also mention these caveats.
>
> Regarding Cheng et al, a similar argument can be made. They also provide an oracle-based algorithm with strong theoretical guarantees and a practical implementation for which they make (or prove) no explicit claims.
>
> Thank you for pointing out these details, which made us dive deeper into these related works. We also plan to update our table accordingly.
>
> Regarding the computational complexity for Xie et al (Appendix D). They need to compute the matrices B and C, which depend on the current policy, and therefore need to be recomputed at each iteration. Computing these matrices requires iterating over the whole dataset, and thus $O(n)$ time at each iteration. Finally, the total number of iterations is set to $T=O(n^{2/5})$, giving a total of $O(n^{7/5})$ run-time cost.
>
> We thank you for the question. We plan to add a paragraph to the appendix about these calculations, to improve the clarity of our paper.
>
> Regarding Linear MDP and linear function approximation. "Linear MDP" refers to properties of the problem, "function approximation" of the solution. They are often used interchangeably in the literature because a linear-MDP assumption allows for sample-efficient learning using linear functions to represent action-value functions. In the setting we consider, both expressions are appropriate, and there should not be any confusion: the linear-MDP assumption is always assumed to hold *and* our action-value functions are always linear.
>
> ## Questions
> 1. Yes, you are right. Thank you for noticing this!
> 2. Thank you for bringing this up, we will try to explain this more clearly in the final draft.
>   Our algorithm as presented in the paper makes use of elementary operations between vectors,  such as sums, inner products, entry-wise, matrix-vector products etc. These operations may have a run-time cost proportional to the dimension of the vectors (which in turn is proportional to $d$), however they do not depend on $n$. The algorithm is made of one external loop which performs $T$ iterations, and one inner loop which each time does $K-1$ iterations. Each of the loops uses one sample per iteration and performs a constant number -- with respect to $n$ -- of elementary vector operations, resulting in $O(1)$ run-time iteration complexity with respect to $n$. Therefore, unrolling the inner loop, yields a cost of $O(K)$ per outer iteration. Multiplying that by the number of iterations of the outer loop we obtain a total cost of $O(T K)= O(n)$. To keep our answer simple we only considered the dependence to $n$, ignoring $d$, $|\mathcal{A}|$ and other smaller quantities.
> 3. The complete version of our results (shown in Theorem A.1 and C.1) are stated for a general c, and can give some insight on how the sample complexity changes for different values of c. However, we have not tried to fully optimize this. Rather, we choose to focus on intuitive choices of c. Interestingly, with $c=1/2$, we have shown that time and sample efficient offline learning is feasible in the discounted and average reward settings under a conceptually weaker notion of coverage (please see our general response). On the other hand, with $c=1$, we provided guarantees under similar conditions without knowledge of the feature covariance under the behaviour policy. We are more than convinced that these findings are relevant in their own right and leave further work such as tuning $c$ for future work.
> 4. Yes, the sample complexity in theorem 3.2 depends on $c$ by $D_\phi^{8c}d^{-2c}$ and also on $D_{\beta}$ since we require that $D_{\beta}\geq C_{\phi,c}(\pi*;\pi_{B})$. However, the simplified upper bound presented in the main text is **only valid for $c\in\{1/2,1\}$**. To get the full picture on how different values of $c$ impact the regret, we kindly refer the reviewer to the complete statement of the Theorem in Appendix A, that works for general $c$.
> 5. No, the c in Line 198 is a numerical constant that has noting to do with the parameter c from Def. 3.1. Thank you for pointing this out.
> 6. Yes, that is a typo, thank you for finding it! It is $C_{\phi,1/2}$ that is equal to the chi-square divergence, as you can check using the definition you provided.
> 7. Indeed, it is how you suggest. We consider finite (but potentially very large) state space for simplicity, but all the derivations extend to infinite state spaces. On the other hand, the finite action space is actually needed, for computational reasons, since we employ softmax policies and compute sums over actions in the algorithm. We will explain this better in the final version.

---

> > ### Comment · Area_Chair_VwKC · 2023-08-16
> > **Comparison to related work**
> >
> > Dear authors,
> >
> > I see several reviewers point out concerns related to comparison to related works and also see your responses. We will have to gather everyone's opinion later after they have a chance of reading your rebuttal and each other's review, but just a very quick point regarding your comment:
> >
> > > the concrete implementation of their algorithm in Appendix D does not match the specifications needed in the analysis (e.g., it uses an unbounded function class which prevents using a covering-number argument in the general analysis)
> >
> > You are right the analysis is incomplete, but I think the computational efficiency is still quite obvious? The objective is just maximizing quadratic function with PSD Hessian (_edit:_ up to a minus sign) thus minimizing a convex function. You are totally right that they use an unbounded function class in the computational argument --- but adding (say) a norm constraint on the parameter is quite straightforward? Detailed computational complexity may be difficult to obtain, but broadly speaking I believe polynomial complexity will hold without issues. So at least it should be an entry in your table with very strong sample complexity whereas the detailed computational complexity is less clear (but polynomial).

---

> > > ### Author Response · Authors · 2023-08-17
> > >
> > >
> > > Thank you for the comment. We find your observation totally reasonable. We will make it more clear in the text and the table, that PSPI, when reduced to the linear MDP setting, achieves $O(\varepsilon^{-3})$ and has polynomial computational complexity.

---

> > > > ### Comment · Area_Chair_VwKC · 2023-08-18
> > > >
> > > > Thanks. Just saw your updated table. Related to the above discussion, I'd suggest maybe put "oracle/convex opt" for PSPI in the computational column.

---

> > > > > ### Author Response · Authors · 2023-08-19
> > > > >
> > > > > Thank you for the suggestion. We will incorporate it into the final draft.

---

> > ### Comment · Reviewer_7fb2 · 2023-08-22
> >
> > Thanks the authors for your reply!
> >
> > Given your clarification on the related work, I will raise my score to six.

---

### Official Review · Reviewer_zEPA · 2023-07-24

**Soundness:** 3 good
**Presentation:** 3 good
**Contribution:** 3 good
**Rating:** 5
**Confidence:** 4

**Summary:**

This paper studies offline reinforcement learning (RL) with linear function approximation and partial data coverage. The authors propose a primal-dual optimization method based on the linear programming (LP) formulation of RL. They prove a $O(\epsilon^{-4})$ sample complexity in both discounted setting and average-reward setting.

**Strengths:**

1.	The algorithm proposed in this paper only requires near-minimal dataset coverage assumption, which is important in offline RL.
2.	The paper also considers average-reward offline RL, which is often neglected by literature.
3.	I like the table for comparison to previous work, which makes the presentation more clear (although I think there is some missing important literature, which I will mention in the weakness section).
4.	The proposed algorithm is both computationally and sample efficient.

**Weaknesses:**

1.	The first concern is the ‘linear function approximation’ setting, which is restricted. Actually, the main motivation that this paper studies function approximation beyond tabular settings is large state (or action) spaces in practice. However, in real settings, the linear function approximation assumption hardly ever holds. Even in Table 1, many algorithms in previous work apply to general function approximation, which further makes the setting studied in this paper restricted.
2.	Algorithm 1 in this paper achieves a $O(\epsilon^{-4})$ sample complexity. This is in terms of expectation (as shown in Theorem 3.2) instead of high probability. The previous results that the authors are comparing to are high probability bound (e.g., [1,2]), so it would be more comparable if the authors could also show a $O(\epsilon^{-4})$ sample complexity bound under high probability. Also, since the previous work studies general function approximation while this paper studies only linear function approximation, it is hard to say that a $O(\epsilon^{-4})$ sample complexity bound in linear function approximation setting is better than a $O(\epsilon^{-5})$ bound in general function approximation setting. Moreover, [3] achieves the near-optimal sample complexity $O(\epsilon^{-2})$ with near-identical settings of [1,2]. (So I disagree with the statement that ‘It is very important to notice that no practical algorithm for this setting so far, including ours, can match the minimax optimal sample complexity rate of $O(\epsilon^{-2})$’. Therefore, a $O(\epsilon^{-4})$ in linear function approximation is not that attractive compared to previous work.
3.	The authors use an LP formulation of offline RL. I think it would be better to compare to other work using LP formulation, e.g. [4,5], where [4] is computational and sample efficient under partial data coverage and general function approximation, and [5] achieves near-optimal sample complexity under similar settings.
4.	The authors compare the computational complexity. However, it is not that direct to compare an $O(n)$ complexity in linear settings to a $O(n^{7/5})$ complexity in general settings. If the authors really want to demonstrate that their algorithm has better computational complexity, it would be better to do some simulations in the same environment (even in some toy examples).
5.	Another advantage that the authors claim is that their algorithm could be adapted to average-reward setting. However, neither did the authors emphasize and explain the importance and challenges of average-reward settings, nor discuss why (or whether) previous work could not be adapted to average-reward settings. I suggest the authors discuss this a bit more.

**References**

[1] Xie, T., Cheng, C. A., Jiang, N., Mineiro, P., & Agarwal, A. (2021). Bellman-consistent pessimism for offline reinforcement learning. Advances in neural information processing systems, 34, 6683-6694.

[2] Cheng, C. A., Xie, T., Jiang, N., & Agarwal, A. (2022, June). Adversarially trained actor critic for offline reinforcement learning. In International Conference on Machine Learning (pp. 3852-3878). PMLR.

[3] Zhu, H., Rashidinejad, P., & Jiao, J. (2023). Importance Weighted Actor-Critic for Optimal Conservative Offline Reinforcement Learning. arXiv preprint arXiv:2301.12714.

[4] Zhan, W., Huang, B., Huang, A., Jiang, N., & Lee, J. (2022, June). Offline reinforcement learning with realizability and single-policy concentrability. In Conference on Learning Theory (pp. 2730-2775). PMLR.

[5] Rashidinejad, P., Zhu, H., Yang, K., Russell, S., & Jiao, J. (2022). Optimal conservative offline rl with general function approximation via augmented lagrangian. arXiv preprint arXiv:2211.00716.

**Questions:**

1.	Would it be easy to convert the result from the expectation form to the high-probability version?
2.	Is it easy to modify previous algorithms for the average-reward settings? If not, what is the difficulty?

---

> ### Author Rebuttal · Authors · 2023-08-09
>
> We thank the reviewer for the valuable feedback and for aknowledging many of the strengths of our work.
>
> We agree. Two of the main weaknesses of our work are that it is limited to the linear MDP setting, while many related works consider a more general function approximation setting; and the absence of bounds in high probability. Expanding our approach to work under more general function approximation and proving bounds in high probability are definitely interesting directions for future work.
>
> > Also, since the previous work studies general function approximation while this paper studies only linear function approximation, it is hard to say that a $O(\varepsilon^{-4})$ sample complexity bound in linear function approximation setting is better than a $O(\varepsilon^{-5})$ bound in general function approximation setting.
>
> We now realize that our wording may have been misleading --- apologies for that! Our intention was not to claim that our method is better in every way than previous works. In fact we believe that comparing rates and constants along a variety of settings is an overly reductive view of progress in the area, and that all contributions should be evaluated on grounds of their intellectual value. We elaborated more about this point in our general comment.
>
> We also thank the reviewer for referencing the papers of [Zhan et al 2022][1], [Rashidinejad et al 2022][2] and [Zhu et al 2023][3]. This line of works is very interesting, and indeed  very much related to our paper. We will add a paragraph to the final draft with a detailed comparison, as well as update our table accordingly (see the attached pdf).
>
> [1]: https://proceedings.mlr.press/v178/zhan22a.html
> [2]: https://arxiv.org/abs/2211.00716
> [3]: https://arxiv.org/abs/2301.12714
>
> > 4. The authors compare the computational complexity. However, it is not that direct to compare an $O(n)$ complexity in linear settings to a $O(n^{7/5})$ complexity in general settings. If the authors really want to demonstrate that their algorithm has better computational complexity, it would be better to do some simulations in the same environment (even in some toy examples).
>
> We again apologize if our wording was not very clear. As said before our intention was not to claim the general superiority of our method. However, regarding a comparison with respect to computational efficiency, please note that most of the works we reference are indeed oracle-efficient, but the computational complexity of the oracles is left unspecified. This may require exhaustive search over large discrete function classes in the worst case. In contrast, our methods come with tight, concrete computational complexity guarantees on the number of elementary operations being used.
>
> > 5. Another advantage that the authors claim is that their algorithm could be adapted to average-reward setting. However, neither did the authors emphasize and explain the importance and challenges of average-reward settings, nor discuss why (or whether) previous work could not be adapted to average-reward settings. I suggest the authors discuss this a bit more.
>
> We agree with your assessment here. We will update our draft accordingly. Please refer to our answer to reviewer C22H for more details about this topic.

---

> > ### Comment · Reviewer_zEPA · 2023-08-19
> >
> > Thank the authors for the response as well as the attached table. However, I still have some concerns regarding the contribution of this work.
> >
> > For the attached table, the sample complexity of ALMIS should be $O(1/\epsilon^2)$ instead of $O(1/\epsilon^3)$. Is this a typo or am I missing something? Also, the authors added a column denoting whether the algorithm is oracle-based. However, all algorithms dealing with general function approximation need an optimization oracle (otherwise, all they can do is enumerate all functions in the function class since it is a general function class without any additional assumptions). Therefore, it is unfair (or unnecessary) to discuss whether it is oracle-based when comparing linear function approximation to general function approximation.
> >
> > I agree with the author that the sample complexity should not be the only dimension to compare different algorithms. However, according to the table, it seems the only dimension among multiple ones that the proposed algorithm beats other ones is whether it can be extended to the average reward case. However, considering restrictions in other dimensions (linear MDP, suboptimal sample complexity, no empirical evaluation), it remains unclear whether the current algorithm contributes enough to the community.
> >
> > The authors also mentioned that their bound reflects the correct comparator-dependent quantity, which I agree with. From my understanding, this quantity corresponds to the partial data coverage assumption, which only requires that the offline dataset covers the optimal policy (or the target policy). However, many recent works also assume partial data coverage, so they all use the correct notion. Given that, the coverage notion does not seem to be a significant advantage over previous works.
> >
> > Overall, I appreciate the point of average reward setting and the new method to formulate linear MDP. However, it still has many restrictions.  I believe it would be great work if (some of) these restrictions could be addressed. For the current state of the paper, I slightly lean towards rejection and would keep my score.

---

> > > ### Comment · Area_Chair_VwKC · 2023-08-19
> > >
> > > > many recent works also assume partial data coverage, so they all use the correct notion
> > >
> > > Well not quite. If you read the discussion in Section 6, there are multiple notions of partial coverage (in the sense that we only need to cover the optimal policy or whichever policy we want to compare to) in linear MDPs that are different from each other (in the sense that what counts as "covering a policy" has multiple definitions), and the one in this paper (for the $c=1/2$ result) is arguably the "most correct" one. Not many papers in the literature use this one. This is also discussed in https://arxiv.org/pdf/2203.05804.pdf and a twitter discussion (https://twitter.com/nanjiang_cs/status/1666994607073230848) which I mentioned in another thread.
> > >
> > > That said, the reason that some prior works didn't use this "most correct" notion is sometimes because people didn't realize these different versions and simply went with what previous people did, and in some cases it is relatively easy to adapt the analysis to obtain the better notion of coverage, so the overall situation is a bit chaotic...

---

> > > > ### Comment · Reviewer_zEPA · 2023-08-19
> > > >
> > > > Thanks for your clarification. Now I understand the meaning of the 'correct' notion. In that case, I would be happy to raise my score to 5. I want to clarify that the reason I raised my score is not that I think the proposed algorithm is comparable (in sample complexity or empirical performance) to some previous algorithms, especially in general value function approximation; the reason I think this work is valuable is due to the new method formulating linear MDP and this tight coverage notion appears in the sample complexity bound, which could inspire other works in both linear and general function approximation to rethink about the choice of coverage notion and thus design more practical algorithms.

---

> > > > ### Author Response · Authors · 2023-08-21
> > > > **Response to Area Chair VwKC**
> > > >
> > > > Thank you! We appreciate your help in clarifying this.

---

> > > ### Author Response · Authors · 2023-08-19
> > >
> > > Thank you for the feedback, and for raising your evaluation of our work. Please see further clarification on your comments below.
> > >
> > > Yes, the sample complexity of ALMIS is $O(\epsilon^{-2})$ not $O(\epsilon^{-3})$. Thank you for bringing this to our attention. We will update our table accordingly.
> > >
> > > Regarding your comment on our table, you do make a fair point. However, our focus in the paper is on linear function approximation; a special case of general function approximation. For this reason, we believe that excluding algorithms that were developed for general function approximation would unfair. That said, we would like to emphasize that, the fact that oracle access is common in works focused on general function approximation is of little interest for our comparison. Rather, what we care about is whether these methods are efficient once they are applied to the linear case. When this is not the case, or is not clear from the papers, such methods were categorized as "oracle-based". This does not imply that these methods cannot be made efficient with some work, but that's beyond our scope.
> > >
> > > We are happy that you agree that sample complexity should not be the only dimension to compare different algorithms. We also appreciate your acknowledgement of our achievement in offline learning in the average reward setting. However, we think our contribution goes beyond tackling the average reward setting. To this end, and, with regards to our concrete contributions to the RL community, please refer to our response to your previous comments as well as paragraph 4 of our general response.
> > >
> > > Once again, thank you for acknowledging our efforts and reconsidering your evaluation of our work. We also appreciate you drawing our attention to the limitations, because, like with every research area, these will make a great foundation for future work.

---

> > > > ### Comment · Reviewer_zEPA · 2023-08-20
> > > >
> > > > Thank the authors for further clarification. Given your explanation of "oracle-based", I think this makes sense to me now. I would recommend authors make it more clear in the revision. Also, I think it would be better to add a column indicating whether the algorithms use the correct (or tight) choice of coverage notion when specializing in linear settings since it is one of the main contributions of this paper. When an algorithm uses a better notion of coverage, even though the sample complexity in terms of $n$ is slightly worse, it would still be valuable and could inspire future work to design algorithms with better coverage notion and better dependence in $n$.

---

> > > > > ### Author Response · Authors · 2023-08-21
> > > > >
> > > > > Thank you for the discussion, and recommendations!
> > > > > We are glad that we could help provide some clarity. Please let us know if you have further questions.

---

### Official Review · Reviewer_rDuz · 2023-07-27

**Soundness:** 3 good
**Presentation:** 3 good
**Contribution:** 2 fair
**Rating:** 5
**Confidence:** 4

**Summary:**

This paper studies offline reinforcement learning with linear function approximation. They propose a primal-dual algorithm, formulating linear RL into a minimax problem and solving it with gradient descent-ascent. Sample complexity analysis is provided for infinite-horizon discounted and average-reward MDPs, where the rate is $O(\frac{1}{\epsilon^4})$ for both settings.

**Strengths:**

1. The algorithm is primal-dual and thus easy to implement in practice.
2. The paper provides rigid theoretical analysis.

**Weaknesses:**

1.  The newly defined coverage ratio $C_{\phi,c}$ is a little strange when $c\neq \frac{1}{2}$. For example, when we choose $c=1$ and thus we don't need the knowledge of $\Lambda$, the coverage ratio $C_{\phi,1}=\sum_{x,a}(\frac{\mu^*(x,a)}{\mu_B(x,a)})^2$. Then when $\mu^*=\mu_B$, $C_{\phi,1}$ will become $|X||A|$. However, in the literature, when the behavior policy is the same as the optimal policy, the coverage is typically 1. The authors claim that we can estimate the $\Lambda$ via the offline dataset so that we can choose $c=\frac{1}{2}$, but do not provide any theoretical analysis about this point. I will be more convinced if the authors can give more rigid proofs for this method.
2. The sample complexity is worse than the typical rate $\frac{1}{\epsilon^2}$.

**Questions:**

None.

**Limitations:**

None.

---

> ### Author Rebuttal · Authors · 2023-08-09
>
>
> > 1. The newly defined coverage ratio $C_{\phi,c}$ is a little strange when $c\neq \frac{1}{2}$. For example, when we choose $c=1$ and thus we don't need the knowledge of $\Lambda$, the coverage ratio $C_{\phi,1}=\sum_{x,a} (\frac{\mu^*(x,a)}{\mu_B(x,a)})^2$. Then when $\mu^*=\mu_B$, $C_{\phi,1}$ will become $|X||A|$. However, in the literature, when the behavior policy is the same as the optimal policy, the coverage is typically 1. The authors claim that we can estimate the $\Lambda$ via the offline dataset so that we can choose $c=$, but do not provide any theoretical analysis about this point. I will be more convinced if the authors can give more rigid proofs for this method.
>
> Reducing the linear function approximation case to the tabular case can be useful to gain some intuitive understanding, however one must be very careful in drawing conclusions from it. The fundamental motivation behind the linear function approximation setting, is that the state space can be too big to handle. For this reason we replace every dependence to $|\mathcal{X}|$ with a dependence on the feature space dimension $d$, which is instead assumed to be small enough to be manageable. Reducing this setting to the tabular case implies taking $d=|\mathcal{X}||\mathcal{A}|$, which violates said assumption, unless we concede that $|\mathcal{X}|$ is now maneagable as well.
>
> Having said this, we anknowledge that our coverage ratio is proportional to $|\mathcal{X}||\mathcal{A}|$ in this case. However, this does not necessarily imply our bounds to be worse, because the coverage ratio should not be considered in isolation but in the context of the bounds it is used in; and in the setting we are now considering having a dependence on $|\mathcal{X}||\mathcal{A}|$ is definitely acceptable. Moreover, even if this resulted in a worse than optimal dependence on $|\mathcal{X}||\mathcal{A}|$, please keep in mind we are considering a degenerate case, and that we had to make design decision to tackle problems which are not present in the tabular case. To be more specific, in the tabular case one could very easily work with state-action occupancy measures and get rid of the double loop of our algorithm, immediately halving the sample complexity.
>
>
> On the subject of estimating $\Lambda$, we find it plausible that one can replace the exact feature covariance matrix with an estimate that is built either offline using some fraction of the overall sample budget, or online using all past data. In particular, in the tabular case, we could simply use all data to estimate the visitation probabilities of each-state action pairs and use them to build an estimator of $\Lambda$. Details of a similar approach have been worked out by Gabbianelli et al. (ALT 2023) for the bandit case. We believe that the same approach should work for more general linear function approximation as well, but that working out the details of such an approach would make our already rather complicated paper even more complex. An idea would be to directly estimate $\Lambda^{-1}$ using the matrix geometric resampling method proposed by Neu and Olkhovskaya (COLT 2020). Finally, we would like to remark that one of the appeals of our method is precisely that you can avoid this step and never estimate $\Lambda$, at the price of inflating the coverage ratio.
>
> > 2. The sample complexity is worse than the typical rate $\frac{1}{\epsilon^2}$.
>
> While it is true that the optimal sample complexity is $O(\varepsilon^{-2})$, it is certainly not *typical*, unless we are considering easier settings.

---

> > ### Comment · Reviewer_rDuz · 2023-08-17
> >
> > Thanks for the response! I don't think sacrificing the coverage ratio for avoiding estimating $\Lambda$ is a good idea since the authors have proposed a feasible method to estimate $\Lambda$, especially when the coverage ratio could be inflated many times potentially. I still recommend the authors to conduct a thorough analysis for the estimation process since the setting is different from the previous works after all and I will maintain my scores for now.

---

> > > ### Author Response · Authors · 2023-08-17
> > >
> > > Thank you for the feedback! We remain at disposal for further questions, and are hopeful you will take into consideration our general rebuttal as well, in your final evaluation.

---

### Official Review · Reviewer_6BYu · 2023-07-27

**Soundness:** 3 good
**Presentation:** 4 excellent
**Contribution:** 3 good
**Rating:** 5
**Confidence:** 3

**Summary:**

This paper considers the problem of offline reinforcement learning (RL) for linear Markov Decision Processes (MDPs) under the infinite-horizon discounted and average-reward settings. The authors propose a primal-dual optimization method based on the linear programming formulation of RL, which allows for efficient learning of near-optimal policies from a fixed dataset of transitions under partial coverage. The proposed algorithms improve the sample complexity compared to previous methods from $O(\epsilon^{-5})$ to $O(\epsilon^{-4})$ under the discounted setting and provide the first line of result in the average-reward setting with realizable linear function approximation and partial coverage.

**Strengths:**

1 The proposed algorithm improves existing algorithms in both statistical efficiency and computational efficiency under the discounted reward setting with the linear function approximation (we note the baseline may handle problems beyond the linear MDPs).

2 The algorithms presented in this paper do not explicitly leverage the principle of pessimism, but focus on the linear programming formulation of MDP, and rely on a new reparametrization trick extended from the tabular case. The technique itself seems to be novel to me.

3 The algorithms present the first line of work for the offline average-reward MDP.

4 The paper is easy to follow, with a thorough comparison with existing work that clearly positions the results in the literature.

**Weaknesses:**

1 I am confused about the requirement of $\Lambda$ to be invertible (line 140) as this seems to be very closely related to the uniform coverage condition where we assume that the smallest eigenvalue of $\Lambda$ is lower bounded from zero. I am wondering what is the key difference between them. Can you elaborate on this with some intuitions or examples?

2 The authors discuss the relationship between the coverage condition considered in this paper and that of [1] and show that the coverage condition is a low-variance version of the standard feature coverage ratio if $c=1/2$. However, in this case, the algorithm explicitly uses $\Lambda$, while the PEVI proposed in [1] does not. In contrast, $c=1$ leads to a worse bound but we do not need the knowledge of $\Lambda$. Could you provide a more detailed characterization or example to illustrate the difference between these two cases?


typo: line 328, $\epsilon^2 \to \epsilon^{-2}$

[1] is pessimism provably efficient for offline rl

**Questions:**

see weakness

**Limitations:**

yes

---

> ### Author Rebuttal · Authors · 2023-08-09
>
> > 1. I am confused about the requirement of $\Lambda$ to be invertible (line 140) as this seems to be very closely related to the uniform coverage condition where we assume that the smallest eigenvalue of $\Lambda$ is lower bounded from zero. I am wondering what is the key difference between them. Can you elaborate on this with some intuitions or examples?
>
> Thank you for raising this very interesting point!
>
> Indeed, in the tabular case $\Lambda$ reduces to $\mathrm{diag}(\mu_B)$ and therefore invertibility implies that the behavior policy explores the whole state-action space (uniform coverage). However, when considering linear function approximation, invertibility of $\Lambda$ just implies coverage of the *feature space*, which is very different than the usual notion of uniform coverage. Infact, many works use similar definitions of **partial** coverage where a $\Lambda^{-1}$ term appears (see our Discussion section and eq. 17 in particular).
>
> Nonetheless, your question is interesting, because it turns out we do not even require uniform coverage of the feature space!
>
> Certainly, if the behavior policy only covers a subspace of the feature space, $\Lambda$ is not invertible. However, this is only a problem if the "optimal direction" $\lambda^* = E_{\mu^*}[\phi(x,a)]$ does not belong to this subspace. In this case we cannot do much, and we must set the coverage to $+\infty$. On the other hand, when the optimal direction *does* belong to the subspace visited by the behavior policy, we can just focus on this subspace. Some other directions may not be seen in the data but this does not matter. This implies we can use the Moore-Penrose pseudoinverse $\Lambda^\dagger$ and the substitution $\lambda \gets \Lambda^\dagger\beta$. This is possible because when $\lambda^*$ is in the range of $\Lambda$, we have $\Lambda^\dagger\Lambda \lambda^* = \lambda^*$ (notice that $\Lambda^\dagger\Lambda$ projects any vector into the range of $\Lambda^T = \Lambda$). The coverage ratio thus becomes $C=(\lambda^*)^T\Lambda^\dagger\lambda^*$.
>
> We will mention this in the final version of the paper to avoid confusing future readers. Thank you again for bringing this up!
>
> > 2. The authors discuss the relationship between the coverage condition considered in this paper and that of [1] and show that the coverage condition is a low-variance version of the standard feature coverage ratio if $c=1/2$. However, in this case, the algorithm explicitly uses $\Lambda$, while the PEVI proposed in [1] does not. In contrast, $c=1$ leads to a worse bound but we do not need the knowledge of $\Lambda$. Could you provide a more detailed characterization or example to illustrate the difference between these two cases?
>
> In the general case, we don't know how to make the numerical comparison of the three ratios ($C_{\phi,1}$, $C_{\phi,1/2}$, and $C^\diamond$, the one from PEVI and Equation 17) more precise than what is already in the discussion section, which the reviewer has correctly summarized: $C_{\phi,1/2}$ is always smaller than $C^\diamond$, but $C_{\phi,1}$ and $C^\diamond$ are not comparable in general. However, we would like to remark a clear advantage of our ratio which holds for any value of $c$: the decoupling of the "optimal feature" vectors $\lambda^*=E_{\mu^*}[\phi(x,a)]$, that we have so far presented in terms of "variance reduction", implies that only the "average optimal direction" $\lambda^*$ needs to be covered by the behavior policy. In comparison, PEVI's ratio $C^\diamond$ requires the behavior policy to cover the whole subspace of features visited by the comparator policy. Using this fact, we can actually construct a small numerical example where $C^\diamond$ is arbitrarily larger than $C_{\phi,1}$:
>
> Consider a one-state linear MDP with four actions and a single relevant step (this can be realized in the discounted setting by adding an absorbing state). One can construct it in such a way that there are only 4 possible feature vectors: $\phi_1 = [1, 0]$, $\phi_2 = [1,1]$, $\phi_3 = [0, 1]$ and $\phi_4 = [-1, -1]$ (the feature dimension is $d=2$).
> The comparator policy visits $\phi_1$ with probability $1/2$ and $\phi_3$ with probability $1/2$.
> The behavior policy visits $\phi_1$ with probability $\epsilon$, $\phi_2$ with probability $(1-\epsilon)/2$, and $\phi_4$ with probability $(1-\epsilon)/2$, where $0<\epsilon<1/2$.
> By plotting the four feature vectors you can already see that the features visited by the optimal policy span all of $\mathbb{R^2}$, while for $\epsilon=0$ the features of the behavior policy would be confined to a $1$-dimensional subspace. However, the average optimal feature $[1/2, 1/2]$ belongs to this line. Indeed, you can check that PEVI's ratio is $C^\diamond = \frac{2-\epsilon}{2\epsilon(1-\epsilon)}$, which can be arbitrarily large for small values of $\epsilon$. Instead, our ratio $C_{\phi,1}=\frac{1}{4(1-\epsilon)^2} < 1$. Of course, $C_{\phi,1/2}=\frac{1}{4(1-\epsilon)}$ is even smaller.
>
> In practice, we expect this "single direction" property to make a big difference when the feature dimension $d$ is large. Finally we would like to remind that PEVI considers the finite-horizon setting, which is considerably easier than the one we consider.
>
> > typo: line 328, $\epsilon^{2}\to\epsilon^{-2}$
>
> Thanks for catching this!

---

> > ### Comment · Reviewer_6BYu · 2023-08-18
> > **thanks for the response**
> >
> > Thanks for the clarification. I would keep my score toward acceptance.

---

> > > ### Author Response · Authors · 2023-08-18
> > >
> > > Thank you for your positive inclination towards our work. We remain available in case you have further questions or concerns.

---

### Official Review · Reviewer_C22H · 2023-08-02

**Soundness:** 3 good
**Presentation:** 3 good
**Contribution:** 2 fair
**Rating:** 4
**Confidence:** 2

**Summary:**

The authors investigate offline RL in linear MDPs and introduce a novel LP-based method. They assert that their proposed approach achieves the lowest sample complexity of $O(1/\epsilon^4)$ among computationally efficient algorithms. In comparison, existing computationally efficient algorithms can achieve $O(1/\epsilon^5)$. Additionally, the author's theory can be extended to the average reward setting.

**Strengths:**

* To the best of the author’s knowledge, in offline linear MDPs, the result in the average-reward setting is novel.
* The LP formulation in linear MDPs is worthwhile to investigate


**Weaknesses:**

* I am uncertain about whether it is appropriate to claim that existing offline RL algorithms in linear MDPs achieve $O(1/\epsilon^5)$. It appears that [38] may have better sample complexity. In Table 1 of the manuscript, the author mentions that [38] cannot handle the discounted setting. However, extending from the finite-horizon to the discounted infinite-horizon setting is relatively straightforward. Hence, this comparison may not be entirely fair. If [38] indeed has better sample complexity, it significantly impacts the author's contribution. Thus, I currently rate the paper with a score of 4.

* I am not entirely certain about the significance of the extension to the average reward case.

* Presently, I cannot determine whether the reason [9] and [36] cannot handle the average reward case is due to the algorithms or their analysis. If this limitation arises from their analysis, their algorithm has the potential to be superior as it can handle more general MDPs.

**Questions:**

Could you provide further elaboration on my concerns regarding (1) the comparison between the author's work and [38], and (2) the third point in the weakness section?

**Limitations:**

They discussed.

---

> ### Author Rebuttal · Authors · 2023-08-09
>
> Thank you for the time spent reviewing our paper, and the feedback you provided!
>
> > * I am uncertain about whether it is appropriate to claim that existing offline RL algorithms in linear MDPs achieve $O(1/\epsilon^5)$. It appears that [38] may have better sample complexity. In Table 1 of the manuscript, the author mentions that [38] cannot handle the discounted setting. However, extending from the finite-horizon to the discounted infinite-horizon setting is relatively straightforward. Hence, this comparison may not be entirely fair. If [38] indeed has better sample complexity, it significantly impacts the author's contribution. Thus, I currently rate the paper with a score of 4.
>
> In [38], the authors do report a sample complexity of $O(\epsilon^{-2})$ when the data generation distribution (behavior policy) is fixed. Indeed, this is an improvement over our $O(\epsilon^{-4})$ guarantee. However, their method specified in Equation 10 exploits a backward induction technique which is only feasible in the finite horizon setting. It is unclear how such arguments can be generalized without compromising the rate to the infinite-horizon settings we consider in the present paper. More generally, extending results on linear MDPs beyond finite horizon, even in the online setting has been historically challenging.
>
> > * I am not entirely certain about the significance of the extension to the average reward case.
>
> Like many people in the RL theory community, we believe that the average-reward setting is the most challenging one. While challenge is obviously not equal to significance, we find it remarkable that our approach works in this setup just as well as in the discounted-reward setting. Indeed, as illustrated by Table 1, most previous work in the literature is specific to either the finite-horizon case or the discounted setting, and transferring ideas from one setting to the other is typically hard (especially when going from finite horizon to discounted). In contrast, our approach readily works for two of the most challenging settings, which we believe is a significant contribution. More generally, we believe that most real-world RL problems are most naturally formulated in one of these two infinite-horizon frameworks, and that the finite-horizon framework, which is very commonly studied in theory, is insufficient for addressing most tasks of practical interest.
>
> > * Presently, I cannot determine whether the reason [9] and [36] cannot handle the average reward case is due to the algorithms or their analysis. If this limitation arises from their analysis, their algorithm has the potential to be superior as it can handle more general MDPs.
>
> The algorithms of [9] and [36] are based on ideas from the approximate dynamic programming literature, which are often limited to the discounted setting. Concretely, their approach makes crucial use of the contractive property of the discounted Bellman operators, which does not generally hold in the average-reward setting (especially not under the general assumptions we make in our work). This not only limits their analysis but also the applicability of the algorithm that relies on a policy evaluation procedure that is not readily available for average-reward MDPs.

---

> > ### Comment · Reviewer_C22H · 2023-08-16
> >
> > Thank you for your comprehensive responses. I still have some uncertainty regarding the statement regarding [38], "which is only feasible in the finite horizon setting." It appears to me that an analogous approach could be applicable to the infinite horizon discounted setting as well. I will maintain my current evaluation score and ask Area Chair to check this point. While I am retaining my score presently, I am open to raising the score later if I am incorrect.

---

> > > ### Author Response · Authors · 2023-08-17
> > >
> > > Thank you for the response, and for being open to raise your score.
> > >
> > > Your initial question prompted us to reason on how to extend the cited work to our setting. However, we were not able to prove a *straightforward* reduction (see below for details). For this reason, we suspect that if such a trivial reduction were possible, it would have been mentioned directly by the authors.
> > >
> > > The central part of the algorithm of Zanette et al (Algorithm 1) is the critic procedure `PLSPE` (shown in Algorithm 2), which is invoked at each iteration $t$ of the main loop. This procedure aims to solve a convex problem (Eq. 10a to 10c) which requires optimizing variables of size $dH$ and has a number of constraints proportional to $H$. Thus, it is impossible to run this procedure "as is" in the setting we consider, because in our case $H=\infty$. One idea could be to try to replace $H$ with the effective-horizon $1/(1-\gamma)$. This might be a good approach, but it is not immediately clear if it would work, and it also presents some downsides. For example, it would result in a non-stationary policy, which in practice is something very unappealing (i.e., imagine having to store 99 neural networks as opposed to 1). Moreover, the effective-horizon is just the average length of an "episode", and it would be unclear which policy to use when the interaction lasts longer than $1/(1-\gamma)$. Even addressing all of these concerns, it remains questionable if their algorithm would give us any advantage in terms of computational efficiency, since it requires computing a quantity proportional to $(d|A|n)$ at each iteration, where $n$ is the size of the dataset (see Eq. 10b).
> > >
> > > Obviously, this does not imply that extending the work of Zanette et al to the infinite-horizon setting is impossible. In fact we are confident that by substantially changing their method it must be possible. For this reason, if you have some specific approach in mind, we encourage you to share it with us, so that we can discuss it together.
> > >
> > >
> > > As an alternative approach, one could also try to completely replace the critic procedure of Zanette et al with something specifically designed for the infinite-horizon setting. In our opinion, this could result in something similar to PSPI (see Algorithm 1 of [Xie et al][1]), which uses a pessimistic policy evaluation procedure, which can be efficiently implemented with a least-squares based approach. However, the resulting method does not attain the optimal sample complexity either. Please refer to our answer to reviewer 7fb2 for a more detailed comparison of our method and PSPI.
> > >
> > > We hope to have helped addressing your uncertainty, and remain available for further questions.
> > >
> > >
> > >
> > >
> > > [1]: https://arxiv.org/pdf/2106.06926.pdf

---

### Author Rebuttal · Authors · 2023-08-09

We thank all reviewers for the work invested into evaluating our paper, and the thoughtful feedback they shared with us. We are also glad about the many strengths of our work which have been highlighted, signifying that the amount of effort we put in this paper has not gone unnoticed.

No work comes without limitations, and overall, we also agree about most of the weaknesses which have been pointed out. In particular, our work is limited to a very specific setting (linear MDPs) while there are works which study more general settings; and we only provide bounds in expectation, as opposed to more desirable high probabilities bounds. We believe these are all interesting directions for future work, which however do not overshadow the significance of our contribution.

Another important factor emerged from the reviews: our comparison with the state of the art could use some small improvements. Luckily, this is something we can easily fix in our final draft, thanks to the feedback shared by the reviewers. We already tried to address all the concerns, as best as possible, in our answers, and hope to have shed some more clarity. The main source of confusion seemed to be our table, which we updated and attached to this answer as a PDF (as suggested by reviewers we fixed some of the rates and added some missing references). It is indeed non-trivial to compare with many different methods, across different settings and assumptions. Anyhow, the questions we received gave us the chance to dig deeper into some of the related works, which in turn strengthened our conviction that our method is very competitive in the settings it is designed for. It is one of the few -- the only we are aware of actually -- computationally efficient method for linear MDPs with *proven* sample complexity guarantees, and as far as we are aware, the first to tackle the average reward setting. However, at the same time, this process made us realize that we probably gave too much importance to the comparison of rates and constants -- tables are good for an overview, but research cannot be reduced to a table. By doing so, we took away some of the light from what we believe are some of the great strengths of our contribution.


For this reason we wish to emphasize that, in our minds, one of the main values of our work lies in the novelty of our approach. While there are other LP-based approaches out there (as some of the reviewers pointed out), these use completely different parametrizations to deal with function approximation. Our parametrization is derived from different principles and we have demonstrated that it can provide bounds that scale with the *correct* comparator-dependent quantity without any pessimistic adjustment (see more about this below). This is a conceptual novelty that we think should be appreciated, and we strongly feel that it is worth sharing these ideas with the community. Indeed, the fact that such a new approach does not immediately attain the best possible sample rate (similarly to other computationally efficient algorithms) is not surprising, and rejecting new ideas because they do not immediately lead to strict improvements along **every** possible dimension is arguably rather shortsighted.


Besides the originality of our method, we would like to point out another significant advantage over previous works that has to do with our definition of coverage ratio. Although we did provide a detailed comparison of different notions of coverage in Section 6 of the paper, additional insights that we gained since the submission allow us to say that our definition of coverage ratio is better in a fundamental way than the more common definitions, and in fact is *the* correct notion for linear function approximation.
This kind of coverage ratio appeared before in the literature, for instance for finite-horizon MDPs (see "Provable Benefits of Actor-Critic Methods" by Zanette et al. (2021)), but never, to our knowledge, for the settings we consider. To keep this remark short, let us compare our ratio from Definition 3.1. when $c=1/2$ with the more common ratio from Equation 17. You can easily see that our ratio only requires the behavior policy to cover well *a single direction* in feature space, namely the expected feature vector under the optimal policy. In comparison, the ratio from Equation 17 requires the behavior policy to cover well *the entire subspace* spanned by features visited by the optimal policy, which is much more demanding. Our "single-direction" property continues to hold when $c=1$. So, although we cannot prove that our ratio $C_{\phi,1}$ is always smaller than other versions, one can use this geometric property to build examples where the classic ratio (eq. 17) is arbitrarily larger than $C_{\phi,1}$ (see the answer to Reviewer 6BYu for one such example). In practice, we expect this property to make a big difference when the feature dimension $d$ is large. To fully complement our discussion from Section 6, note that the ratio from Equation 18, which is the "linear" specialization of the notion of coverage that is commonly considered in works on general function approximation (such as Xie et al.), does *not* display the single-direction property either.

---

> ### Comment · Area_Chair_VwKC · 2023-08-17
>
> Quick comment on your last point about coverage ratio: I think more and more people in the community are realizing that this is indeed the right def for linear MDPs, that you only need to cover one direction. See page 10 of https://arxiv.org/pdf/2203.05804.pdf, which has a closely related discussion, and a twitter thread https://twitter.com/nanjiang_cs/status/1666994607073230848.

---

> > ### Author Response · Authors · 2023-08-17
> >
> > Thank you for the pointers!

---

### Decision · Program_Chairs · 2023-09-21

**Decision:**

Reject

**Comment:**

The paper develops a new algorithm and analysis for offline RL in linear MDPs based on the primal-dual formulation. The discussion during rebuttal phase were centered around fair comparisons to related works, as the original manuscript seems leaning too conservative when it comes to interpreting other works' results; multiple reviewers gave good comments about how to improve the comparisons, and the authors demonstrated good understanding of the literature and I am confident that they will be able to make the needed changes.

At the final discussion phase though, the requirement of knowledge of $\Lambda$ in the $c=1/2$ case is still a major weakness that puts the paper as borderline (see details below). Multiple reviewers who voted positively or increased their scores during rebuttal did not notice such an issue, and some changed their mind when this was brought up during the final discussion. Most reviewers agree that the paper should be accepted had the $c=1/2$ case been fully worked out.

The paper has two results: a complete result for $c=1$ that does not require the knowledge of $\Lambda$ but comes with a worse notion of coverage, and a result for $c=1/2$ that requires knowledge of $\Lambda$ and comes with a superior notion of coverage. As reviewer rDuz pointed out, the coverage coefficient for the $c=1$ case isn't really very attractive given its scale in the tabular case; even if we do not worry about the tabular case (as argued in the rebuttal to rDuz), $c=1$ does not give the most satisfying result, given that the discussion in Section 6 is mostly advocating for the superiority of the coverage coefficient under $c=1/2$, which was the main reason that reviewer zEPA increased their score.

Regarding this caveat, the author wrote in rebuttal that

> ... working out the details of such an approach would make our already rather complicated paper even more complex.

It is true that theory papers often get reviewer requests to consider complicated extensions that add little to the main message of the paper, but we think the situation here is different. The most competitive result ($c=1/2$) crucially relies on the assumption of knowing $\Lambda$; I think we can all agree this is not reasonable. Given this, it is not unfair to say that the $c=1/2$ result is incomplete. The authors have sketched approaches for estimating $\Lambda$, but it is unclear if replacing the exact knowledge with estimate would increase the sample complexity, which is generally plausible unless one carefully works out the details. If the authors are very sure it wouldn't, why not just work it out? I don't think the added complexity is a good excuse here, since if the details of estimating $\Lambda$ turns out to be routine and "boring", the authors can leave the main paper largely intact and push the complications into the appendix. The sentiment from the reviewers is that we'd rather see a complete result for $c=1/2$ even at the cost of 20 more pages in the appendices.

Now, I did consider the question of whether the paper can be accepted even given the incompleteness of the $c=1/2$ result, along the lines of what authors wrote in the rebuttal:

> [the table of comparison] took away some of the light from what we believe are some of the great strengths of our contribution.

> one of the main values of our work lies in the novelty of our approach... Our parametrization is derived from different principles and we have demonstrated that it can provide bounds that scale with the correct comparator-dependent quantity...

To accept the paper based on this line of reasoning, I think the paper needs to do a better job in highlighting the technical novelty in writing, which is difficult to do in a well-studied model like linear MDPs. The easy way to sell a technique is to appeal to the novel guarantees it leads to (like what the authors did in the rebuttal: "it can provide bounds..."), which still depends on the completeness of the $c=1/2$ result. Another selling point is the average-reward guarantees, but the writing of the paper does not make it a highlight or priority: it presents everything in the discounted case, and only extends to the average-reward setting at the end, making it look like an after-thought. Plus as Reviewer C22H pointed out, it is not unreasonable to suspect that extension to the finite-horizon setting can be similarly done in other related works.